# Trained Mamba Emulates Online Gradient Descent in In-Context Linear Regression

**Jiarui Jiang**[*1], **Wei Huang**[*2], **Miao Zhang**[†1], **Taiji Suzuki**[3 2], **Liqiang Nie**[1]

[1]Harbin Institute of Technology, Shenzhen
[2]RIKEN AIP
[3]University of Tokyo
jiaruij@outlook.com, wei.huang.vr@riken.jp, zhangmiao@hit.edu.cn
taiji@mist.i.u-tokyo.ac.jp, nieliqiang@gmail.com

## Abstract

State-space models (SSMs), particularly Mamba, emerge as an efficient Transformer alternative with linear complexity for long-sequence modeling. Recent empirical works demonstrate Mamba's in-context learning (ICL) capabilities competitive with Transformers, a critical capacity for large foundation models. However, theoretical understanding of Mamba's ICL remains limited, restricting deeper insights into its underlying mechanisms. Even fundamental tasks such as linear regression ICL, widely studied as a standard theoretical benchmark for Transformers, have not been thoroughly analyzed in the context of Mamba. To address this gap, we study the training dynamics of Mamba on the linear regression ICL task. By developing novel techniques tackling non-convex optimization with gradient descent related to Mamba's structure, we establish an exponential convergence rate to ICL solution, and derive a loss bound that is comparable to Transformer's. Importantly, our results reveal that Mamba can perform a variant of *online gradient descent* to learn the latent function in context. This mechanism is different from that of Transformer, which is typically understood to achieve ICL through gradient descent emulation. The theoretical results are verified by experimental simulation.

## 1 Introduction

State-space models (SSMs), notably Mamba (Gu and Dao, 2024), have recently emerged as compelling alternatives to Transformer-based architectures (Vaswani et al., 2017). Mamba integrates gating, convolutions, and state-space modeling with selection mechanisms, enabling linear-time complexity. This effectively addresses the quadratic computational costs typically associated with self-attention mechanisms in Transformers. Consequently, Mamba demonstrates superior efficiency in processing long sequences while maintaining or even surpassing Transformer performance across diverse benchmarks (Gu and Dao, 2024; Dao and Gu, 2024; Patro and Agneeswaran, 2024; Liu et al., 2024; Ahamed and Cheng, 2024; Li et al., 2024a,b).

In-context learning (ICL) (Brown et al., 2020) is a powerful paradigm that enables models to generalize to unseen tasks by dynamically leveraging contextual examples (such as input-output pairs) without task-specific fine-tuning. This capability has become a defining characteristic of large foundation models, significantly enhancing their flexibility and adaptability. While extensive research has provided substantial insights into Transformer-based ICL mechanisms (Garg et al., 2022; Gatmiry et al., 2024; Sander et al., 2024; Zheng et al., 2024; Zhang et al., 2025), the principles underlying

---

[*]Equal contribution
[†]Corresponding author

39th Conference on Neural Information Processing Systems (NeurIPS 2025).

Mamba's ability to perform in-context learning remain largely unexplored, highlighting a compelling research gap.

Recent empirical studies have examined Mamba's (ICL) capabilities, showing it matches Transformers on many standard ICL tasks, while surpassing them in specialized scenarios like sparse parity (Park et al., 2024; Grazzi et al., 2024). Bondaschi et al. (2025) theoretically analyzed its representational capacity for in-context learning of Markov chains, and Li et al. (2025a) investigated binary classification tasks with outliers. (Yang et al., 2024, 2025; Behrouz et al., 2025b,a) leverage the connection between SSMs and online learning to design new architectures. However, even the linear regression model, a canonical setting widely used for theoretical analysis of Transformer-based ICL mechanisms, remains theoretically underexplored in the context of Mamba. To fill this gap, we analyze Mamba's training dynamics on in-context linear regression tasks. More precisely, following the previous ICL analysis in Transformers (Garg et al., 2022; Zhang et al., 2024; Ahn et al., 2023), this paper focuses on a data generative model with $N$ input-output pairs ($\{\boldsymbol{x}_i, y_i\}_{i=1}^N$) and a query input ($\boldsymbol{x}_q$) satisfying $y = f(\boldsymbol{x}) = \boldsymbol{w}^\top \boldsymbol{x}$, where $\boldsymbol{x}$ denotes the input and $y$ denotes the output, and $\boldsymbol{w}$ is randomly sampled from Gaussian distribution, termed the *context*. In this work, we develop a rigorous theoretical framework to analyze how randomly initialized Mamba models, when trained through gradient descent, evolve to implement in-context learning. We demonstrate that the trained Mamba architecture dynamically leverages the input context to perform implicit estimation of the vector $\boldsymbol{w}$. This estimation is achieved through hidden state updates that mimic online gradient descent steps, finally implementing prediction for $y_q = f(\boldsymbol{x}_q) = \boldsymbol{w}^\top \boldsymbol{x}_q$. We also provide a loss bound that is comparable to Transformers'. Our contributions are summarized as follows:

- We construct a Mamba architecture (S6: S4 with selection) capable of ICL, establishing its exponential convergence rate to ICL solution, and further derive the loss bound after convergence. The loss matches that of Transformers.

- Technically, we develop novel techniques to address optimization challenges induced by random initialization and gradient descent, rigorously characterizing Mamba's training dynamics when trained from scratch.

- We reveal how trained Mamba achieves in-context linear regression by progressively aligning its hidden states with the *context* through sequential token processing. This finding provides a new perspective for understanding Mamba's ICL mechanism, distinct from Transformer-based approaches. All the above results are verified by experiments.

## 2 Related Work

**In-Context Learning**    The seminal work of Brown et al. (2020) demonstrated the in-context learning capability in Transformers, showing their ability to infer functional mappings from input-output exemplars without weight updates. Garg et al. (2022) initiated the investigation of ICL from the perspective of learning particular function classes. Following these, a line of research analyze this phenomenon through the lens of algorithm imitation: Transformers can be trained to implement various learning algorithms that can mimic the latent functions in context, including: a single step of gradient descent (Von Oswald et al., 2023; Akyürek et al., 2023), statistical algorithms (Bai et al., 2023), reinforcement learning algorithm (Lin et al., 2024), multi-step gradient descent (Gatmiry et al., 2024), mesa-optimization (Zheng et al., 2024), Newton's method (Giannou et al., 2025), weighted preconditioned gradient descent (Li et al., 2025b), in context classification (Bu et al., 2024; Shen et al., 2024; Bu et al., 2025) among others.

Recent work extends ICL analysis beyond Transformers: (Lee et al., 2024; Park et al., 2024) empirically compared popular architectures (e.g., RNNs, CNNs, SSMs, Transformers) on synthetic ICL tasks, identifying capability variations across model types and task demands. Tong and Pehlevan (2024) demonstrate that MLPs can learn in-context a series of classical tasks such as regression and classification with less computation than Transformers. Sushma et al. (2024) show that state space models augmented with *local self-attention* can learn linear regression in-context. Unlike existing research on ICL, this work focuses on the ICL mechanism of Mamba (specifically S4 with selection) and its training dynamics.

**Theoretical Understanding of SSMs**    As Gu et al. (2022) introduce structured state spaces models in modeling long sequence and further be extended to Mamba (Gu and Dao, 2024), which gained

significant attention as alternatives to Transformers, extensive research has sought to theoretically understand the mechanisms and capabilities of state-spaces models (SSMs). Dao and Gu (2024) propose the framework of state space duality, which establishes a connection between SSMs and attention variants through the lens of structured matrices. Vankadara et al. (2024) provide a scaling analysis of signal propagation in SSMs through the lens of feature learning. Cirone et al. (2024) draw the link of SSMs to linear CDEs (controlled differential equations) and use tools from rough path theory to study their expressivity. Chen et al. (2025) establish the computational limits of SSMs and Mamba via circuit complexity analysis, questioning the prevailing belief that Mamba possesses superior computational expressivity compared to Transformers. Nishikawa and Suzuki (2025) demonstrate that state space models integrated with nonlinear layers achieve dynamic token selection capabilities comparable to Transformers. Different from the above, we provide theoretical understanding of Mamba from the perspective of ICL.

## 3 Problem Setup

In this section, we outline the ICL data model, the Mamba model, the prediction strategy, and the gradient descent training algorithm.

**Data Model.** We consider an in-context linear regression task where each prompt corresponds to a new function $f(\boldsymbol{x}) = \boldsymbol{w}^\top \boldsymbol{x}$ with weights $\boldsymbol{w} \sim \mathcal{N}(0, \boldsymbol{I}_d)$ and $d > 1$. For each task, we generate $N$ i.i.d. input-output pairs $\{(\boldsymbol{x}_i, y_i)\}_{i=1}^N$ and a query $\boldsymbol{x}_q$, where all inputs $\boldsymbol{x}_i, \boldsymbol{x}_q \sim \mathcal{N}(0, \boldsymbol{I}_d)$ are independent Gaussian vectors, and the outputs satisfy $y_i = f(\boldsymbol{x}_i)$. The goal is to predict $y_q = f(\boldsymbol{x}_q)$ for the query.

To enable sequential processing of prompts in the Mamba model, we implement an embedding strategy where:

1. The $i$-th context token is encoded as $\boldsymbol{e}_i = (\boldsymbol{x}_i^\top, y_i)^\top$, formed by concatenating input $\boldsymbol{x}_i$ with its corresponding label $y_i$.

2. The query token is represented as $\boldsymbol{e}_q = (\boldsymbol{x}_q^\top, 0)^\top$, masking the unknown target value with a zero placeholder.

In many theoretical analyses of Transformer-based in-context learning, token embeddings are conventionally concatenated into a single matrix to enable parallel computation of global attention (Zhang et al., 2024; Ahn et al., 2023; Huang et al., 2023; Mahankali et al., 2024; Wu et al., 2024). In contrast, since Mamba operates as a sequential model, we feed the embeddings of context tokens one by one, and finally the query token ($\boldsymbol{e}_1 \to \boldsymbol{e}_2 \to \cdots \to \boldsymbol{e}_N \to \boldsymbol{e}_q$).

**Mamba Model.** We consider a S6 layer of Mamba $\boldsymbol{o}_{1:L} = \mathbf{Mamba}(\boldsymbol{\theta}; \boldsymbol{u}_{1:L})$ with selection, discretization, and linear recurrence components, where $\boldsymbol{u}_l, \boldsymbol{o}_l \in \mathbb{R}^{d_e}$. It can be described as follows:

$$\boldsymbol{h}_l^{(i)} = \overline{\boldsymbol{A}}_l \boldsymbol{h}_{l-1}^{(i)} + \overline{\boldsymbol{B}}_l u_l^{(i)} \in \mathbb{R}^{d_h \times 1}, \quad (1a) \quad \overline{\boldsymbol{A}}_l = \exp(\Delta_l \boldsymbol{A}) \in \mathbb{R}^{d_h \times d_h}, \tag{2a}$$

$$o_l^{(i)} = \boldsymbol{C}_l^\top \boldsymbol{h}_l^{(i)}, \quad \boldsymbol{C}_l \in \mathbb{R}^{d_h \times 1}, \quad (1b) \quad \overline{\boldsymbol{B}}_l = (\Delta_l \boldsymbol{A})^{-1}(\exp(\Delta_l \boldsymbol{A}) - \boldsymbol{I})\Delta_l \boldsymbol{B}_l \in \mathbb{R}^{d_h \times 1} \tag{2b}$$

for $i \in [d_e]$. Here, the superscript $(i)$ denotes the $i$-th independent processing channel, where each channel operates on a unique feature dimension of the input $\boldsymbol{u}_l$ and output $\boldsymbol{o}_l$ vectors (i.e., $u_l^{(i)}$ and $o_l^{(i)}$ correspond to the $i$-th elements of $\boldsymbol{u}_l$ and $\boldsymbol{o}_l$, respectively). The hidden state $\boldsymbol{h}_l^{(i)}$ is initialized as $\boldsymbol{h}_0^{(i)} = \boldsymbol{0}$ and evolves according to $\overline{\boldsymbol{A}}_l \in \mathbb{R}^{d_h \times d_h}, \overline{\boldsymbol{B}}_l \in \mathbb{R}^{d_h \times 1}$ and the input $u_l^{(i)}$. $\boldsymbol{C}_l \in \mathbb{R}^{d_h \times 1}$ maps the hidden state $\boldsymbol{h}_l^{(i)}$ to the output $o_l^{(i)}$. As shown in (2), $\overline{\boldsymbol{A}}_l$ and $\overline{\boldsymbol{B}}_l$ are computed using the zero-order hold (ZOH) discretization method applied to $\boldsymbol{A} \in \mathbb{R}^{d_h \times d_h}, \boldsymbol{B}_l \in \mathbb{R}^{d_h \times 1}$ and the timestep $\Delta_l \in \mathbb{R}$. Next, we describe the selection mechanism.

$$\boldsymbol{B}_l = \boldsymbol{W}_B \boldsymbol{u}_l + \boldsymbol{b}_B, \quad (3) \quad \boldsymbol{C}_l = \boldsymbol{W}_C \boldsymbol{u}_l + \boldsymbol{b}_C, \quad (4) \quad \Delta_l = \text{softplus}(\boldsymbol{w}_\Delta^\top \boldsymbol{u}_l + b_\Delta), \quad (5)$$

Here, $\text{softplus(x)} = \log(1 + \exp(x))$. $\boldsymbol{W}_B, \boldsymbol{W}_C \in \mathbb{R}^{d_h \times d_e}$, $\boldsymbol{b}_B, \boldsymbol{b}_C \in \mathbb{R}^{d_h \times 1}$, $\boldsymbol{w}_\Delta \in \mathbb{R}^{d_e \times 1}$, $b_\Delta \in \mathbb{R}$, along with $\boldsymbol{A} \in \mathbb{R}^{d_h \times d_h}$ are the parameters of the Mamba model. We use $\boldsymbol{\theta}$ to denote the collection of all the parameters.

Unlike previous work (Sushma et al., 2024) that introduce *local self-attention* component to augment SSMs, which may inherit the Transformer's ICL ability, our model adheres to Mamba's original selective state-space framework (Gu and Dao, 2024). This alignment ensures us to mechanistically analyze how Mamba's architecture enables in-context learning (ICL).

**Linear Regression Prediction.**    In this work, we set $d_e = d + 1$, enabling the Mamba model to process the embeddings $e_{1:N}, e_q$. Given the prompt $(e_1, \ldots, e_N, e_q)$, the Mamba model will output a sequence $o_{1:N+1} = \mathbf{Mamba}(\theta; e_1, \ldots, e_N, e_q)$. The prediction for the linear regression target $y_q = w^\top x_q$ is extracted from the terminal position of the output matrix (corresponding to the zero placeholder in the query token $e_q = (x_q^\top, 0)^\top$). Concretely, $\hat{y}_q = o_{N+1}^{(d+1)}$.

**Training Algorithm.**    To train a Mamba model over the in-context linear regression task, we consider minimizing the following population loss:

$$\mathcal{L}(\theta) = \mathbb{E}_{x_{1:N}, x_q, w} \left[ \frac{1}{2} (\hat{y}_q - y_q)^2 \right]. \tag{6}$$

Given a Mamba model, we use gradient descent to minimize population loss $\mathcal{L}(\theta)$, and the update of trainable parameters $\theta' = \{W_B, W_C, b_B, b_C\}$ can be written as follows:

$$\theta'(t+1) = \theta'(t) - \eta \nabla_{\theta'} \mathcal{L}(\theta(t)). \tag{7}$$

# 4   Main Results

This section presents our main theoretical results that characterize the convergence state of Mamba and its final loss. We also compare the results with other models.

**Assumption 4.1** *(1) Matrix $A = -I_{d_h}$. (2) The vector $w_\Delta$ is fixed as zero $\mathbf{0}$, and $b_\Delta$ is fixed as $\ln(\exp((\ln 2)/N) - 1)$. (3) Matrices $W_B$, $W_C$ are initialized with entries drawn i.i.d. from the standard Gaussian distribution $\mathcal{N}(0,1)$. (4) The hidden state dimension satisfies: $d_h = \widetilde{\Omega}(d^2)$. (5) The learning rate satisfies: $\eta = O(d^{-2} d_h^{-1})$. (6) Bias vectors $b_B$, $b_C$ are initialized as zero $\mathbf{0}$. (7) Token length $N = \Omega(d)$.*

(1) The negative-definite matrix $A = -I_{d_h}$ guarantees the stable convergence of hidden states $h_l^{(i)}$. (2) Given the zero-mean and symmetric distribution of embeddings, $w_\Delta$ can naturally converge to $\mathbf{0}$ during gradient descent, and we fix it as $\mathbf{0}$ for simplicity. We further fix $b_\Delta$ to an appropriate constant to maintain a suitable timestep $\Delta_l$, enabling us to concentrate our theoretical analysis on $W_B$, $W_C$, $b_B$, and $b_C$. In prior works on Transformer-based in-context learning, merging key-query weights (e.g., $W := W_Q W_K$) and specific initializations (e.g., $W_Q = W_K = I$) are often adopted to simplify optimization analysis (Zhang et al., 2024; Ahn et al., 2023; Huang et al., 2023; Mahankali et al., 2024; Wu et al., 2024). (3, 4, 5) In contrast, our Gaussian initialization of $W_B$ and $W_C$ demonstrates more practicality, which requires a sufficiently large hidden state dimension $d_h$ and a sufficiently small learning rate $\eta$ to ensure favorable loss landscape properties. Assumption (6) is intended to simplify the analysis. (7) Token length should be larger enough than the dimension of $w$ to capture sufficient contextual information.

**Theorem 4.1** *Under Assumption 4.1, if the Mamba is trained with gradient descent, and given a new prompt $(e_1, \ldots, e_N, e_q)$, then with probability at least $1 - \delta$ for some $\delta \in (0, 1)$, the trainable parameters $\theta'(t) = \{W_B(t), W_C(t), b_B(t), b_C(t)\}$ converge as $t \to \infty$ to parameters that satisfies:*

*(a) Projected hidden state: $(W_C^\top)_{[1:d,:]}(t) h_l^{(d+1)} = \alpha (W_C^\top(t))_{[1:d,:]} h_{l-1}^{(d+1)} + (1 - \alpha) \beta y_l x_l$,*

*(b) Prediction for target: $\hat{y}_q = x_q^\top \sum_{i=0}^{N-1} (1 - \alpha) \alpha^{i+1} \beta y_{N-i} x_{N-i}$,*

*(c) Population loss: $\mathcal{L}(\theta(t)) \leq \frac{3d(d+1)}{2N}$,*

*where $\alpha = \exp((-\ln 2)/N)$, $\beta = \frac{2(1+\alpha)}{\alpha \left( 3(1-\alpha)d + 4 - 2\alpha \right)}$.*

Theorem 4.1 characterizes the in-context learning (ICL) mechanism of Mamba and establishes an upper bound on its population loss. Specifically, (Thm 4.1 (a)) shows how the hidden state is updated according the given prompt $\boldsymbol{e}_l = (\boldsymbol{x}_l^\top, y_l)^\top$. (Thm 4.1 (b)) presents the final prediction given prompt $(\boldsymbol{e}_1, \ldots, \boldsymbol{e}_N, \boldsymbol{e}_q)$. (Thm 4.1 (c)) provides the upper bound for the population loss, which is comparable to that of the Transformer (Zhang et al., 2024). Next, we'll discuss it in more detail.

**Update of Hidden State.** If we define $\tilde{\boldsymbol{h}}_l := (\boldsymbol{W}_C^\top)_{[1:d,:]} \boldsymbol{h}_l^{(d+1)}$, then (Thm 4.1 (a)) can be rewritten as follows:
$$\tilde{\boldsymbol{h}}_l = \alpha \tilde{\boldsymbol{h}}_{l-1} + (1-\alpha)\beta y_l \boldsymbol{x}_l = \tilde{\boldsymbol{h}}_{l-1} + (1-\alpha)(\beta y_l \boldsymbol{x}_l - \tilde{\boldsymbol{h}}_{l-1}). \tag{8}$$
We observe its intrinsic connection to **online gradient descent**, which updates the model parameters $(\tilde{\boldsymbol{h}}_l)$ with only one currently arriving sample $(\boldsymbol{e}_l = (\boldsymbol{x}_l^\top, y_l)^\top)$ at each step. Specifically, the system gradually updates $\tilde{\boldsymbol{h}}_l$ along the pseudo-gradient direction $\beta y_l \boldsymbol{x}_l$, with a fixed step size $(1-\alpha)$.

For a newly defined task $f(\boldsymbol{x}) = \boldsymbol{w}^\top \boldsymbol{x}$, given that $\mathbb{E}[y_l \boldsymbol{x}_l] = \boldsymbol{w}$, the direction of $\tilde{\boldsymbol{h}}_l$ converges toward $\boldsymbol{w}$ as mamba processes multiple prompts. This demonstrates mamba's ability to internalize $f(\boldsymbol{x})$ through prompt processing, which ultimately ensures that predictions for query token $\boldsymbol{e}_q = (\boldsymbol{x}_q^\top, 0)^\top$ closely approximate $f(\boldsymbol{x}_q)$.

Previous works have shown that Transformer can mimic a single step of gradient descent to achieve in-context learning ability (Zhang et al., 2024; Mahankali et al., 2024). Concretely, a trained Transformer can be described as follows

$$\textbf{Transformer}(\boldsymbol{e}_1, \ldots, \boldsymbol{e}_N, \boldsymbol{e}_q) \approx \boldsymbol{x}_q^\top \Big( \frac{1}{N} \sum_{i=1}^N y_i \boldsymbol{x}_i \Big) \approx \boldsymbol{x}_q^\top \boldsymbol{w}. \tag{9}$$

Our theoretical analysis reveals that Mamba and Transformer have different in-context learning mechanisms. This divergence stems from their inherent architectural biases: Transformers process contexts globally through self-attention, while Mamba enforces local sequential dependencies via recurrent state transitions. These findings provide fundamental insights into the contrasting capabilities of Transformer-based and Mamba-based models for in-context learning. As experimental work shows, transformers can learn vector-valued MQAR tasks in the context which Mamba cannot, while Mamba succeeds in sparse-parity in-context learning tasks where Transformers fail(Park et al., 2024).

**Prediction Outcome.** Comparing equations Thm 4.1 (b) and (9), we found both similarities and distinctions in how Transformer and Mamba implement in-context learning (ICL). Both models leverage a weighted aggregation of $y_i \boldsymbol{x}_i$, aligning with the intuition that learning $f(\boldsymbol{x}) = \boldsymbol{x}^\top \boldsymbol{w}$ from context reduces to estimating the latent parameter $\boldsymbol{w}$, since $\mathbb{E}[y_i \boldsymbol{x}_i] = \boldsymbol{w}$. Notably, their token weighting strategies diverge: Transformer's global attention mechanism implicitly assigns nearly uniform weights ($\sim \frac{1}{N}$, where $N$ is the token length) to all $y_i \boldsymbol{x}_i$, while Mamba's linear recurrence imposes position-dependent weight variations. This difference arises from Mamba's iterative state update rule, where the influence of prompt tokens $\boldsymbol{e}_i$ on the hidden state $\boldsymbol{h}_l$ depends on their sequential placement, governed by the model's linear recurrence dynamics.

The derived upper bound (Thm 4.1 (c)) establishes an $O(1/N)$ convergence rate for the loss (ignoring dimension factor), demonstrating that Mamba matches the sample complexity scaling of Transformers in linear regression ICL tasks (Zhang et al., 2024).

**Compare with S4.** Mamba extends the structured state space model (S4) (Gu et al., 2022) by integrating a selection mechanism, which is critical for enabling ICL. In S4 model, the matrices $\boldsymbol{A} \in \mathbb{R}^{d_h \times d_h}$, and $\boldsymbol{B}, \boldsymbol{C} \in \mathbb{R}^{d_h \times 1}$ are static, leading to a fixed linear combination of inputs:

$$o_l^{(i)} = \sum_{j=1}^l \boldsymbol{C}^\top \overline{\boldsymbol{A}}^{l-j} \overline{\boldsymbol{B}} u_j^{(i)}, \tag{10}$$

where the coefficients $\boldsymbol{C}^\top \overline{\boldsymbol{A}}^{l-j} \overline{\boldsymbol{B}}$ are *task-agnostic*. This formulation inherently limits S4's ability to adapt to *task-specific* parameters $\boldsymbol{w}$ in ICL scenarios, as the model cannot adjust its inductive bias to match distinct $\boldsymbol{w}$ across different tasks. Therefore, the S4 model cannot truly learn in-context.

In contrast, Mamba's selection mechanism dynamically adjusts $\boldsymbol{B}_l$ and $\boldsymbol{C}_l$ (and optionally $\boldsymbol{A}_l$) based on the input tokens $(\boldsymbol{u}_1, \ldots, \boldsymbol{u}_N)$. This allows the model to implicitly adapt its hidden state to

align with the latent $\boldsymbol{w}$ of each task, effectively transforming the linear combination weights into context-dependent functions $f(\boldsymbol{x}) = \boldsymbol{x}^\top \boldsymbol{w}$. Such adaptability is essential for ICL, as it enables Mamba to reconstruct diverse $\boldsymbol{w}$ from input prompts without task-specific fine-tuning.

## 5 Proof Sketch

This section outlines the main technical ideas to prove Theorem 4.1. The complete proofs are given in the appendix.

**Linear Recurrence.** To start with, we show how the hidden states update when receiving token $\boldsymbol{e}_l = (\boldsymbol{x}_l^\top, y_l)^\top$. By (Eq. (5)) and Assumption 4.1(2), we have $\Delta_l = (\ln 2)/N$. Combining it with (Eq. (1)(2)) and get:

$$\boldsymbol{h}_l^{(d+1)} = \alpha \boldsymbol{h}_{l-1}^{(d+1)} + (1 - \alpha) y_l \boldsymbol{B}_l, \tag{11}$$

where $\alpha := \exp(-\Delta_l) = \exp((-\ln 2)/N)$, the second equality is by discretization rule (2), the third equality is by Assumption 4.1(2) and $\exp(-\Delta_l \boldsymbol{I}) = \exp(-\Delta_l)\boldsymbol{I}$.

**Prediction Output.** We next derive the expression of $\hat{y}_q$. By recurring (Eq.(11)), the hidden state after receiving the first $l$ context prompts $\boldsymbol{e}_{1:l}$ is given by $\boldsymbol{h}_l^{(d+1)} = (1 - \alpha) \sum_{i=0}^{l-1} \alpha^i y_{l-i} \boldsymbol{B}_{l-i}$. Receiving all the prompt tokens $\boldsymbol{e}_{1:N}$ and the query token $\boldsymbol{e}_q = (\boldsymbol{x}_q^\top, 0)^\top$, we have:

$$\boldsymbol{h}_{N+1}^{(d+1)} = \alpha \boldsymbol{h}_N^{(d+1)} + (1 - \alpha) \cdot 0 \cdot \boldsymbol{B}_N = (1 - \alpha) \sum_{i=0}^{N-1} \alpha^{i+1} y_{N-i} \boldsymbol{B}_{N-i}. \tag{12}$$

Finally, the prediction output is as follows

$$\hat{y}_q = \boldsymbol{C}_{N+1}^\top \boldsymbol{h}_{N+1}^{(d+1)} = (1 - \alpha)(\boldsymbol{W}_C \boldsymbol{e}_q + \boldsymbol{b}_C)^\top \sum_{i=0}^{N-1} \alpha^{i+1} y_{N-i} (\boldsymbol{W}_B \boldsymbol{e}_{N-i} + \boldsymbol{b}_B). \tag{13}$$

To handle $\boldsymbol{W}_C \boldsymbol{e}_q$ and $\boldsymbol{W}_B \boldsymbol{e}_{N-i}$, we further decompose $\boldsymbol{W}_B = [\boldsymbol{B}\, \boldsymbol{b}]$ and $\boldsymbol{W}_C = [\boldsymbol{C}\, \boldsymbol{c}]$, where $\boldsymbol{B}, \boldsymbol{C} \in \mathbb{R}^{d_h \times d}, \boldsymbol{b}, \boldsymbol{c} \in \mathbb{R}^{d_h \times 1}$. Then we write another form of (Eq. (13)):

$$\hat{y}_q = (1 - \alpha)(\boldsymbol{C}\boldsymbol{x}_q + \boldsymbol{b}_C)^\top \sum_{i=0}^{N-1} \alpha^{i+1} y_{N-i} (\boldsymbol{B}\boldsymbol{x}_{N-i} + y_{N-i}\boldsymbol{b} + \boldsymbol{b}_B). \tag{14}$$

The loss becomes:

$$\mathcal{L}(\boldsymbol{\theta}) = \frac{1}{2} \mathbb{E}\Big[\Big((1 - \alpha)(\boldsymbol{C}\boldsymbol{x}_q + \boldsymbol{b}_C)^\top \sum_{i=0}^{N-1} \alpha^{i+1} y_{N-i}(\boldsymbol{B}\boldsymbol{x}_{N-i} + y_{N-i}\boldsymbol{b} + \boldsymbol{b}_B) - \boldsymbol{w}^\top \boldsymbol{x}_q\Big)^2\Big]. \tag{15}$$

By computing the gradient of $\boldsymbol{C}, \boldsymbol{b}_C, \boldsymbol{B}, \boldsymbol{b}$ and $\boldsymbol{b}_B$ with respect to $\mathcal{L}(\boldsymbol{\theta}(t))$, we derive the following update rule according to Eq. (7).

**Lemma 5.1 (Update Rule)** *Let $\eta$ be the learning rate and we use gradient descent to update the weights $\boldsymbol{W}_B, \boldsymbol{W}_C, \boldsymbol{b}_B, \boldsymbol{b}_C$, for $t \geq 0$ we have*

$$\boldsymbol{B}(t + 1) = \boldsymbol{B}(t) + \eta \beta_3 \boldsymbol{C}(t) - \eta \beta_1 \boldsymbol{C}(t)\boldsymbol{C}(t)^\top \boldsymbol{B}(t),$$

$$\boldsymbol{C}(t + 1) = \boldsymbol{C}(t) + \eta \beta_3 \boldsymbol{B}(t) - \eta \beta_1 \boldsymbol{B}(t)\boldsymbol{B}(t)^\top \boldsymbol{C}(t) - \eta \beta_2 \boldsymbol{b}(t)\boldsymbol{b}(t)^\top \boldsymbol{C}(t),$$

$$\boldsymbol{b}(t + 1) = \boldsymbol{b}(t) - \eta \beta_2 \boldsymbol{C}(t)\boldsymbol{C}(t)^\top \boldsymbol{b}(t), \quad \boldsymbol{b}_B(t) = \boldsymbol{b}_C(t) = \boldsymbol{0},$$

*where $\beta_1 = \mathbb{E}\Big[\sum_{i=0}^{N-1}\sum_{j=0}^{N-1}(1-\alpha)^2\alpha^{i+j+2}y_{N-i}y_{N-j}\boldsymbol{x}_{N-i}\boldsymbol{x}_{N-j}^\top\Big]$, $\beta_2 = \mathbb{E}\Big[\sum_{i=0}^{N-1}\sum_{j=0}^{N-1}(1-\alpha)^2\alpha^{i+j+2}y_{N-i}^2 y_{N-j}^2\Big]$, $\beta_3 = \mathbb{E}\Big[\sum_{i=0}^{N-1}(1-\alpha)\alpha^{i+1}y_{N-i}\boldsymbol{x}_{N-i}\boldsymbol{w}^\top\Big]$.*

**Technical Challenges.** Unlike many prior Transformer-based ICL analyses that simplify dynamics via merged weights or special initializations, our Gaussian-initialized $\boldsymbol{W}_B$, $\boldsymbol{W}_C$ and discrete-time gradient descent introduces more complexity (cf. assumption 4.1). To solve the optimization problem described in Lemma 5.1, we have the following three questions to answer: (1) Convergence Target: Where do the parameters converge? (2) Convergence Proof: How to rigorously establish convergence? (3) Saddle Point Avoidance: How to avoid saddle points? To answer these three questions, we propose two key techniques: *Vector-coupled Dynamic*, *Negative Feedback Convergence*, and apply them with a *Fine-grained Induction*. We next describe them in detail.

## 5.1 Vector-coupled Dynamics

We can verify by Lemma 5.1 that $\boldsymbol{C}^\top \boldsymbol{B} = \mathrm{Diag}(a_1, \dots, a_d)$ with $a_i \in \{0, \frac{\beta_3}{\beta_1}\}$, $\boldsymbol{C}^\top \boldsymbol{b} = \boldsymbol{0}$ are the fixed points for the parameters $\boldsymbol{W}_B$, $\boldsymbol{W}_C$.

Combining the loss function Eq. 15 and $\boldsymbol{b}_B(t) = \boldsymbol{b}_C(t) = \boldsymbol{0}$ in Lemma 5.1, the loss function can be rewritten as

$$\mathcal{L}(\boldsymbol{\theta}) = \frac{1}{2}\mathbb{E}\left[\left((1-\alpha)\sum_{i=0}^{N-1}\alpha^{i+1}(\boldsymbol{x}_q^\top \boldsymbol{C}^\top \boldsymbol{B} y_{N-i}\boldsymbol{x}_{N-i} + y_{N-i}^2\boldsymbol{x}_q^\top \boldsymbol{C}^\top \boldsymbol{b}) - \boldsymbol{w}^\top \boldsymbol{x}_q\right)^2\right].$$

To minimize loss, the term $(1-\alpha)\sum_{i=0}^{N-1}\alpha^{i+1}(\boldsymbol{x}_q^\top \boldsymbol{C}^\top \boldsymbol{B} y_{N-i}\boldsymbol{x}_{N-i} + y_{N-i}^2\boldsymbol{x}_q^\top \boldsymbol{C}^\top \boldsymbol{b})$ should approximate $\boldsymbol{w}^\top \boldsymbol{x}_q$. Given $\mathbb{E}[y_{N-i}\boldsymbol{x}_{N-i}] = \boldsymbol{w}$ and $\mathbb{E}[y_{N-i}^2] > 0$, we derive that $\boldsymbol{C}^\top \boldsymbol{B}$ should converge to $\frac{\beta_3}{\beta_1}\boldsymbol{I}$, while $\boldsymbol{C}^\top \boldsymbol{b}$ converges to $\boldsymbol{0}$ to minimize the loss. However, as mentioned above, $\boldsymbol{C}^\top \boldsymbol{B} = \mathrm{Diag}(a_1, \dots, a_d)$ with partial $a_i = 0$ can also enable convergence, which is an undesirable scenario.

To analyze the convergence behavior of $\boldsymbol{C}^\top \boldsymbol{B}$ and $\boldsymbol{C}^\top \boldsymbol{b}$, we introduce the *Vector-coupled Dynamics* technique, which studies the inner product dynamics between decomposed column vectors of $\boldsymbol{B}$ and $\boldsymbol{C}$. Specifically, we decompose $\boldsymbol{B}$ and $\boldsymbol{C}$ into $\boldsymbol{B} = [\boldsymbol{b}_1 \dots \boldsymbol{b}_d]$, $\boldsymbol{C} = [\boldsymbol{c}_1 \dots \boldsymbol{c}_d]$. Then we have another form of Lemma 5.1 for $\boldsymbol{B}$, $\boldsymbol{C}$ and $\boldsymbol{b}$ as the following lemma.

**Lemma 5.2 (Vectors Update Rule)** *Let $\eta$ be the learning rate and we use gradient descent to update the weights $\boldsymbol{W}_B, \boldsymbol{W}_C, \boldsymbol{b}_B, \boldsymbol{b}_C$, for $i \in [d]$, $t \geq 0$ we have*

$$\boldsymbol{b}_i(t+1) = \boldsymbol{b}_i(t) + \eta\left(\left(\beta_3 - \beta_1\boldsymbol{c}_i^\top(t)\boldsymbol{b}_i(t)\right)\boldsymbol{c}_i(t) - \beta_1\sum_{k\neq i}^{d}\boldsymbol{c}_k^\top(t)\boldsymbol{b}_i(t)\cdot\boldsymbol{c}_k(t)\right),$$

$$\boldsymbol{c}_i(t+1) = \boldsymbol{c}_i(t) + \eta\left(\left(\beta_3 - \beta_1\boldsymbol{c}_i^\top(t)\boldsymbol{b}_i(t)\right)\boldsymbol{b}_i(t) - \beta_1\sum_{k\neq i}^{d}\boldsymbol{c}_i^\top(t)\boldsymbol{b}_k(t)\cdot\boldsymbol{b}_k(t) - \beta_2\boldsymbol{c}_i^\top(t)\boldsymbol{b}(t)\cdot\boldsymbol{b}(t)\right),$$

$$\boldsymbol{b}(t+1) = \boldsymbol{b}(t) - \eta\left(\beta_2\sum_{k=1}^{d}\boldsymbol{c}_k^\top(t)\boldsymbol{b}(t)\cdot\boldsymbol{c}_k(t)\right).$$

With Lemma 5.2, we can further analyze the dynamics of the inner products $\boldsymbol{c}_i^\top(t)\boldsymbol{b}_i(t)$, $\boldsymbol{c}_i^\top(t)\boldsymbol{b}_j(t)$ and $\boldsymbol{c}_i^\top(t)\boldsymbol{b}(t)$, precisely characterizing the behavior of $\boldsymbol{C}^\top \boldsymbol{B}$ and $\boldsymbol{C}^\top \boldsymbol{b}$. This technique helps answer the question "*Where do the parameters converge?*"

## 5.2 Negative Feedback Convergence

As we discuss in Section 5.1, to minimize loss, the following conditions must be satisfied for all $i, j \in [d]$ with $i \neq j$: $\boldsymbol{c}_i^\top(t)\boldsymbol{b}_i(t) \to \frac{\beta_3}{\beta_1}$, $\boldsymbol{c}_i^\top(t)\boldsymbol{b}_j(t) \to 0$, $\boldsymbol{c}_i^\top(t)\boldsymbol{b}(t) \to 0$. To establish the convergence, we introduce the *Negative Feedback Convergence* technique. This technique leverages the negative feedback terms in the dynamical equations of $\boldsymbol{c}_i^\top(t)\boldsymbol{b}_i(t)$, $\boldsymbol{c}_i^\top(t)\boldsymbol{b}_j(t)$, and $\boldsymbol{c}_i^\top(t)\boldsymbol{b}(t)$ to derive an exponential convergence rate. Taking $\boldsymbol{c}_i^\top(t)\boldsymbol{b}_i(t)$ as an example, we derive the following

update rule by Lemma 5.2.

$$\left(\beta_3 - \beta_1 \boldsymbol{c}_i^\top (t+1) \boldsymbol{b}_i(t+1)\right) = \underline{\beta_3 - \beta_1 \boldsymbol{c}_i^\top (t) \boldsymbol{b}_i(t)}$$

$$\underbrace{-\eta\beta_1 \left(\beta_3 - \beta_1 \boldsymbol{c}_i^\top(t)\boldsymbol{b}_i(t)\right) \boldsymbol{b}_i^\top(t)\boldsymbol{b}_i(t) - \eta\beta_1 \left(\beta_3 - \beta_1 \boldsymbol{c}_i^\top(t)\boldsymbol{b}_i(t)\right) \boldsymbol{c}_i^\top(t)\boldsymbol{c}_i(t)}_{negative\ feedback\ term}$$

$$+ \eta\beta_1^2 \sum_{k\neq i}^d \boldsymbol{c}_i^\top(t)\boldsymbol{b}_k(t) \cdot \boldsymbol{b}_k^\top(t)\boldsymbol{b}_i(t) + \eta\beta_1^2 \sum_{k\neq i}^d \boldsymbol{c}_k^\top(t)\boldsymbol{b}_i(t) \cdot \boldsymbol{c}_i^\top(t)\boldsymbol{c}_k(t)$$

$$+ \eta\beta_1\beta_2 \boldsymbol{c}_i^\top(t)\boldsymbol{b}(t) \cdot \boldsymbol{b}_i^\top(t)\boldsymbol{b}(t) - \beta_1\big(\boldsymbol{c}(t+1) - \boldsymbol{c}(t)\big)^\top \big(\boldsymbol{b}(t+1) - \boldsymbol{b}(t)\big). \quad (16)$$

The term $\left(\beta_3 - \beta_1 \boldsymbol{c}_i^\top(t+1)\boldsymbol{b}_i(t+1)\right)$ decomposes into its previous state $\left(\beta_3 - \beta_1 \boldsymbol{c}_i^\top(t)\boldsymbol{b}_i(t)\right)$ (marked with underline) plus the remaining terms (increment terms). The increment terms includes a *negative feedback term*, which induces a tendency to drive $\left(\beta_3 - \beta_1 \boldsymbol{c}_i^\top(t)\boldsymbol{b}_i(t)\right)$ to 0 ($\boldsymbol{c}_i^\top(t)\boldsymbol{b}_i(t) \to \frac{\beta_3}{\beta_1}$).

Intuitively, $\boldsymbol{b}_i^\top(t)\boldsymbol{b}_i(t)$ and $\boldsymbol{c}_i^\top(t)\boldsymbol{c}_i(t)$ are much larger than $\boldsymbol{b}_k^\top(t)\boldsymbol{b}_i(t)$, $\boldsymbol{c}_i^\top(t)\boldsymbol{c}_k(t)$ and $\boldsymbol{b}_i^\top(t)\boldsymbol{b}(t)$ at Gaussian initialization with high probability. Also, as $\boldsymbol{c}_i^\top(t)\boldsymbol{b}_j(t), \boldsymbol{b}_i^\top(t)\boldsymbol{b}(t) \to 0$ with $i \neq j$ and $\eta$ is small enough, the effect of *negative feedback term* is the dominant term in the increment terms. Therefore, denoting $y(t) = \beta_3 - \beta_1 \boldsymbol{c}_i^\top(t)\boldsymbol{b}_i(t)$ and $\xi(t) = y(t+1) - y(t) - negative\ feedback\ term$ we can model the update rule of (Eq. 16) as follows:

$$y(t+1) = \big(1 - \eta\beta_1(\boldsymbol{b}_i^\top(t)\boldsymbol{b}_i(t) + \boldsymbol{c}_i^\top(t)\boldsymbol{c}_i(t))\big)y(t) + \xi(t).$$

Recur this formula from 0 to $t$, we have:

$$y(t+1) = \prod_{s=0}^t \Big(1 - \eta\beta_1\big(\boldsymbol{b}_i^\top(s)\boldsymbol{b}_i(s) + \boldsymbol{c}_i^\top(s)\boldsymbol{c}_i(s)\big)\Big)y(0)$$

$$+ \sum_{s=0}^t \prod_{s'=s+1}^t \Big(1 - \eta\beta_1\big(\boldsymbol{b}_i^\top(s')\boldsymbol{b}_i(s') + \boldsymbol{c}_i^\top(s')\boldsymbol{c}_i(s')\big)\Big)\xi(s'). \quad (17)$$

Denoting $\gamma = \min\{\boldsymbol{b}_i^\top(s)\boldsymbol{b}_i(s), \boldsymbol{c}_i^\top(s)\boldsymbol{c}_i(s)\}$ for $s \in [0,t]$, the first term on the RHS of (Eq. (17)) can be upper bounded by $(1 - 2\eta\beta_1\gamma)^{t+1}y(0)$. if $\xi(s')$ has an exponentially decaying upper bound (it can be proved when $\boldsymbol{c}_i^\top(t)\boldsymbol{b}_j(t) \to 0, \boldsymbol{c}_i^\top(t)\boldsymbol{b}(t) \to 0$ with an exponential rate), the second term on the RHS of (Eq. (17)) has an exponentially decaying upper bound. Therefore, we can establish an exponential convergence rate for $\boldsymbol{c}_i^\top(t)\boldsymbol{b}_i(t) \to \frac{\beta_3}{\beta_1}$. The similar method can be used on $\boldsymbol{c}_i^\top(t)\boldsymbol{b}_j(t) \to 0, \boldsymbol{c}_i^\top(t)\boldsymbol{b}(t) \to 0$. This technique helps answer the question "*How to rigorously establish convergence?*"

## 5.3 Fine-grained Induction

The exponential convergence of $\boldsymbol{c}_i^\top(t)\boldsymbol{b}_i(t) \to \frac{\beta_3}{\beta_1}$ under the *Negative Feedback Convergence* framework requires the following two conditions for all $i, j \in [d]$ with $i \neq j$:

(1) $\boldsymbol{b}_i^\top(t)\boldsymbol{b}_i(t)$ and $\boldsymbol{c}_i^\top(t)\boldsymbol{c}_i(t)$ dominate $\boldsymbol{b}_i^\top(t)\boldsymbol{b}_j(t)$, $\boldsymbol{c}_i^\top(t)\boldsymbol{c}_j(t)$ and $\boldsymbol{b}_i^\top(t)\boldsymbol{b}(t)$ in magnitude.

(2) $\boldsymbol{c}_i^\top(t)\boldsymbol{b}_j(t) \to 0, \boldsymbol{c}_i^\top(t)\boldsymbol{b}(t) \to 0$ at an exponentially decaying rate.

On the one hand, condition (1) at initialization ($t = 0$) can be established via concentration inequalities, and critically, the preservation of Condition (1) for $t > 0$ relies on the rapid decay of $\boldsymbol{c}_i^\top(t)\boldsymbol{b}_i(t), \boldsymbol{c}_i^\top(t)\boldsymbol{b}_j(t)$, and $\boldsymbol{c}_i^\top(t)\boldsymbol{b}(t)$ (condition (2)). On the other hand, under the framework of *Negative Feedback Convergence*, $\boldsymbol{c}_i^\top(t)\boldsymbol{b}_j(t) \to 0$ in Condition (2) also relies on Condition (1) and the rapid decay of $\boldsymbol{c}_i^\top(t)\boldsymbol{b}_i(t) \to \frac{\beta_3}{\beta_1}, \boldsymbol{c}_i^\top(t)\boldsymbol{b}(t) \to 0$. This implies mutual dependencies among the bounds of these *Vector-coupled* inner products.

To handle these dependencies and establish stable bounds, we introduce the technique *Fine-grained Induction*: Divide the inner products into three groups: (1) Squared norms: $\boldsymbol{b}_i^\top(t)\boldsymbol{b}_i(t), \boldsymbol{c}_i^\top(t)\boldsymbol{c}_i(t), \boldsymbol{b}^\top(t)\boldsymbol{b}(t)$. (2) Target terms: $\boldsymbol{c}_i^\top(t)\boldsymbol{b}_i(t), \boldsymbol{c}_i^\top(t)\boldsymbol{b}_j(t), \boldsymbol{c}_i^\top(t)\boldsymbol{b}(t)$. (3) Cross-interactions: $\boldsymbol{b}_i^\top(t)\boldsymbol{b}_j(t), \boldsymbol{c}_i^\top(t)\boldsymbol{c}_j(t), \boldsymbol{b}_i^\top(t)\boldsymbol{b}(t)$. And then carefully give bounds for them with an induction.

Specifically, denoting $\delta(t) = \max_{s\in[0,t]}\{2\sqrt{d_h\log(4d(2d+1)/\delta)}, |\boldsymbol{b}_i^\top(s)\boldsymbol{b}_j(s)|, |\boldsymbol{c}_i^\top(s)\boldsymbol{c}_j(s)|,$
$|\boldsymbol{b}_i^\top(s)\boldsymbol{b}(s)|\}$ and $\gamma = \frac{1}{2}d_h \leq \min_{t\geq 0}\{\boldsymbol{b}_i^\top(t)\boldsymbol{b}_i(t), \boldsymbol{c}_i^\top(t)\boldsymbol{c}_i(t), \boldsymbol{b}^\top(t)\boldsymbol{b}(t)\}$, we establish the following three properties $\mathcal{A}(t)$, $\mathcal{B}(t)$, and $\mathcal{C}(t)$ simultaneously for $t \geq 0$:

$$\mathcal{A}(t): \qquad\qquad d_h/2 \leq \boldsymbol{b}_i^\top(t)\boldsymbol{b}_i(t), \boldsymbol{c}_i^\top(t)\boldsymbol{c}_i(t), \boldsymbol{b}^\top(t)\boldsymbol{b}(t) \leq 2d_h.$$

$$\mathcal{B}(t): \qquad |\beta_3 - \beta_1\boldsymbol{c}_i^\top(t)\boldsymbol{b}_i(t)| \leq \delta(t)\exp(-\eta\beta_1\gamma t), \quad |\boldsymbol{c}_i^\top(t)\boldsymbol{b}_j(t)| \leq 2\delta(t)\exp(-\eta\beta_1\gamma t),$$

$$|\boldsymbol{c}_i^\top(t)\boldsymbol{b}(t)| \leq 2\delta(t)\exp(-\eta\beta_2\gamma t) + \frac{\delta(t)}{\beta_2}\exp(-\eta\beta_1\gamma t).$$

$$\mathcal{C}(t): \qquad |\boldsymbol{b}_i^\top(t)\boldsymbol{b}_j(t)|, |\boldsymbol{c}_i^\top(t)\boldsymbol{c}_j(t)|, |\boldsymbol{b}_i^\top(t)\boldsymbol{b}(t)| \leq \delta(t) \leq 3\sqrt{d_h\log(4d(2d+1)/\delta)}.$$

The initial conditions $\mathcal{A}(0)$, $\mathcal{B}(0)$, and $\mathcal{C}(0)$ are established with high probability by concentration inequalities. We also provide the following claims to establish $\mathcal{A}(t)$, $\mathcal{B}(t)$, and $\mathcal{C}(t)$ for $t \geq 0$:

**Claim 5.1** $\mathcal{A}(0),\ldots,\mathcal{A}(T), \mathcal{B}(0),\ldots,\mathcal{B}(T), \mathcal{C}(0),\ldots,\mathcal{C}(T) \implies \mathcal{A}(T+1)$.

**Claim 5.2** $\mathcal{A}(0),\ldots,\mathcal{A}(T), \mathcal{B}(0),\ldots,\mathcal{B}(T), \mathcal{C}(0),\ldots,\mathcal{C}(T) \implies \mathcal{B}(T+1)$.

**Claim 5.3** $\mathcal{A}(0),\ldots,\mathcal{A}(T), \mathcal{B}(0),\ldots,\mathcal{B}(T), \mathcal{C}(0),\ldots,\mathcal{C}(T) \implies \mathcal{C}(T+1)$.

This induction answers the question "*How to avoid saddle points?*" because $\mathcal{B}(t)$ guarantees that $\boldsymbol{C}^\top\boldsymbol{B} \to \frac{\beta_3}{\beta_1}\boldsymbol{I}$ and $\boldsymbol{C}^\top\boldsymbol{b} \to \boldsymbol{0}$, preventing stagnation of partial diagonal entries of $\boldsymbol{C}^\top\boldsymbol{B}$ at zero. Theorem 4.1 can be proved by substituting $\boldsymbol{C}^\top\boldsymbol{B} = \frac{\beta_3}{\beta_1}\boldsymbol{I}, \boldsymbol{C}^\top\boldsymbol{b} = \boldsymbol{0}, \boldsymbol{b}_B = \boldsymbol{b}_C = \boldsymbol{0}$ into (Eq. (11), (14), (15))

# 6 Experimental Results

We present simulation results on synthetic data to verify our theoretical results. More experimental results can be found in Appendix E.

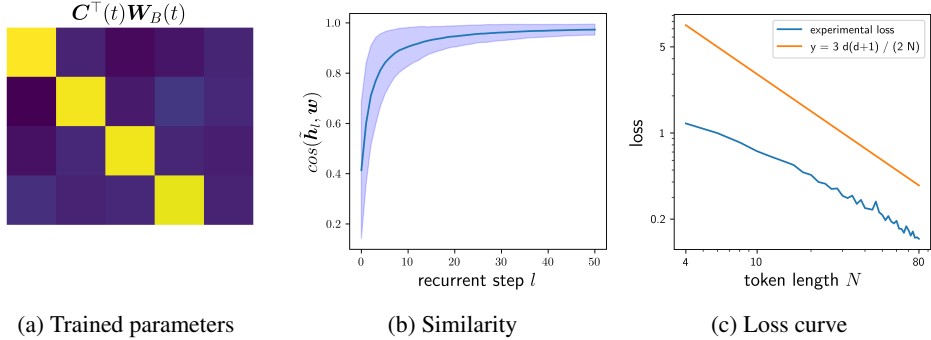

|  | |  |
|:---:|:---:|:---:|
| (a) Trained parameters | (b) Similarity | (c) Loss curve |

Figure 1: (a) Post-training visualization of matrix product $\boldsymbol{C}^\top\boldsymbol{W}_B$; (b) Cosine similarity evolution between $\boldsymbol{w}$ and $\tilde{\boldsymbol{h}}_l = (\boldsymbol{W}_C^\top)_{[1:d,:]}\boldsymbol{h}_l^{(d+1)}$ across recurrent steps $l$ (after processing prompts $\boldsymbol{e}_{1:l}$); (c) Test loss versus token sequence length $N$. Blue curve: experimental results; orange curve: theoretical upper bound.

**Experiments Setting** We follow Section 3 to generate the dateset and initialize the model. Specifically, we set dimension $d = 4$, $d_h = 80$, prompt token length $N = 50$, and train the Mamba model on 3000 sequences by gradient descent. After training, we save the model and test it on 1000 new generated sequences, tracking the cosine similarity between $\tilde{\boldsymbol{h}}_l(:= (\boldsymbol{W}_C^\top)_{[1:d,:]}\boldsymbol{h}_l^{(d+1)})$ and $\boldsymbol{w}$. Moreover, we vary the length of the prompt token $N$ from 4 to 80 and compare the test loss with the theoretical upper bound. For each $N$, we conduct 10 independent experiments and report the averaged results. All experiments are performed on an NVIDIA A800 GPU.

**Experiment Result**    Recalling that we denote $\boldsymbol{W}_B = [\boldsymbol{B}\,\boldsymbol{b}]$, Figure 1a reveals the convergence of $\boldsymbol{C}^\top \boldsymbol{B}$ to a diagonal matrix and $\boldsymbol{C}^\top \boldsymbol{b}$ to $\mathbf{0}$, confirming the theoretical induction presented in Section 5, also consistent with (Thm 4.1 (b)). Figure 1b shows that the projected hidden state $\tilde{\boldsymbol{h}}_l$ gradually aligns with $\boldsymbol{w}$ as more prompt tokens are processed, consistent with (Thm 4.1 (a)). Figure 1c demonstrates that the experimental loss has an upper bound $\frac{3d(d+1)}{2N}$ that decays linearly with N, aligning with (Thm 4.1 (c)).

# 7    Conclusion

This paper study Mamba's in-context learning mechanism, and rigorously establish its convergence and loss bound. By analysing the *Vector-coupled Dynamics*, we provide an exponential convergence rate with *Negative Feedback Convergence* technique in a *Fine-grained Induction*, and finally establish a $O(1/N)$ loss bound. The loss bound is comparable to that of Transformer. Our theoretical results reveal the different mechanism between Transformer and Mamba on ICL, where Mamba emulates a variant of *online gradient descent* to perform in-context, while Transformers approximate a single step of gradient descent. Furthermore, our comparison with the S4 model demonstrates that the selection components are essential for Mamba to perform ICL.

**Limitations and Social Impact**    Our analysis focuses on one-layer Mamba model, thus the behavior of Mamba with multi-layer or augmented with other components such as MLP is still unclear. We believe that our work will provide insight for those cases and can be used to study more data models such as nonlinear features. This paper is mainly a theoretical investigation, and we do not see an immediate social impact.

# Acknowledgements

We thank the anonymous reviewers for their insightful comments to improve the paper. Miao Zhang was partially sponsored by the National Natural Science Foundation of China under Grant 62306084 and U23B2051, Shenzhen College Stability Support Plan under Grant GXWD20231128102243003, and Shenzhen Science and Technology Program under Grant ZDSYS20230626091203008 and KJZD20230923115113026.

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

# Appendix

## Contents

Table 1: Key notations

| Symbols | Definitions |
|---|---|
| $\boldsymbol{x}_i, \boldsymbol{x}_q, \boldsymbol{w}, y_i, y_q$ | $\boldsymbol{x}_i, \boldsymbol{x}_q, \boldsymbol{w}$ are i.i.d. sampled from Gaussian distribution $\mathcal{N}(0, \boldsymbol{I}_d)$. $y_i = \boldsymbol{w}^\top \boldsymbol{x}_i, \quad y_q = \boldsymbol{w}^\top \boldsymbol{x}_q.$ |
| $\bar{\boldsymbol{b}}_i(t), \bar{\boldsymbol{c}}_i(t), \bar{\boldsymbol{b}}(t)$ | $\bar{\boldsymbol{b}}_i(t) = \frac{1}{\eta}\big(\boldsymbol{b}_i(t+1) - \boldsymbol{b}_i(t)\big),$ $\bar{\boldsymbol{c}}_i(t) = \frac{1}{\eta}\big(\boldsymbol{c}_i(t+1) - \boldsymbol{c}_i(t)\big),$ $\bar{\boldsymbol{b}}(t) = \frac{1}{\eta}\big(\boldsymbol{b}(t+1) - \boldsymbol{b}(t)\big).$ |
| $\boldsymbol{B}, \boldsymbol{C}, \boldsymbol{b}, \boldsymbol{c}, \boldsymbol{b}_i, \boldsymbol{c}_i$ | Decompose the matrices $\boldsymbol{W}_B, \boldsymbol{W}_C$ into colums of vectors: $\boldsymbol{W}_B = [\boldsymbol{B}\,\boldsymbol{b}] = [\boldsymbol{b}_1, \ldots, \boldsymbol{b}_d\,\boldsymbol{b}], \boldsymbol{W}_C = [\boldsymbol{C}\,\boldsymbol{c}] = [\boldsymbol{c}_1, \ldots, \boldsymbol{c}_d\,\boldsymbol{c}]$ where $\boldsymbol{W}_B, \boldsymbol{W}_C \in \mathbb{R}^{d_h \times (d+1)}$, $\boldsymbol{B}, \boldsymbol{C} \in \mathbb{R}^{d_h \times d}$, $\boldsymbol{b}, \boldsymbol{c}, \boldsymbol{b}_i, \boldsymbol{c}_i \in \mathbb{R}^{d_h \times 1}$ |
| $\boldsymbol{b}_i(t)^\top \boldsymbol{b}_j(t), \boldsymbol{c}_i(t)^\top \boldsymbol{c}_j(t), \boldsymbol{b}(t)^\top \boldsymbol{b}(t)$ $\boldsymbol{c}_i(t)^\top \boldsymbol{b}_j(t), \boldsymbol{b}_i(t)^\top \boldsymbol{b}(t), \boldsymbol{c}_i(t)^\top \boldsymbol{b}(t)$ | inner product of the vectors $\boldsymbol{b}, \boldsymbol{b}_i, \boldsymbol{c}_i$ with $i, j \in [1, d]$. e.g. $\boldsymbol{b}_i(t)^\top \boldsymbol{b}_j(t)$ is the inner product of $\boldsymbol{b}_i(t)$ and $\boldsymbol{b}_i(t)$. |
| $\alpha$ | A factor, $\alpha := \exp(-\Delta_l) = \exp((-\ln 2)/N)$. |
| $\beta_1, \beta_2, \beta_3$ | The factors appearing in the gradient equation. Specifically, $\beta_1 = \left(\alpha^2\big(1-\alpha^N\big)^2 + \frac{(d+1)\alpha^2(1-\alpha)\big(1-\alpha^{2N}\big)}{(1+\alpha)}\right),$ $\beta_2 = \left(d^2\alpha^2\big(1-\alpha^N\big)^2 + \frac{(2d^2+6d)\alpha^2(1-\alpha)\big(1-\alpha^{2N}\big)}{(1+\alpha)}\right),$ $\beta_3 = \alpha\big(1-\alpha^N\big)$ |
| $\gamma$ | The lower bound of squared norms $\boldsymbol{b}_i^\top(t)\boldsymbol{b}_i(t), \boldsymbol{c}_i^\top(t)\boldsymbol{c}_i(t),$ and $\boldsymbol{b}^\top(t)\boldsymbol{b}(t)$. Specifically, $\gamma = \frac{1}{2}d_h$. |
| $\delta(T)$ | The upper bound of cross-interactions: $\boldsymbol{b}_i^\top(t)\boldsymbol{b}_j(t), \boldsymbol{c}_i^\top(t)\boldsymbol{c}_j(t),$ and $\boldsymbol{b}_i^\top(t)\boldsymbol{b}(t)$. Specifically, $\delta(t) = \max_{s \in [0,t]}\{2\sqrt{d_h \log(4d(2d+1)/\delta)},$ $|\boldsymbol{b}_i^\top(s)\boldsymbol{b}_j(s)|, |\boldsymbol{c}_i^\top(s)\boldsymbol{c}_j(s)|, |\boldsymbol{b}_i^\top(s)\boldsymbol{b}(s)|\}.$ |

## A  Basic Calculations

This Section provide the data statistics related to gaussian distribution, and compute the expressions for the output, loss, gradient, training dynamics (particularly *Vector-coupled Dynamics*) of the Mamba model. Section B presents the *Fine-grained Induction* with *Negative Feedback Convergence* technique, and finally establish the results for Theorem 4.1. Section C details the complete proofs for Section A and Section B. In Section D, we discuss about orthogonal initialization and compare our framework with other techniques. In Section E, we give more experimental results.

### A.1  Data Statistics

**Lemma A.1 (Concentration Inequalities)** *Let $\boldsymbol{b}_i(0)$ be the i-th colum of $\boldsymbol{B}(0)$, $\boldsymbol{c}_i(0)$ be the i-th colum of $\boldsymbol{C}(0)$, and suppose that $\delta > 0$ and $d_h = \Omega(\log(4(2d+1)/\delta))$, with probability at least $1 - \delta$, we have:*

$$\frac{3d_h}{4} \le \boldsymbol{b}_i(0)^\top \boldsymbol{b}_i(0), \boldsymbol{c}_i(0)^\top \boldsymbol{c}_i(0), \boldsymbol{b}(0)^\top \boldsymbol{b}(0) \le \frac{5d_h}{4},$$

$$\left|\boldsymbol{c}_i(0)^\top \boldsymbol{b}_i(0)\right|, \left|\boldsymbol{c}_i(0)^\top \boldsymbol{b}_j(0)\right|, \left|\boldsymbol{c}_i(0)^\top \boldsymbol{b}(0)\right| \le 2\sqrt{d_h \log(4d(2d+1)/\delta)},$$

$$\left|\boldsymbol{b}_i(0)^\top \boldsymbol{b}_j(0)\right|, \left|\boldsymbol{c}_i(0)^\top \boldsymbol{c}_j(0)\right|, \left|\boldsymbol{b}_i(0)^\top \boldsymbol{b}(0)\right| \le 2\sqrt{d_h \log(4d(2d+1)/\delta)}$$

*for $i, j \in [d], i \ne j$.*

*Proof of Lemma A.1.* By Bernstein's inequality, with probability at least $1 - \delta/2(2d+1)$ we have

$$\left|\boldsymbol{b}_i(0)^\top \boldsymbol{b}_i(0) - d_h\right| = O\left(\sqrt{d_h \log(4(2d+1)/\delta)}\right).$$

Therefore, as long as $d_h = \Omega(\log(4(2d+1)/\delta))$, we have $3d_h/4 \leq \boldsymbol{b}_i(0)^\top \boldsymbol{b}_i(0) \leq 5d_h/4$. Similarly, we have

$$\frac{3d_h}{4} \leq \boldsymbol{c}_i(0)^\top \boldsymbol{c}_i(0), \boldsymbol{b}(0)^\top \boldsymbol{b}(0) \leq \frac{5d_h}{4}.$$

For $i, j \in [d], i \neq j$, By Bernstein's inequality, with probability at least $1 - \delta/2d(2d+1)$, we have

$$\left| \boldsymbol{c}_i(0)^\top \boldsymbol{b}_i(0) \right|, \left| \boldsymbol{c}_i(0)^\top \boldsymbol{b}_j(0) \right|, \left| \boldsymbol{c}_i(0)^\top \boldsymbol{b}(0) \right| \leq 2\sqrt{d_h \log(4d(2d+1)/\delta)},$$

$$\left| \boldsymbol{b}_i(0)^\top \boldsymbol{b}_j(0) \right|, \left| \boldsymbol{c}_i(0)^\top \boldsymbol{c}_j(0) \right|, \left| \boldsymbol{b}_i(0)^\top \boldsymbol{b}(0) \right| \leq 2\sqrt{d_h \log(4d(2d+1)/\delta)}.$$

We can apply a union bound to complete the proof.

**Lemma A.2** *If vectors $\boldsymbol{x}$ and $\boldsymbol{w}$ are iid generated from $\mathcal{N}(0, \boldsymbol{I}_d)$, $y = \boldsymbol{x}^\top \boldsymbol{w}$ we have the following expectations:*

$$\mathbb{E}\left[ \boldsymbol{x}\boldsymbol{x}^\top \boldsymbol{w}\boldsymbol{w}^\top \boldsymbol{x}\boldsymbol{x}^\top \right] = (d+2)\boldsymbol{I},$$

$$\mathbb{E}\left[ y^2 \right] = d,$$

$$\mathbb{E}\left[ y^4 \right] = 3d(d+2).$$

The proof of lemma A.2 is in Section C.1.

**Lemma A.3** *If vectors $\boldsymbol{x}_i$ and $\boldsymbol{w}$ are iid generated from $\mathcal{N}(0, \boldsymbol{I}_d)$, $y = \boldsymbol{x}_i^\top \boldsymbol{w}$ we have the following expectations:*

$$\mathbb{E}\left[ \sum_{i=0}^{N-1} \sum_{j=0}^{N-1} \alpha^{i+j+2} y_{N-i} y_{N-j} \boldsymbol{x}_{N-i} \boldsymbol{x}_{N-j}^\top \right] = \left( \frac{\alpha^2 (1-\alpha^N)^2}{(1-\alpha)^2} + \frac{(d+1)\alpha^2(1-\alpha^{2N})}{(1-\alpha)(1+\alpha)} \right) \cdot \boldsymbol{I},$$

$$\mathbb{E}\left[ \sum_{i=0}^{N-1} \sum_{j=0}^{N-1} \alpha^{i+j+2} y_{N-i} y_{N-j}^2 \boldsymbol{x}_{N-i} \right] = \boldsymbol{0},$$

$$\mathbb{E}\left[ \sum_{i=0}^{N-1} \sum_{j=0}^{N-1} \alpha^{i+j+2} y_{N-i}^2 y_{N-j}^2 \right] = \frac{d^2 \alpha^2 \left(1-\alpha^N\right)^2}{(1-\alpha)^2} + \frac{(2d^2+6d)\alpha^2\left(1-\alpha^{2N}\right)}{(1-\alpha)(1+\alpha)},$$

$$\mathbb{E}\left[ \sum_{i=0}^{N-1} \alpha^{i+1} y_{N-i} \boldsymbol{x}_{N-i} \boldsymbol{w}^\top \right] = \alpha \left( \frac{1-\alpha^N}{1-\alpha} \right) \cdot \boldsymbol{I},$$

$$\mathbb{E}\left[ \sum_{i=0}^{N-1} \alpha^{i+1} y_{N-i}^2 \boldsymbol{w} \right] = \boldsymbol{0},$$

$$\mathbb{E}\left[ \sum_{i=0}^{N-1} \sum_{j=0}^{N-1} \alpha^{i+j+2} \boldsymbol{x}_{N-i} \underbrace{\boldsymbol{x}_{N-i}^\top \boldsymbol{w}}_{y_{N-i}} \underbrace{\boldsymbol{x}_{N-j}^\top \boldsymbol{w}}_{y_{N-j}} \right] = \boldsymbol{0},$$

$$\mathbb{E}\left[ \sum_{i=0}^{N-1} \sum_{j=0}^{N-1} \alpha^{i+j+2} y_{N-i}^2 y_{N-j} \right] = \boldsymbol{0},$$

$$\mathbb{E}\left[ \sum_{i=0}^{N-1} \sum_{j=0}^{N-1} \alpha^{i+j+2} y_{N-i} y_{N-j} \right] = \frac{d\alpha^2\left(1-\alpha^{2N}\right)}{(1-\alpha)(1+\alpha)},$$

$$\mathbb{E}\left[ \sum_{i=0}^{N-1} \alpha^{i+1} y_{N-i} \boldsymbol{w} \right] = \boldsymbol{0}.$$

The proof of lemma A.3 is in Section C.2.

## A.2 Output, Loss, Gradient

This section we derive the output of Mamba given sequence $\{e_{1:N}, e_q\}$, and establishe the loss function formulation with its gradient expression.

**Linear Recurrence.** To start with, we show how the hidden states update when receiving token $e_l = (x_l^\top, y_l)^\top$. By (Eq. (5)) and Assumption 4.1(2), we have $\Delta_l = \text{softplus}(\ln(\exp((\ln 2)/N) - 1)) = (\ln 2)/N$. Combining it with (Eq. (1)(2)) and get:

$$
\begin{aligned}
h_l^{(d+1)} &= \overline{A}_l h_{l-1}^{(d+1)} + \overline{B}_l y_l \\
&= \exp(\Delta_l A) h_{l-1}^{(d+1)} + y_l (\Delta_l A)^{-1} (\exp(\Delta_l A) - I) \Delta_l B_l \\
&= \exp(-\Delta_l) I h_{l-1}^{(d+1)} - y_l \Delta_l^{-1} (\exp(-\Delta_l) I - I) \Delta_l B_l \\
&= \exp(-\Delta_l) h_{l-1}^{(d+1)} + (1 - \exp(-\Delta_l)) y_l B_l \\
&= \alpha h_{l-1}^{(d+1)} + (1 - \alpha) y_l B_l
\end{aligned}
\tag{18}
$$

where $\alpha := \exp(-\Delta_l) = \exp((-\ln 2)/N)$, the second equality is by discretization rule (2), the third equality is by Assumption 4.1(2) and $exp(-\Delta_l I) = exp(-\Delta_l) I$. (Eq. (18)) is similar to theorem 1 in Gu and Dao (2024)

**Prediction Output.** We next derive the expression of $\hat{y}_q$. Based on (Eq.(11)), the hidden state after receiving the first $l$ context prompts $e_{1:l}$ is given by:

$$
\begin{aligned}
h_l^{(d+1)} &= \alpha h_{l-1}^{(d+1)} + (1 - \alpha) y_l B_l \\
&= \alpha^2 h_{l-2}^{(d+1)} + (1 - \alpha) y_l B_l + (1 - \alpha) \alpha y_{l-1} B_{l-1} \\
&= \dots \\
&= (1 - \alpha) \sum_{i=0}^{l-1} \alpha^i y_{l-i} B_{l-i}
\end{aligned}
\tag{19}
$$

Receiving the query token $e_q = (x_q^\top, 0)^\top$, we have:

$$
h_{N+1}^{(d+1)} = \alpha h_N^{(d+1)} + (1 - \alpha) \cdot 0 \cdot B_N = (1 - \alpha) \sum_{i=0}^{N-1} \alpha^{i+1} y_{N-i} B_{N-i}
\tag{20}
$$

Finally, the prediction output is as follows

$$
\begin{aligned}
\hat{y}_q &= C_{N+1}^\top h_{N+1}^{(d+1)} = (1 - \alpha) C_{N+1}^\top \sum_{i=0}^{N-1} \alpha^{i+1} y_{N-i} B_{N-i} \\
&= (1 - \alpha) (W_C e_q + b_C)^\top \sum_{i=0}^{N-1} \alpha^{i+1} y_{N-i} (W_B e_{N-i} + b_B)
\end{aligned}
\tag{21}
$$

To handle $W_C e_q$ and $W_B e_{N-i}$, we further denote $W_B = [B \, b]$ and $W_C = [C \, c]$, where $B, C \in \mathbb{R}^{d_h \times d}$, $b, c \in \mathbb{R}^{d_h \times 1}$. Then we write another form of (Eq. (21)):

$$
\hat{y}_q = (1 - \alpha)(C x_q + b_C)^\top \sum_{i=0}^{N-1} \alpha^{i+1} y_{N-i} (B x_{N-i} + y_{N-i} b + b_B)
\tag{22}
$$

The loss becomes:

$$
\mathcal{L}(\theta) = \frac{1}{2} \mathbb{E}\left[ \left( (1 - \alpha)(C x_q + b_C)^\top \sum_{i=0}^{N-1} \alpha^{i+1} y_{N-i} (B x_{N-i} + y_{N-i} b + b_B) - w^\top x_q \right)^2 \right]
\tag{23}
$$

The following lemma provides the gradient of $B, C, b, b_B, b_C$ with respect to loss (Eq. (23)).

**Lemma A.4 (Gradient)** *The gradient of trainable parameters $\boldsymbol{\theta}' = \{\boldsymbol{B}, \boldsymbol{C}, \boldsymbol{b}, \boldsymbol{b}_B, \boldsymbol{b}_C\}$ with respect to loss (Eq. (23)) are as follows:*

$$\nabla_{\boldsymbol{b}_B}\mathcal{L}(\boldsymbol{\theta}) = \boldsymbol{0},$$

$$\nabla_{\boldsymbol{b}_C}\mathcal{L}(\boldsymbol{\theta}) = \boldsymbol{0},$$

$$\nabla_{\boldsymbol{B}}\mathcal{L}(\boldsymbol{\theta}) = \underbrace{\left(\alpha^2\big(1-\alpha^N\big)^2 + \frac{(d+1)\alpha^2(1-\alpha)\big(1-\alpha^{2N}\big)}{(1+\alpha)}\right)}_{:=\beta_1}\boldsymbol{C}\boldsymbol{C}^\top\boldsymbol{B} - \underbrace{\alpha\big(1-\alpha^N\big)}_{:=\beta_3}\boldsymbol{C},$$

$$\nabla_{\boldsymbol{b}}\mathcal{L}(\boldsymbol{\theta}) = \underbrace{\left(d^2\alpha^2\big(1-\alpha^N\big)^2 + \frac{(2d^2+6d)\alpha^2(1-\alpha)\big(1-\alpha^{2N}\big)}{(1+\alpha)}\right)}_{:=\beta_2}\boldsymbol{C}\boldsymbol{C}^\top\boldsymbol{b},$$

$$\nabla_{\boldsymbol{C}}\mathcal{L}(\boldsymbol{\theta}) = \underbrace{\left(\alpha^2\big(1-\alpha^N\big)^2 + \frac{(d+1)\alpha^2(1-\alpha)\big(1-\alpha^{2N}\big)}{(1+\alpha)}\right)}_{:=\beta_1}\boldsymbol{B}\boldsymbol{B}^\top\boldsymbol{C}$$

$$+ \underbrace{\left(d^2\alpha^2\big(1-\alpha^N\big)^2 + \frac{(2d^2+6d)\alpha^2(1-\alpha)\big(1-\alpha^{2N}\big)}{(1+\alpha)}\right)}_{:=\beta_2}\boldsymbol{b}\boldsymbol{b}^\top\boldsymbol{C}$$

$$- \underbrace{\alpha\big(1-\alpha^N\big)}_{:=\beta_3}\boldsymbol{B}.$$

Here, we denote $\beta_1 = \left(\alpha^2\big(1-\alpha^N\big)^2 + \frac{(d+1)\alpha^2(1-\alpha)\big(1-\alpha^{2N}\big)}{(1+\alpha)}\right)$, $\beta_2 = \big(d^2\alpha^2\big(1-\alpha^N\big)^2 + \frac{(2d^2+6d)\alpha^2(1-\alpha)\big(1-\alpha^{2N}\big)}{(1+\alpha)}\big)$, $\beta_3 = \alpha\big(1-\alpha^N\big)$ for simplicity. The proof of lemma A.4 is in Section C.3.

## A.3 Training Dynamics

With the gradient in lemma A.4, we further provide the update rule for Mamba's parameters and the *Vector-coupled Dynamics*.

Using gradient descent algorithm $\boldsymbol{\theta}'(t+1) = \boldsymbol{\theta}'(t) - \eta\nabla_{\boldsymbol{\theta}'}\mathcal{L}(\boldsymbol{\theta}(t))$ with training rate $\eta$, we have the following update rule base on lemma A.4.

**Lemma A.5 (Update Rule, restatement of lemma 5.1)** *Let $\eta$ be the learning rate and we use gradient descent to update the weights $\boldsymbol{W}_B, \boldsymbol{W}_C, \boldsymbol{b}_B, \boldsymbol{b}_C$, for $t \geq 0$ we have*

$$\boldsymbol{B}(t+1) = \boldsymbol{B}(t) + \eta\beta_3\boldsymbol{C}(t) - \eta\beta_1\boldsymbol{C}(t)\boldsymbol{C}(t)^\top\boldsymbol{B}(t),$$

$$\boldsymbol{C}(t+1) = \boldsymbol{C}(t) + \eta\beta_3\boldsymbol{B}(t) - \eta\beta_1\boldsymbol{B}(t)\boldsymbol{B}(t)^\top\boldsymbol{C}(t) - \eta\beta_2\boldsymbol{b}(t)\boldsymbol{b}(t)^\top\boldsymbol{C}(t),$$

$$\boldsymbol{b}(t+1) = \boldsymbol{b}(t) - \eta\beta_2\boldsymbol{C}(t)\boldsymbol{C}(t)^\top\boldsymbol{b}(t),$$

$$\boldsymbol{b}_B(t) = \boldsymbol{b}_C(t) = \boldsymbol{0}.$$

We decompose $\boldsymbol{B}$ and $\boldsymbol{C}$ as $\boldsymbol{B} = [\boldsymbol{b}_1 \ldots \boldsymbol{b}_d]$, $\boldsymbol{C} = [\boldsymbol{c}_1 \ldots \boldsymbol{c}_d]$, and provide the update rule for $\boldsymbol{b}_i, \boldsymbol{c}_i$ and $\boldsymbol{b}$ with $i \in [1:d]$ as the following lemma.

**Lemma A.6 (Vectors Update Rule, restatement of lemma 5.2)** *Let $\eta$ be the learning rate and we use gradient descent to update the weights $\boldsymbol{W}_B, \boldsymbol{W}_C, \boldsymbol{b}_B, \boldsymbol{b}_C$, for $i \in [d]$, $t \geq 0$ we have*

$$\boldsymbol{b}_i(t+1) = \boldsymbol{b}_i(t) + \eta\left(\big(\beta_3 - \beta_1\boldsymbol{c}_i^\top(t)\boldsymbol{b}_i(t)\big)\boldsymbol{c}_i(t) - \beta_1\sum_{k\neq i}^{d}\boldsymbol{c}_k^\top(t)\boldsymbol{b}_i(t)\cdot\boldsymbol{c}_k(t)\right)$$

$$=: \boldsymbol{b}_i(t) + \eta \bar{\boldsymbol{b}}_i(t)$$

$$\boldsymbol{c}_i(t+1) = \boldsymbol{c}_i(t) + \eta\Big( \big(\beta_3 - \beta_1 \boldsymbol{c}_i^\top(t)\boldsymbol{b}_i(t)\big)\boldsymbol{b}_i(t) - \beta_1 \sum_{k\neq i}^d \boldsymbol{c}_i^\top(t)\boldsymbol{b}_k(t) \cdot \boldsymbol{b}_k(t) - \beta_2 \boldsymbol{c}_i^\top(t)\boldsymbol{b}(t) \cdot \boldsymbol{b}(t) \Big)$$

$$=: \boldsymbol{c}_i(t) + \eta \bar{\boldsymbol{c}}_i(t)$$

$$\boldsymbol{b}(t+1) = \boldsymbol{b}(t) - \eta\Big( \beta_2 \sum_{k=1}^d \boldsymbol{c}_k^\top(t)\boldsymbol{b}(t) \cdot \boldsymbol{c}_k(t) \Big) =: \boldsymbol{b}(t) + \eta \bar{\boldsymbol{b}}(t)$$

Here, we denote $\eta \bar{\boldsymbol{b}}_i(t) = \boldsymbol{b}_i(t+1) - \boldsymbol{b}_i(t)$, $\eta \bar{\boldsymbol{c}}_i(t) = \boldsymbol{c}_i(t+1) - \boldsymbol{c}_i(t)$, and $\eta \bar{\boldsymbol{b}}(t) = \boldsymbol{b}(t+1) - \boldsymbol{b}(t)$ for simplicity.

Next, we provide the dynamics for the inner products of these vectors.

**Lemma A.7 (Vector-coupled Dynamics)** *Let $\eta$ be the learning rate and we use gradient descent to update the weights $\boldsymbol{W}_B, \boldsymbol{W}_C, \boldsymbol{b}_B, \boldsymbol{b}_C$, we have*

$$\boldsymbol{b}_i^\top(t+1)\boldsymbol{b}_i(t+1)$$

$$= \boldsymbol{b}_i^\top(t)\boldsymbol{b}_i(t) + 2\eta\Big( \big(\beta_3 - \beta_1 \boldsymbol{c}_i^\top(t)\boldsymbol{b}_i(t)\big)\boldsymbol{c}_i^\top(t)\boldsymbol{b}_i(t) - \beta_1 \sum_{k\neq i}^d \big(\boldsymbol{c}_k^\top(t)\boldsymbol{b}_i(t)\big)^2 \Big)$$

$$+ \eta^2 \big\| \bar{\boldsymbol{b}}_i(t) \big\|_2^2$$

$$\boldsymbol{b}_i^\top(t+1)\boldsymbol{b}_j(t+1)$$
$$= \boldsymbol{b}_i^\top(t)\boldsymbol{b}_j(t) + \eta\Big( 2\big(\beta_3 - \beta_1 \boldsymbol{c}_i^\top(t)\boldsymbol{b}_i(t)\big)\boldsymbol{c}_i^\top(t)\boldsymbol{b}_j(t) + 2\big(\beta_3 - \beta_1 \boldsymbol{c}_j^\top(t)\boldsymbol{b}_j(t)\big)\boldsymbol{c}_j^\top(t)\boldsymbol{b}_i(t)$$

$$- \beta_3\big(\boldsymbol{c}_i^\top(t)\boldsymbol{b}_j(t) + \boldsymbol{c}_j^\top(t)\boldsymbol{b}_i(t)\big) - 2\beta_1 \sum_{k\neq i, k\neq j}^d \boldsymbol{c}_k^\top(t)\boldsymbol{b}_i(t) \cdot \boldsymbol{c}_k^\top(t)\boldsymbol{b}_j(t) \Big) + \eta^2 \bar{\boldsymbol{b}}_i^\top(t)\bar{\boldsymbol{b}}_j(t)$$

$$\boldsymbol{c}_i^\top(t+1)\boldsymbol{c}_i(t+1)$$

$$= \boldsymbol{c}_i^\top(t)\boldsymbol{c}_i(t) + 2\eta\Big( \big(\beta_3 - \beta_1 \boldsymbol{c}_i^\top(t)\boldsymbol{b}_i(t)\big)\boldsymbol{c}_i^\top(t)\boldsymbol{b}_i(t) - \beta_1 \sum_{k\neq i}^d \big(\boldsymbol{c}_i^\top(t)\boldsymbol{b}_k(t)\big)^2$$

$$- \beta_2\big(\boldsymbol{c}_i^\top(t)\boldsymbol{b}(t)\big)^2 \Big) + \eta^2 \big\| \bar{\boldsymbol{c}}_i(t) \big\|_2^2$$

$$\boldsymbol{c}_i^\top(t+1)\boldsymbol{c}_j(t+1)$$
$$= \boldsymbol{c}_i^\top(t)\boldsymbol{c}_j(t) + \eta\Big( 2\big(\beta_3 - \beta_1 \boldsymbol{c}_i^\top(t)\boldsymbol{b}_i(t)\big)\boldsymbol{c}_j^\top(t)\boldsymbol{b}_i(t) + 2\big(\beta_3 - \beta_1 \boldsymbol{c}_j^\top(t)\boldsymbol{b}_j(t)\big)\boldsymbol{c}_i^\top(t)\boldsymbol{b}_j(t)$$

$$- \beta_3\big(\boldsymbol{c}_i^\top(t)\boldsymbol{b}_j(t) + \boldsymbol{c}_j^\top(t)\boldsymbol{b}_i(t)\big) - 2\beta_1 \sum_{k\neq i, k\neq j}^d \boldsymbol{c}_i^\top(t)\boldsymbol{b}_k(t) \cdot \boldsymbol{c}_j^\top(t)\boldsymbol{b}_k(t)$$

$$- 2\beta_2 \boldsymbol{c}_i^\top(t)\boldsymbol{b}(t) \cdot \boldsymbol{c}_j^\top(t)\boldsymbol{b}(t) \Big) + \eta^2 \bar{\boldsymbol{c}}_i^\top(t)\bar{\boldsymbol{c}}_j(t)$$

$$\boldsymbol{c}_i^\top(t+1)\boldsymbol{b}_i(t+1)$$

$$= \boldsymbol{c}_i^\top(t)\boldsymbol{b}_i(t) + \eta\Big( \big(\beta_3 - \beta_1 \boldsymbol{c}_i^\top(t)\boldsymbol{b}_i(t)\big)\boldsymbol{b}_i^\top(t)\boldsymbol{b}_i(t) - \beta_1 \sum_{k\neq i}^d \boldsymbol{c}_i^\top(t)\boldsymbol{b}_k(t) \cdot \boldsymbol{b}_k^\top(t)\boldsymbol{b}_i(t)$$

$$- \beta_2 \boldsymbol{c}_i^\top(t)\boldsymbol{b}(t) \cdot \boldsymbol{b}_i^\top(t)\boldsymbol{b}(t) + \big(\beta_3 - \beta_1 \boldsymbol{c}_i^\top(t)\boldsymbol{b}_i(t)\big)\boldsymbol{c}_i^\top(t)\boldsymbol{c}_i(t)$$

$$- \beta_1 \sum_{k \neq i} \boldsymbol{c}_k^\top(t)\boldsymbol{b}_i(t) \cdot \boldsymbol{c}_i^\top(t)\boldsymbol{c}_k(t) \Big) + \eta^2 \bar{\boldsymbol{c}}_i^\top(t)\bar{\boldsymbol{b}}_i(t)$$

$$\boldsymbol{c}_i^\top(t+1)\boldsymbol{b}_j(t+1)$$
$$= \Big(1 - \eta\beta_1\big(\boldsymbol{c}_i^\top(t)\boldsymbol{c}_i(t) + \boldsymbol{b}_j^\top(t)\boldsymbol{b}_j(t)\big)\Big)\boldsymbol{c}_i^\top(t)\boldsymbol{b}_j(t)$$
$$+ \eta\Big( \big(\beta_3 - \beta_1\boldsymbol{c}_i^\top(t)\boldsymbol{b}_i(t)\big)\boldsymbol{b}_i^\top(t)\boldsymbol{b}_j(t) - \beta_1 \sum_{k \neq i, k \neq j}^{d} \boldsymbol{c}_i^\top(t)\boldsymbol{b}_k(t) \cdot \boldsymbol{b}_k^\top(t)\boldsymbol{b}_j(t)$$
$$- \beta_2\boldsymbol{c}_i^\top(t)\boldsymbol{b}(t) \cdot \boldsymbol{b}_j^\top(t)\boldsymbol{b}(t) + \big(\beta_3 - \beta_1\boldsymbol{c}_j^\top(t)\boldsymbol{b}_j(t)\big)\boldsymbol{c}_i^\top(t)\boldsymbol{c}_j(t)$$
$$- \beta_1 \sum_{k \neq i, k \neq j}^{d} \boldsymbol{c}_k^\top(t)\boldsymbol{b}_j(t) \cdot \boldsymbol{c}_i^\top(t)\boldsymbol{c}_k(t) \Big) + \eta^2 \bar{\boldsymbol{c}}_i^\top(t)\bar{\boldsymbol{b}}_j(t)$$

$$\boldsymbol{b}^\top(t+1)\boldsymbol{b}(t+1) = \boldsymbol{b}^\top(t)\boldsymbol{b}(t) - 2\eta\Big(\beta_2 \sum_{k=1}^{d} \big(\boldsymbol{c}_k^\top(t)\boldsymbol{b}(t)\big)^2\Big) + \eta^2 \big\|\bar{\boldsymbol{b}}(t)\big\|_2^2$$

$$\boldsymbol{b}_i^\top(t+1)\boldsymbol{b}(t+1)$$
$$= \boldsymbol{b}_i^\top(t)\boldsymbol{b}(t) + \eta\Big( \big(\beta_3 - \beta_1\boldsymbol{c}_i^\top(t)\boldsymbol{b}_i(t)\big)\boldsymbol{c}_i^\top(t)\boldsymbol{b}(t) - \beta_1 \sum_{k \neq i}^{d} \boldsymbol{c}_k^\top(t)\boldsymbol{b}_i(t) \cdot \boldsymbol{c}_k^\top(t)\boldsymbol{b}(t)$$
$$- \beta_2 \sum_{k=1}^{d} \boldsymbol{c}_k^\top(t)\boldsymbol{b}(t) \cdot \boldsymbol{c}_k^\top(t)\boldsymbol{b}_i(t) \Big) + \eta^2 \bar{\boldsymbol{b}}_i^\top(t)\bar{\boldsymbol{b}}(t)$$

$$\boldsymbol{c}_i^\top(t+1)\boldsymbol{b}(t+1)$$
$$= \Big(1 - \eta\beta_2\big(\boldsymbol{b}^\top(t)\boldsymbol{b}(t) + \boldsymbol{c}_i^\top(t)\boldsymbol{c}_i(t)\big)\Big)\boldsymbol{c}_i^\top(t)\boldsymbol{b}(t)$$
$$+ \eta\Big( \big(\beta_3 - \beta_1\boldsymbol{c}_i^\top(t)\boldsymbol{b}_i(t)\big)\boldsymbol{b}_i^\top(t)\boldsymbol{b}(t) - \beta_1 \sum_{k \neq i}^{d} \boldsymbol{c}_i^\top(t)\boldsymbol{b}_k(t) \cdot \boldsymbol{b}_k^\top(t)\boldsymbol{b}(t)$$
$$- \beta_2 \sum_{k \neq i}^{d} \boldsymbol{c}_k^\top(t)\boldsymbol{b}(t) \cdot \boldsymbol{c}_k^\top(t)\boldsymbol{c}_i(t) \Big) + \eta^2 \bar{\boldsymbol{c}}_i^\top(t)\bar{\boldsymbol{b}}(t)$$

Lemma A.7 is derive by calculating the inner products of the vectors update rule in lemma A.6. For example, $\boldsymbol{b}^\top(t+1)\boldsymbol{b}(t+1)$ is derived as follow:

$$\boldsymbol{b}^\top(t+1)\boldsymbol{b}(t+1) = \Big(\boldsymbol{b}(t) + \eta\bar{\boldsymbol{b}}(t)\Big)^\top \Big(\boldsymbol{b}(t) + \eta\bar{\boldsymbol{b}}(t)\Big)$$
$$= \boldsymbol{b}^\top(t)\boldsymbol{b}(t) - 2\eta\bar{\boldsymbol{b}}(t)^\top\boldsymbol{b}(t) + \eta^2 \big\|\bar{\boldsymbol{b}}(t)\big\|_2^2$$
$$= \boldsymbol{b}^\top(t)\boldsymbol{b}(t) - 2\eta\Big(\beta_2 \sum_{k=1}^{d} \big(\boldsymbol{c}_k^\top(t)\boldsymbol{b}(t)\big)^2\Big) + \eta^2 \big\|\bar{\boldsymbol{b}}(t)\big\|_2^2$$

The other equations are similar to it.

# B  Proof of Theorem 4.1

In this section, we present the framework of *Fine-grained Induction*, and establish the results of Theorem 4.1 after convergence.

**Fine-gained Induction**    Specifically, denoting $\delta(t) = \max_{s \in [0,t]}\{|\boldsymbol{b}_i^\top(s)\boldsymbol{b}_j(s)|, |\boldsymbol{c}_i^\top(s)\boldsymbol{c}_j(s)|, |\boldsymbol{b}_i^\top(s)\boldsymbol{b}(s)|\}$ and $\gamma = \min_{t \geq 0}\{\boldsymbol{b}_i^\top(t)\boldsymbol{b}_i(t), \boldsymbol{c}_i^\top(t)\boldsymbol{c}_i(t), \boldsymbol{b}^\top(t)\boldsymbol{b}(t)\}$, we establish the following three properties $\mathcal{A}(t)$, $\mathcal{B}(t)$, and $\mathcal{C}(t)$ simultaneously for $t \geq 0$:

- $\mathcal{A}(t)$ :
$$d_h/2 \leq \boldsymbol{b}_i^\top(t)\boldsymbol{b}_i(t), \boldsymbol{c}_i^\top(t)\boldsymbol{c}_i(t), \boldsymbol{b}^\top(t)\boldsymbol{b}(t) \leq 2d_h$$

- $\mathcal{B}(t)$ :
$$|\beta_3 - \beta_1\boldsymbol{c}_i^\top(t)\boldsymbol{b}_i(t)| \leq \delta(t)\exp(-\eta\beta_1\gamma t)$$
$$|\boldsymbol{c}_i^\top(t)\boldsymbol{b}_j(t)| \leq 2\delta(t)\exp(-\eta\beta_1\gamma t)$$
$$|\boldsymbol{c}_i^\top(t)\boldsymbol{b}(t)| \leq 2\delta(t)\exp(-\eta\beta_2\gamma t) + \frac{\delta(t)}{\beta_2}\exp(-\eta\beta_1\gamma t)$$

- $\mathcal{C}(t)$ :
$$|\boldsymbol{b}_i^\top(t)\boldsymbol{b}_j(t)|, |\boldsymbol{c}_i^\top(t)\boldsymbol{c}_j(t)|, |\boldsymbol{b}_i^\top(t)\boldsymbol{b}(t)| \leq \delta(t) \leq 3\sqrt{d_h\log(4d(2d+1)/\delta)} =: \delta_{\max}$$

Here, $i, j \in [1, d], i \neq j$. The initial conditions $\mathcal{A}(0)$, $\mathcal{B}(0)$, and $\mathcal{C}(0)$ are established with high probability by concentration inequalities (lemma A.1). We also provide the following claims to establish $\mathcal{A}(t)$, $\mathcal{B}(t)$, and $\mathcal{C}(t)$ for $t \geq 0$:

**Claim B.1** $\mathcal{A}(0), \ldots, \mathcal{A}(T), \mathcal{B}(0), \ldots, \mathcal{B}(T), \mathcal{C}(0), \ldots, \mathcal{C}(T) \Longrightarrow \mathcal{A}(T+1)$

**Claim B.2** $\mathcal{A}(0), \ldots, \mathcal{A}(T), \mathcal{B}(0), \ldots, \mathcal{B}(T), \mathcal{C}(0), \ldots, \mathcal{C}(T) \Longrightarrow \mathcal{B}(T+1)$

**Claim B.3** $\mathcal{A}(0), \ldots, \mathcal{A}(T), \mathcal{B}(0), \ldots, \mathcal{B}(T), \mathcal{C}(0), \ldots, \mathcal{C}(T) \Longrightarrow \mathcal{C}(T+1)$

**Remark.** Property $\mathcal{A}(t)$ establishes the stability of quadratic norms:
$$\min\left\{\boldsymbol{b}_i(t)^\top\boldsymbol{b}_i(t), \boldsymbol{c}_i(t)^\top\boldsymbol{c}_i(t), \boldsymbol{b}(t)^\top\boldsymbol{b}(t)\right\} \geq d_h/2.$$

This norm lower bound induces two critical effects:

1. Convergence Rate: As we can see in property $\mathcal{B}(t)$, The upper bound of $\boldsymbol{c}_i^\top(t)\boldsymbol{b}_i(t)$, $\boldsymbol{c}_i^\top(t)\boldsymbol{b}_j(t)$ and $\boldsymbol{c}_i^\top(t)\boldsymbol{b}(t)$ is related to $\gamma$ (lower bound of the squared norms), thus the stability of quadratic norms ensure a stable rapid convergence rate for property $\mathcal{B}(t)$.

2. Saddle Point Avoidance: The strict positivity ($> 0$) of $|\boldsymbol{b}_i|^2$ and $|\boldsymbol{c}_i|^2$ prevents the dynamics collapse to undesirable solutions $\boldsymbol{b}_i = \boldsymbol{c}_i = \boldsymbol{0}$, which would permanently make $\boldsymbol{c}_i^\top\boldsymbol{b}_i = 0$ (saddle points).

Property $\mathcal{B}(t)$ establishes a rapid exponential convergence rate:
$$\boldsymbol{C}^\top\boldsymbol{B} \to \frac{\beta_3}{\beta_1}\boldsymbol{I}, \quad \boldsymbol{C}^\top\boldsymbol{b} \to \boldsymbol{0}$$

The rapid convergence rate ensures that the variations of *Squared norms* (in property $\mathcal{A}(t)$) and *Cross-interactions* (in property $\mathcal{C}(t)$) remain bounded, thereby establishing their constraints. For example, at initialization, $\boldsymbol{b}_i^\top(0)\boldsymbol{b}_i(0)$ is bounded by $3d_h/4 \leq \boldsymbol{b}_i^\top(0)\boldsymbol{b}_i(0) \leq 5d_h/4$. Further, thanks to the exponential convergence rate in property $\mathcal{B}(t)$, we can prove that $|\boldsymbol{b}_i^\top(t)\boldsymbol{b}_i(t) - \boldsymbol{b}_i^\top(0)\boldsymbol{b}_i(0)| \leq d_h/4$, and therefore $d_h/2 \leq \boldsymbol{b}_i^\top(t)\boldsymbol{b}_i(t), \boldsymbol{c}_i^\top(t)\boldsymbol{c}_i(t), \boldsymbol{b}^\top(t)\boldsymbol{b}(t) \leq 3d_h/2 \leq 2d_h$.

Property $\mathcal{C}(t)$ establishes the upper bound for the *Cross-interactions*. As we discuss in section 5.2, if the *Squared norms* ( $\boldsymbol{c}_i^\top(t)\boldsymbol{b}_i(t)$, $\boldsymbol{c}_i^\top(t)\boldsymbol{b}_j(t)$ and $\boldsymbol{c}_i^\top(t)\boldsymbol{b}(t)$ ) are larger enough than the *Cross-interactions* ( $\boldsymbol{b}_i^\top(t)\boldsymbol{b}_j(t), \boldsymbol{c}_i^\top(t)\boldsymbol{c}_j(t)$, and $\boldsymbol{b}_i^\top(t)\boldsymbol{b}(t)$ ), we can make use of the *negative feedback term* to establish an exponential convergence rate. Thus property $\mathcal{C}(t)$ is also important.

The proof of claim B.1, claim B.2, and claim B.3 are in section C.4, section C.5, and section C.6 respectively.

**Proof of Theorem 4.1** After convergence ($t \to 0$), we will have $C^\top B = \frac{\beta_3}{\beta_1} I, C^\top b = 0$ (by property $\mathcal{B}(t)$), and $b_B(t) = b_C(t) = 0$ (by lemma A.5).

We will restate some equality for ease of reference.

**Linear Recurrence** (restatement of (Eq. (18)))

$$h_l^{(d+1)} = \alpha h_{l-1}^{(d+1)} + (1-\alpha)y_l B_l \tag{24}$$

**Prediction Output** (restatement of (Eq. (22)))

$$\hat{y}_q = (1-\alpha)(C x_q + b_C)^\top \sum_{i=0}^{N-1} \alpha^{i+1} y_{N-i}(B x_{N-i} + y_{N-i}b + b_B) \tag{25}$$

**Loss** (restatement of (Eq. (23)))

$$\mathcal{L}(\theta) = \frac{1}{2}\mathbb{E}\left[\left((1-\alpha)(C x_q + b_C)^\top \sum_{i=0}^{N-1} \alpha^{i+1} y_{N-i}(B x_{N-i} + y_{N-i}b + b_B) - w^\top x_q\right)^2\right] \tag{26}$$

Based on (Eq. (24)), we have

$$
\begin{aligned}
(W_C^\top)_{[1:d,:]}(t)h_l^{(d+1)} &= \alpha(W_C^\top)_{[1:d,:]}(t)h_{l-1}^{(d+1)} + (1-\alpha)y_l(W_C^\top)_{[1:d,:]}(t)B_l \\
&= \alpha(W_C^\top)_{[1:d,:]}(t)h_{l-1}^{(d+1)} + (1-\alpha)y_l C^\top(t)(B(t)x_l + y_l b(t) + b_B(t)) \\
&= \alpha(W_C^\top)_{[1:d,:]}(t)h_{l-1}^{(d+1)} + (1-\alpha)y_l C^\top(t)B(t)x_l + (1-\alpha)y_l^2 C^\top(t)b(t) \\
&= \alpha(W_C^\top)_{[1:d,:]}(t)h_{l-1}^{(d+1)} + (1-\alpha)\frac{\beta_3}{\beta_1}y_l x_l \\
&= \alpha(W_C^\top)_{[1:d,:]}(t)h_{l-1}^{(d+1)} + \frac{2(1+\alpha)(1-\alpha)}{\alpha(3(1-\alpha)d + 4 - 2\alpha)}y_l x_l
\end{aligned}
\tag{27}
$$

where the second equality is by selection rule $B_l = W_B e_l + b_B$ (Eq. (3)), and $W_B = [B\ b]$, $e_l = (x_l^\top, y_l)^\top$. The third equality is by $b_B(t) = 0$. The fourth equality is by $C^\top B = \frac{\beta_3}{\beta_1}I$ and $C^\top b = 0$. (Eq. (27)) establish the first equation (Thm 4.1 (a)) of the Theorem.

Based on (Eq. (25)), we have

$$
\begin{aligned}
\hat{y}_q &= (1-\alpha)(C x_q + b_C)^\top \sum_{i=0}^{N-1} \alpha^{i+1} y_{N-i}(B x_{N-i} + y_{N-i}b + b_B) \\
&= x_q^\top C^\top \sum_{i=0}^{N-1}(1-\alpha)\alpha^{i+1} y_{N-i}(B x_{N-i} + y_{N-i}b) \\
&= x_q^\top \sum_{i=0}^{N-1}(1-\alpha)\alpha^{i+1} y_{N-i}C^\top B x_{N-i} + x_q^\top \sum_{i=0}^{N-1}(1-\alpha)\alpha^{i+1} y_{N-i}^2 C^\top b \\
&= x_q^\top \sum_{i=0}^{N-1}(1-\alpha)\alpha^{i+1}\frac{\beta_3}{\beta_1}y_{N-i}x_{N-i} \\
&= x_q^\top \sum_{i=0}^{N-1}\frac{2\alpha^i(1+\alpha)(1-\alpha)}{(3(1-\alpha)d + 4 - 2\alpha)}y_{N-i}x_{N-i}
\end{aligned}
\tag{28}
$$

where the second equality is by $b_B(t) = b_C(t) = 0$. The fourth equality is by $C^\top B = \frac{\beta_3}{\beta_1}I$ and $C^\top b = 0$. (Eq. (28)) establish the second equation (Thm 4.1 (b)) of the Theorem.

Based on (Eq. (25)), we have

$$
\begin{aligned}
\mathcal{L}(\boldsymbol{\theta}) &= \frac{1}{2}\mathbb{E}\Big[\Big((1-\alpha)(\boldsymbol{C}\boldsymbol{x}_q+\boldsymbol{b}_C)^\top \sum_{i=0}^{N-1}\alpha^{i+1}y_{N-i}(\boldsymbol{B}\boldsymbol{x}_{N-i}+y_{N-i}\boldsymbol{b}+\boldsymbol{b}_B)-\boldsymbol{w}^\top\boldsymbol{x}_q\Big)^2\Big] \\
&= \frac{1}{2}\mathbb{E}\Big[\Big(\frac{\beta_3}{\beta_1}\boldsymbol{x}_q^\top\sum_{i=0}^{N-1}(1-\alpha)\alpha^{i+1}y_{N-i}\boldsymbol{x}_{N-i}-\boldsymbol{w}^\top\boldsymbol{x}_q\Big)^2\Big] \\
&= \frac{1}{2}\underbrace{\mathbb{E}\Big[\Big(\frac{\beta_3}{\beta_1}\boldsymbol{x}_q^\top\sum_{i=0}^{N-1}(1-\alpha)\alpha^{i+1}y_{N-i}\boldsymbol{x}_{N-i}\Big)^2\Big]}_{\spadesuit} \\
&\quad -\underbrace{\mathbb{E}\Big[\frac{\beta_3}{\beta_1}\boldsymbol{x}_q^\top\sum_{i=0}^{N-1}(1-\alpha)\alpha^{i+1}y_{N-i}\boldsymbol{x}_{N-i}\cdot\boldsymbol{w}^\top\boldsymbol{x}_q\Big]}_{\clubsuit} \\
&\quad +\frac{1}{2}\underbrace{\mathbb{E}\Big[\big(\boldsymbol{w}^\top\boldsymbol{x}_q\big)^2\Big]}_{=\,d\ (\text{by lemma A.2})}
\end{aligned}
\tag{29}
$$

We compute terms $\spadesuit$ and $\clubsuit$ as follows:

$$
\begin{aligned}
\spadesuit &= \mathbb{E}\Big[\Big(\frac{\beta_3}{\beta_1}\boldsymbol{x}_q^\top\sum_{i=0}^{N-1}(1-\alpha)\alpha^{i+1}y_{N-i}\boldsymbol{x}_{N-i}\Big)^2\Big] \\
&= \frac{\beta_3^2}{\beta_1^2}\mathbb{E}\Big[\boldsymbol{x}_q^\top\big(\sum_{i=0}^{N-1}(1-\alpha)\alpha^{i+1}y_{N-i}\boldsymbol{x}_{N-i}\big)\big(\sum_{i=0}^{N-1}(1-\alpha)\alpha^{i+1}y_{N-i}\boldsymbol{x}_{N-i}\big)^\top\boldsymbol{x}_q\Big] \\
&= \frac{(1-\alpha)^2\beta_3^2}{\beta_1^2}\mathbb{E}_{\boldsymbol{x}_q}\Big[\boldsymbol{x}_q^\top\mathbb{E}_{\boldsymbol{x}_{N-i},\boldsymbol{x}_{N-j},\boldsymbol{w}}\Big[\sum_{i=0}^{N-1}\sum_{j=0}^{N-1}\alpha^{i+j+2}y_{N-i}y_{N-j}\boldsymbol{x}_{N-i}\boldsymbol{x}_{N-j}^\top\Big]\boldsymbol{x}_q\Big] \\
&= \frac{(1-\alpha)^2\beta_3^2}{\beta_1^2}\cdot\Big(\frac{\alpha^2\big(1-\alpha^N\big)^2}{(1-\alpha)^2}+\frac{(d+1)\alpha^2\big(1-\alpha^{2N}\big)}{(1-\alpha)(1+\alpha)}\Big)\mathbb{E}\Big[\boldsymbol{x}_q^\top\boldsymbol{I}\boldsymbol{x}_q\Big] \\
&= \frac{d\beta_3^2}{\beta_1}
\end{aligned}
$$

For the fourth equality, $\mathbb{E}_{\boldsymbol{x}_{N-i},\boldsymbol{x}_{N-j},\boldsymbol{w}}\Big[\sum_{i=0}^{N-1}\sum_{j=0}^{N-1}\alpha^{i+j+2}y_{N-i}y_{N-j}\boldsymbol{x}_{N-i}\boldsymbol{x}_{N-j}^\top\Big]=\Big(\frac{\alpha^2\big(1-\alpha^N\big)^2}{(1-\alpha)^2}+\frac{(d+1)\alpha^2\big(1-\alpha^{2N}\big)}{(1-\alpha)(1+\alpha)}\Big)\cdot\boldsymbol{I}$ by lemma A.3. The last equality is by $\beta_1=\Big(\alpha^2\big(1-\alpha^N\big)^2+\frac{(d+1)\alpha^2(1-\alpha)\big(1-\alpha^{2N}\big)}{(1+\alpha)}\Big)$

$$
\begin{aligned}
\clubsuit &= \mathbb{E}\Big[\frac{\beta_3}{\beta_1}\boldsymbol{x}_q^\top\sum_{i=0}^{N-1}(1-\alpha)\alpha^{i+1}y_{N-i}\boldsymbol{x}_{N-i}\cdot\boldsymbol{w}^\top\boldsymbol{x}_q\Big] \\
&= \frac{(1-\alpha)\beta_3}{\beta_1}\mathbb{E}_{\boldsymbol{x}_q}\Big[\boldsymbol{x}_q^\top\mathbb{E}_{\boldsymbol{x}_{N-i},\boldsymbol{w}}\Big[\sum_{i=0}^{N-1}\alpha^{i+1}y_{N-i}\boldsymbol{x}_{N-i}\boldsymbol{w}^\top\Big]\boldsymbol{x}_q\Big] \\
&= \frac{(1-\alpha)\beta_3}{\beta_1}\cdot\alpha\Big(\frac{1-\alpha^N}{1-\alpha}\Big)\mathbb{E}\Big[\boldsymbol{x}_q^\top\boldsymbol{I}\boldsymbol{x}_q\Big] \\
&= \frac{d\beta_3^2}{\beta_1}
\end{aligned}
$$

For the third equality, $\mathbb{E}_{\boldsymbol{x}_{N-i},\boldsymbol{w}}\Big[\sum_{i=0}^{N-1}\alpha^{i+1}y_{N-i}\boldsymbol{x}_{N-i}\boldsymbol{w}^\top\Big]=\alpha\Big(\frac{1-\alpha^N}{1-\alpha}\Big)\boldsymbol{I}$ by lemma A.3. The last equality is by $\beta_3=\alpha\big(1-\alpha^N\big)$.

Substituting ♠ and ♣ into (Eq. (29)) and get:

$$
\begin{aligned}
\mathcal{L}(\boldsymbol{\theta}) &= \frac{d\beta_3^2}{2\beta_1} - \frac{d\beta_3^2}{\beta_1} + \frac{1}{2}d \\
&= \frac{d}{2}\left(1 - \frac{\beta_3^2}{\beta_1}\right) \\
&= \frac{d(d+1)\alpha^2(1-\alpha)\left(1-\alpha^{2N}\right)}{2(1+\alpha)\beta_1} \\
&= d(d+1)(1-\alpha) \cdot \frac{\alpha^2}{\beta_1} \cdot \frac{\left(1-\alpha^{2N}\right)}{2(1+\alpha)} \\
&\leq \frac{d(d+1)}{N} \cdot 4 \cdot \frac{3}{8} \\
&= \frac{3d(d+1)}{2N}
\end{aligned}
\tag{30}
$$

Recall $\beta_1 = \left(\alpha^2\left(1-\alpha^N\right)^2 + \frac{(d+1)\alpha^2(1-\alpha)\left(1-\alpha^{2N}\right)}{(1+\alpha)}\right)$, $\beta_3 = \alpha\left(1-\alpha^N\right)$ and $\alpha = \exp((-\ln 2)/N)$.
For the inequality, $1 - \alpha = 1 - \exp((-\ln 2)/N) \leq \frac{\ln 2}{N} \leq \frac{1}{N}$, $\beta_1 \geq \alpha^2\left(1-\alpha^N\right)^2 = \frac{1}{4}\alpha^2$,
$1 - \alpha^{2N} = 1 - \frac{1}{4} = \frac{3}{4}$, thus $d(d+1)(1-\alpha) \leq \frac{d(d+1)}{N}$, $\frac{\alpha^2}{\beta_1} \leq 4$, $\frac{\left(1-\alpha^{2N}\right)}{2(1+\alpha)} \leq \frac{3}{8(1+\alpha)} \leq \frac{3}{8}$. (Eq. (30)) establish the third equation (Thm 4.1 (c)) of the Theorem.

## C Complete Proof

This section presents the complete proof for the above results. To begin with, we provide the exact assumptions for $N$, $\eta$ and $d_h$ as part of Assumption 4.1.

**Assumption**

$$
N = \Omega(d) \geq \max\{\frac{2\ln 2}{\ln 6 - \ln 5}, \frac{3(d+1)\ln 2}{2}\}
$$

$$
\eta = O(d^{-2}d_h^{-1}) \leq \frac{1}{2d^2 d_h} \leq \frac{\ln 2}{\beta_2 d_h}
$$

$$
d_h = \widetilde{\Omega}(d^2) \geq \max\{\lambda_1, \ldots, \lambda_{11}\}
$$

where

$$
\lambda_1 = \left(1728\log(4d(2d+1)/\delta) + 576(d-1)\beta_1\log(4d(2d+1)/\delta)\right)/\beta_1
$$

$$
\lambda_2 = \left(576\log(4d(2d+1)/\delta) + 192\log(4d(2d+1)/\delta)\right)/\beta_1
$$

$$
\lambda_3 = \left(1728\log(4d(2d+1)/\delta) + (576d+1872)\beta_1\log(4d(2d+1)/\delta)\right)/\beta_1
$$

$$
\begin{aligned}
\lambda_4 = \ &576\log(4d(2d+1)/\delta)/\beta_1 + 192(d-1)\log(4d(2d+1)/\delta) \\
&+ 384\log(4d(2d+1)/\delta)/\beta_1 + 3840\ln 2\log(4d(2d+1)/\delta)
\end{aligned}
$$

$$
\lambda_5 = 2448d\log(4d(2d+1)/\delta)
$$

$$
\lambda_6 = 816d\log(4d(2d+1)/\delta) + 768\ln 2d^2\log(4d(2d+1)/\delta) + 48\log(4d(2d+1)/\delta)/\beta_1
$$

$$
\lambda_7 = \left(\frac{1}{\sqrt{\log(4d(2d+1)/\delta)}} + 24\sqrt{\log(4d(2d+1)/\delta)}\left(8\beta_1(d-1) + 10 + 6\beta_1 + 12\eta\beta_1/d\right)\right)^2
$$

$$
\lambda_8 = 36\log(4d(2d+1)/\delta)\left(\frac{8}{\beta_1} + 8(d-2) + 6 + \frac{12}{d}\right)^2
$$

$$
\lambda_9 = 36\log(4d(2d+1)/\delta)\left(4(d-1) + 56d\ln 2\right)^2
$$

$$
\lambda_{10} = 36\log(4d(2d+1)/\delta)\left(6 + 4\beta_1(d-1) + 2(d-1)\right)^2
$$

$$\lambda_{11} = \frac{36}{(\ln(3/2))^2} \log(4d(2d+1)/\delta) \left( \frac{32d}{\beta_1} + \frac{8\beta_3}{\beta_1} + 32d + \frac{8}{\beta_1} + \frac{4}{\beta_1\beta_2} + 80\ln 2 \right)^2$$

Note that under the assumption of $N \geq \max\{\frac{2\ln 2}{\ln 6 - \ln 5}, \frac{3(d+1)\ln 2}{2}\}$ and combining $\alpha = \exp((-\ln 2)/N)$, we have the following:

$$\frac{5}{6} \leq \alpha^2, \quad 4\beta_1 \leq \beta_2, \quad \frac{5}{24} \leq \beta_1 \leq \frac{3}{4}, \quad 1 \leq \frac{1}{2}d \leq \beta_2, \quad \beta_3 = \Theta(1)$$

These condition will be use to prove some bounds.

## C.1 Proof of Lemma A.2

**Lemma C.1 (restatement of lemma A.2)** *If vectors $\boldsymbol{x}$ and $\boldsymbol{w}$ are iid generated from $\mathcal{N}(0, \boldsymbol{I}_d)$, $y = \boldsymbol{x}^\top \boldsymbol{w}$ we have the following expectations:*

$$\mathbb{E}\left[\boldsymbol{x}\boldsymbol{x}^\top \boldsymbol{w}\boldsymbol{w}^\top \boldsymbol{x}\boldsymbol{x}^\top\right] = (d+2)\boldsymbol{I},$$

$$\mathbb{E}\left[y^2\right] = d,$$

$$\mathbb{E}\left[y^4\right] = 3d(d+2).$$

**Proof.** For $(i,j)$-th element of $\mathbb{E}\left[\boldsymbol{x}\boldsymbol{x}^\top \boldsymbol{w}\boldsymbol{w}^\top \boldsymbol{x}\boldsymbol{x}^\top\right]$, we have:

$$\mathbb{E}\left[\boldsymbol{x}\boldsymbol{x}^\top \boldsymbol{w}\boldsymbol{w}^\top \boldsymbol{x}\boldsymbol{x}^\top\right]_{[i,j]} = \mathbb{E}\left[\boldsymbol{x}_{[i]} \sum_{k=1}^d \left(\boldsymbol{x}_{[k]}\boldsymbol{w}_{[k]}\right) \sum_{l=1}^d \left(\boldsymbol{w}_{[l]}\boldsymbol{x}_{[l]}\right)\boldsymbol{x}_{[j]}\right]$$

$$= \sum_{k=1}^d \sum_{l=1}^d \mathbb{E}\left[\boldsymbol{x}_{[i]}\boldsymbol{x}_{[k]}\boldsymbol{x}_{[l]}\boldsymbol{x}_{[j]}\right]\mathbb{E}\left[\boldsymbol{w}_{[k]}\boldsymbol{w}_{[l]}\right]$$

According to the distribution of $\boldsymbol{w}$, we have $\mathbb{E}\left[\boldsymbol{w}_{[k]}\boldsymbol{w}_{[l]}\right] = \delta_{kl}$, where $\delta_{kl}$ is the Kronecker delta defined as:

$$\delta_{kl} = \begin{cases} 1 & \text{if } k = l, \\ 0 & \text{if } k \neq l, \end{cases}$$

By Isserlis Theorem, we have:

$$\mathbb{E}\left[\boldsymbol{x}_{[i]}\boldsymbol{x}_{[k]}\boldsymbol{x}_{[l]}\boldsymbol{x}_{[j]}\right]$$
$$= \mathbb{E}\left[\boldsymbol{x}_{[i]}\boldsymbol{x}_{[k]}\right]\mathbb{E}\left[\boldsymbol{x}_{[l]}\boldsymbol{x}_{[j]}\right] + \mathbb{E}\left[\boldsymbol{x}_{[i]}\boldsymbol{x}_{[l]}\right]\mathbb{E}\left[\boldsymbol{x}_{[k]}\boldsymbol{x}_{[j]}\right] + \mathbb{E}\left[\boldsymbol{x}_{[i]}\boldsymbol{x}_{[j]}\right]\mathbb{E}\left[\boldsymbol{x}_{[k]}\boldsymbol{x}_{[l]}\right]$$
$$= \delta_{ik}\delta_{lj} + \delta_{il}\delta_{kj} + \delta_{ij}\delta_{kl}$$

Then we have:

$$\mathbb{E}\left[\boldsymbol{x}\boldsymbol{x}^\top \boldsymbol{w}\boldsymbol{w}^\top \boldsymbol{x}\boldsymbol{x}^\top\right]_{[i,j]}$$
$$= \sum_{k=1}^d \sum_{l=1}^d \mathbb{E}\left[\boldsymbol{x}_{[i]}\boldsymbol{x}_{[k]}\boldsymbol{x}_{[l]}\boldsymbol{x}_{[j]}\right]\mathbb{E}\left[\boldsymbol{w}_{[k]}\boldsymbol{w}_{[l]}\right]$$
$$= \sum_{k=1}^d \sum_{l=1}^d \left(\delta_{ik}\delta_{lj} + \delta_{il}\delta_{kj} + \delta_{ij}\delta_{kl}\right)\delta_{kl}$$
$$= \sum_{k=1}^d \left(2\delta_{ik}\delta_{kj} + \delta_{ij}\delta_{kk}\right)$$
$$= (d+2)\delta_{ij}$$

Then we have:

$$\mathbb{E}\Big[\boldsymbol{x}\boldsymbol{x}^\top\boldsymbol{w}\boldsymbol{w}^\top\boldsymbol{x}\boldsymbol{x}^\top\Big] = (d+2)\boldsymbol{I}$$

$$\mathbb{E}\Big[y^2\Big] = \mathbb{E}\Big[\boldsymbol{x}^\top\boldsymbol{w}\cdot\boldsymbol{x}^\top\boldsymbol{w}\Big]$$

$$= \mathbb{E}\Big[\sum_{i=1}^d \big(\boldsymbol{x}_{[i]}\boldsymbol{w}_{[i]}\big)\sum_{j=1}^d \big(\boldsymbol{x}_{[j]}\boldsymbol{w}_{[j]}\big)\Big]$$

$$= \sum_{i=1}^d\sum_{j=1}^d \mathbb{E}\Big[\boldsymbol{x}_{[i]}\boldsymbol{x}_{[j]}\Big]\mathbb{E}\Big[\boldsymbol{w}_{[i]}\boldsymbol{w}_{[j]}\Big]$$

$$= \sum_{i=1}^d\sum_{j=1}^d \delta_{ij}^2$$

$$= d$$

$$\mathbb{E}\Big[y^4\Big] = \mathbb{E}\Big[\boldsymbol{x}^\top\boldsymbol{w}\cdot\boldsymbol{x}^\top\boldsymbol{w}\cdot\boldsymbol{x}^\top\boldsymbol{w}\cdot\boldsymbol{x}^\top\boldsymbol{w}\Big]$$

$$= \mathbb{E}\Big[\sum_{i=1}^d \big(\boldsymbol{x}_{[i]}\boldsymbol{w}_{[i]}\big)\sum_{j=1}^d \big(\boldsymbol{x}_{[j]}\boldsymbol{w}_{[j]}\big)\sum_{k=1}^d \big(\boldsymbol{x}_{[k]}\boldsymbol{w}_{[k]}\big)\sum_{l=1}^d \big(\boldsymbol{x}_{[l]}\boldsymbol{w}_{[l]}\big)\Big]$$

$$= \sum_{i=1}^d\sum_{j=1}^d\sum_{k=1}^d\sum_{l=1}^d \mathbb{E}\Big[\boldsymbol{x}_{[i]}\boldsymbol{x}_{[j]}\boldsymbol{x}_{[k]}\boldsymbol{x}_{[l]}\Big]\mathbb{E}\Big[\boldsymbol{w}_{[i]}\boldsymbol{w}_{[j]}\boldsymbol{w}_{[k]}\boldsymbol{w}_{[l]}\Big]$$

$$= \sum_{i=1}^d\sum_{j=1}^d\sum_{k=1}^d\sum_{l=1}^d \big(\delta_{ik}\delta_{lj} + \delta_{il}\delta_{kj} + \delta_{ij}\delta_{kl}\big)^2$$

$$= \Big(\underbrace{\sum_{i=j=k=l}}_{=d} + \underbrace{\sum_{i=j\neq k=l} + \sum_{i=k\neq j=l} + \sum_{i=k\neq j=l}}_{=3\cdot d(d-1)}\Big)\cdot\big(\delta_{ik}\delta_{lj} + \delta_{il}\delta_{kj} + \delta_{ij}\delta_{kl}\big)^2$$

$$= d\cdot 3^2 + 3\cdot d(d-1)\cdot 1^2$$

$$= 3d(d+2)$$

## C.2 Proof of Lemma A.3

**Lemma C.2 (restatement of lemma A.3)** *If vectors $\boldsymbol{x}_i$ and $\boldsymbol{w}$ are iid generated from $\mathcal{N}(0, \boldsymbol{I}_d)$, $y = \boldsymbol{x}_i^\top\boldsymbol{w}$ we have the following expectations:*

$$\mathbb{E}\Big[\sum_{i=0}^{N-1}\sum_{j=0}^{N-1}\alpha^{i+j+2}y_{N-i}y_{N-j}\boldsymbol{x}_{N-i}\boldsymbol{x}_{N-j}^\top\Big] = \Big(\frac{\alpha^2\big(1-\alpha^N\big)^2}{(1-\alpha)^2} + \frac{(d+1)\alpha^2\big(1-\alpha^{2N}\big)}{(1-\alpha)(1+\alpha)}\Big)\cdot\boldsymbol{I},$$

$$\mathbb{E}\Big[\sum_{i=0}^{N-1}\sum_{j=0}^{N-1}\alpha^{i+j+2}y_{N-i}y_{N-j}^2\boldsymbol{x}_{N-i}\Big] = \boldsymbol{0},$$

$$\mathbb{E}\Big[\sum_{i=0}^{N-1}\sum_{j=0}^{N-1}\alpha^{i+j+2}y_{N-i}^2 y_{N-j}^2\Big] = \frac{d^2\alpha^2\big(1-\alpha^N\big)^2}{(1-\alpha)^2} + \frac{(2d^2+6d)\alpha^2\big(1-\alpha^{2N}\big)}{(1-\alpha)(1+\alpha)},$$

$$\mathbb{E}\Big[\sum_{i=0}^{N-1}\alpha^{i+1}y_{N-i}\boldsymbol{x}_{N-i}\boldsymbol{w}^\top\Big] = \alpha\Big(\frac{1-\alpha^N}{1-\alpha}\Big)\cdot\boldsymbol{I},$$

$$\mathbb{E}\Big[\sum_{i=0}^{N-1}\alpha^{i+1}y_{N-i}^2\boldsymbol{w}\Big]=\boldsymbol{0},$$

$$\mathbb{E}\Big[\sum_{i=0}^{N-1}\sum_{j=0}^{N-1}\alpha^{i+j+2}\boldsymbol{x}_{N-i}\boldsymbol{x}_{N-i}^\top\boldsymbol{w}\boldsymbol{x}_{N-j}^\top\boldsymbol{w}\Big]=\boldsymbol{0},$$

$$\mathbb{E}\Big[\sum_{i=0}^{N-1}\sum_{j=0}^{N-1}\alpha^{i+j+2}y_{N-i}^2 y_{N-j}\Big]=\boldsymbol{0},$$

$$\mathbb{E}\Big[\sum_{i=0}^{N-1}\sum_{j=0}^{N-1}\alpha^{i+j+2}y_{N-i}y_{N-j}\Big]=\frac{d\alpha^2\big(1-\alpha^{2N}\big)}{(1-\alpha)(1+\alpha)},$$

$$\mathbb{E}\Big[\sum_{i=0}^{N-1}\alpha^{i+1}y_{N-i}\boldsymbol{w}\Big]=\boldsymbol{0}.$$

**Proof.**

$$\mathbb{E}\Big[\sum_{i=0}^{N-1}\sum_{j=0}^{N-1}\alpha^{i+j+2}y_{N-i}y_{N-j}\boldsymbol{x}_{N-i}\boldsymbol{x}_{N-j}^\top\Big]$$

$$=\mathbb{E}\Big[\sum_{i=0}^{N-1}\sum_{j=0}^{N-1}\alpha^{i+j+2}\boldsymbol{x}_{N-i}\underbrace{\boldsymbol{x}_{N-i}^\top\boldsymbol{w}}_{y_{N-i}}\underbrace{\boldsymbol{w}^\top\boldsymbol{x}_{N-j}}_{y_{N-j}}\boldsymbol{x}_{N-j}^\top\Big]$$

$$=\sum_{i\neq j}\alpha^{i+j+2}\underbrace{\mathbb{E}\big[\boldsymbol{x}_{N-i}\boldsymbol{x}_{N-i}^\top\big]}_{=\boldsymbol{I}}\underbrace{\mathbb{E}\big[\boldsymbol{w}\boldsymbol{w}^\top\big]}_{=\boldsymbol{I}}\underbrace{\mathbb{E}\big[\boldsymbol{x}_{N-j}\boldsymbol{x}_{N-j}^\top\big]}_{=\boldsymbol{I}}$$

$$+\sum_{i=j}\alpha^{i+j+2}\underbrace{\mathbb{E}\big[\boldsymbol{x}_{N-i}\boldsymbol{x}_{N-i}^\top\boldsymbol{w}\boldsymbol{w}^\top\boldsymbol{x}_{N-j}\boldsymbol{x}_{N-j}^\top\big]}_{=(d+2)\,\boldsymbol{I},\text{ by lemma }A.2}$$

$$=\Big(\Big(\sum_{i=0}^{N-1}\alpha^{i+1}\Big)^2-\Big(\sum_{i=0}^{N-1}\alpha^{2i+2}\Big)\Big)\boldsymbol{I}$$

$$+\Big(\sum_{i=0}^{N-1}\alpha^{2i+2}\Big)(d+2)\boldsymbol{I}$$

$$=\Big(\frac{\alpha^2\big(1-\alpha^N\big)^2}{(1-\alpha)^2}+\frac{(d+1)\alpha^2\big(1-\alpha^{2N}\big)}{(1-\alpha)(1+\alpha)}\Big)\cdot\boldsymbol{I}$$

Here, $\sum_{i\neq j}\alpha^{i+j+2}=\Big(\sum_{i=0}^{N-1}\alpha^{i+1}\Big)^2-\Big(\sum_{i=0}^{N-1}\alpha^{2i+2}\Big)$ for the third equality.

$$\mathbb{E}\Big[\sum_{i=0}^{N-1}\sum_{j=0}^{N-1}\alpha^{i+j+2}y_{N-i}y_{N-j}^2\boldsymbol{x}_{N-i}\Big]$$

$$=\mathbb{E}\Big[\sum_{i=0}^{N-1}\sum_{j=0}^{N-1}\alpha^{i+j+2}\boldsymbol{x}_{N-i}\underbrace{\boldsymbol{x}_{N-i}^\top\boldsymbol{w}}_{y_{N-i}}\underbrace{\boldsymbol{x}_{N-j}^\top\boldsymbol{w}}_{y_{N-j}}\underbrace{\boldsymbol{x}_{N-j}^\top\boldsymbol{w}}_{y_{N-j}}\Big]$$

$$=\boldsymbol{0}$$

Notice that $\boldsymbol{w}$ appears three (odd) times in the second equality, if we define a function $g(\boldsymbol{w})=\boldsymbol{x}_{N-i}\boldsymbol{x}_{N-i}^\top\boldsymbol{w}\boldsymbol{x}_{N-j}^\top\boldsymbol{w}\boldsymbol{x}_{N-j}^\top\boldsymbol{w}$, we can see that $g(-\boldsymbol{w})=-g(\boldsymbol{w})$, and further $\mathbb{E}_{\boldsymbol{w}}\big[g(\boldsymbol{w})\big]=\boldsymbol{0}$. Therefore, the above expectation equals to $\boldsymbol{0}$. We will use the similar property in some of the following equations.

$$\mathbb{E}\Big[\sum_{i=0}^{N-1}\sum_{j=0}^{N-1}\alpha^{i+j+2}y_{N-i}^2 y_{N-j}^2\Big]$$

$$=\sum_{i\neq j}\alpha^{i+j+2}\mathbb{E}\Big[y_{N-i}^2\Big]\mathbb{E}\Big[y_{N-j}^2\Big]+\sum_{i=j}\alpha^{i+j+2}\mathbb{E}\Big[y_{N-i}^4\Big]$$

$$=\Big(\Big(\sum_{i=0}^{N-1}\alpha^{i+1}\Big)^2-\Big(\sum_{i=0}^{N-1}\alpha^{2i+2}\Big)\Big)d^2+\Big(\sum_{i=0}^{N-1}\alpha^{2i+2}\Big)\cdot 3d(d+2)$$

$$=\frac{d^2\alpha^2\Big(1-\alpha^N\Big)^2}{(1-\alpha)^2}+\frac{(2d^2+6d)\alpha^2\Big(1-\alpha^{2N}\Big)}{(1-\alpha)(1+\alpha)}$$

where the second equality is by $\mathbb{E}\Big[y_{N-i}^2\Big]=\mathbb{E}\Big[y_{N-j}^2\Big]=d$ and $\mathbb{E}\Big[y_{N-i}^4\Big]=3d(d+2)$ (lemma A.2).

$$\mathbb{E}\Big[\sum_{i=0}^{N-1}\alpha^{i+1}y_{N-i}\boldsymbol{x}_{N-i}\boldsymbol{w}^\top\Big]$$

$$=\sum_{i=0}^{N-1}\alpha^{i+1}\mathbb{E}\Big[\boldsymbol{x}_{N-i}\underbrace{\boldsymbol{x}_{N-i}^\top\boldsymbol{w}}_{y_{N-i}}\boldsymbol{w}^\top\Big]$$

$$=\sum_{i=0}^{N-1}\alpha^{i+1}\underbrace{\mathbb{E}\Big[\boldsymbol{x}_{N-i}\boldsymbol{x}_{N-i}^\top\Big]}_{=\,\boldsymbol{I}}\underbrace{\mathbb{E}\Big[\boldsymbol{w}\boldsymbol{w}^\top\Big]}_{=\,\boldsymbol{I}}$$

$$=\sum_{i=0}^{N-1}\alpha^{i+1}\cdot\boldsymbol{I}$$

$$=\alpha\Big(\frac{1-\alpha^N}{1-\alpha}\Big)\cdot\boldsymbol{I}$$

$$\mathbb{E}\Big[\sum_{i=0}^{N-1}\alpha^{i+1}y_{N-i}^2\boldsymbol{w}\Big]$$

$$=\mathbb{E}\Big[\sum_{i=0}^{N-1}\alpha^{i+1}\boldsymbol{w}\underbrace{\boldsymbol{x}_{N-i}^\top\boldsymbol{w}}_{y_{N-i}}\underbrace{\boldsymbol{x}_{N-i}^\top\boldsymbol{w}}_{y_{N-i}}\Big]$$

$$=\boldsymbol{0}$$

Notice that $\boldsymbol{w}$ appears three (odd) times in the second equality, thus this expectation equals to $\boldsymbol{0}$.

$$\mathbb{E}\Big[\sum_{i=0}^{N-1}\sum_{j=0}^{N-1}\alpha^{i+j+2}\boldsymbol{x}_{N-i}\boldsymbol{x}_{N-i}^\top\boldsymbol{w}\boldsymbol{x}_{N-j}^\top\boldsymbol{w}\Big]$$

$$=\sum_{i=j}\alpha^{i+j+2}\mathbb{E}\Big[\boldsymbol{x}_{N-i}\boldsymbol{x}_{N-i}^\top\boldsymbol{w}\boldsymbol{x}_{N-i}^\top\boldsymbol{w}\Big]$$

$$+\sum_{i\neq j}\alpha^{i+j+2}\mathbb{E}\Big[\boldsymbol{x}_{N-i}\boldsymbol{x}_{N-i}^\top\Big]\mathbb{E}\Big[\boldsymbol{w}\boldsymbol{x}_{N-j}^\top\boldsymbol{w}\Big]$$

$$=\boldsymbol{0}+\boldsymbol{0}=\boldsymbol{0}$$

Notice that $\boldsymbol{x}_{N-i}$ appears three (odd) times in $\mathbb{E}\Big[\boldsymbol{x}_{N-i}\boldsymbol{x}_{N-i}^\top\boldsymbol{w}\boldsymbol{x}_{N-i}^\top\boldsymbol{w}\Big]$, and $\boldsymbol{x}_{N-j}^\top$ appears once (odd) in $\mathbb{E}\Big[\boldsymbol{w}\boldsymbol{x}_{N-j}^\top\boldsymbol{w}\Big]$, thus this expectation equals to $\boldsymbol{0}$.

$$\mathbb{E}\Big[\sum_{i=0}^{N-1}\sum_{j=0}^{N-1}\alpha^{i+j+2}y_{N-i}^2 y_{N-j}\Big]$$

$$=\sum_{i=0}^{N-1}\sum_{j=0}^{N-1}\alpha^{i+j+2}\mathbb{E}\Big[\underbrace{\boldsymbol{x}_{N-i}^\top\boldsymbol{w}}_{y_{N-i}}\underbrace{\boldsymbol{x}_{N-i}^\top\boldsymbol{w}}_{y_{N-i}}\underbrace{\boldsymbol{x}_{N-j}^\top\boldsymbol{w}}_{y_{N-j}}\Big]$$

$$=\boldsymbol{0}$$

Notice that $\boldsymbol{w}$ appears three (odd) times in the second equality, thus this expectation equals to $\boldsymbol{0}$.

$$\mathbb{E}\Big[\sum_{i=0}^{N-1}\sum_{j=0}^{N-1}\alpha^{i+j+2}y_{N-i}y_{N-j}\Big]$$

$$=\sum_{i\neq j}\alpha^{i+j+2}\mathbb{E}\Big[y_{N-i}y_{N-j}\Big]+\sum_{i=j}\alpha^{i+j+2}\mathbb{E}\Big[y_{N-i}^2\Big]$$

$$=\sum_{i\neq j}\alpha^{i+j+2}\mathbb{E}\Big[\underbrace{\boldsymbol{x}_{N-i}^\top\boldsymbol{w}}_{y_{N-i}}\underbrace{\boldsymbol{x}_{N-j}^\top\boldsymbol{w}}_{y_{N-j}}\Big]+\sum_{i=0}^{N-1}\alpha^{2i+2}\mathbb{E}\Big[y_{N-i}^2\Big]$$

$$=\frac{d\alpha^2\Big(1-\alpha^{2N}\Big)}{(1-\alpha)(1+\alpha)}$$

Notice that $\boldsymbol{x}_{N-i}^\top$ appears once (odd) in $\mathbb{E}\Big[\boldsymbol{x}_{N-i}^\top\boldsymbol{w}\boldsymbol{x}_{N-j}^\top\boldsymbol{w}\Big]$ where $i\neq j$, thus $\sum_{i\neq j}\alpha^{i+j+2}\mathbb{E}\Big[y_{N-i}y_{N-j}\Big]=0$. Moreover, $\mathbb{E}\Big[y_{N-i}^2\Big]=d$ by lemma A.2.

$$\mathbb{E}\Big[\sum_{i=0}^{N-1}\alpha^{i+1}y_{N-i}\boldsymbol{w}\Big]$$

$$=\sum_{i=0}^{N-1}\alpha^{i+1}\mathbb{E}\Big[\boldsymbol{w}\underbrace{\boldsymbol{w}^\top\boldsymbol{x}_{N-i}}_{y_{N-i}}\Big]$$

$$=\sum_{i=0}^{N-1}\alpha^{i+1}\mathbb{E}\Big[\boldsymbol{w}\boldsymbol{w}^\top\Big]\mathbb{E}\Big[\boldsymbol{x}_{N-i}\Big]$$

$$=\boldsymbol{0}$$

## C.3 Proof of Lemma A.4

**Lemma C.3 (restatement of lemma A.4)** *The gradient of trainable parameters* $\boldsymbol{\theta}' = \{\boldsymbol{B},\boldsymbol{C},\boldsymbol{b},\boldsymbol{b}_B,\boldsymbol{b}_C\}$ *with respect to loss (Eq. (23)) are as follows:*

$$\nabla_{\boldsymbol{b}_B}\mathcal{L}(\boldsymbol{\theta})=\boldsymbol{0},$$

$$\nabla_{\boldsymbol{b}_C}\mathcal{L}(\boldsymbol{\theta})=\boldsymbol{0},$$

$$\nabla_{\boldsymbol{B}}\mathcal{L}(\boldsymbol{\theta})=\underbrace{\Big(\alpha^2\big(1-\alpha^N\big)^2+\frac{(d+1)\alpha^2(1-\alpha)\big(1-\alpha^{2N}\big)}{(1+\alpha)}\Big)}_{:=\beta_1}\boldsymbol{C}\boldsymbol{C}^\top\boldsymbol{B}-\underbrace{\alpha\big(1-\alpha^N\big)}_{:=\beta_3}\boldsymbol{C},$$

$$\nabla_{\boldsymbol{b}}\mathcal{L}(\boldsymbol{\theta})=\underbrace{\Big(d^2\alpha^2\big(1-\alpha^N\big)^2+\frac{(2d^2+6d)\alpha^2(1-\alpha)\big(1-\alpha^{2N}\big)}{(1+\alpha)}\Big)}_{:=\beta_2}\boldsymbol{C}\boldsymbol{C}^\top\boldsymbol{b},$$

$$\nabla_C \mathcal{L}(\boldsymbol{\theta}) = \underbrace{\left(\alpha^2\left(1-\alpha^N\right)^2 + \frac{(d+1)\alpha^2(1-\alpha)\left(1-\alpha^{2N}\right)}{(1+\alpha)}\right)}_{:=\beta_1} \boldsymbol{B}\boldsymbol{B}^\top \boldsymbol{C}$$

$$+ \underbrace{\left(d^2\alpha^2\left(1-\alpha^N\right)^2 + \frac{(2d^2+6d)\alpha^2(1-\alpha)\left(1-\alpha^{2N}\right)}{(1+\alpha)}\right)}_{:=\beta_2} \boldsymbol{b}\boldsymbol{b}^\top \boldsymbol{C}$$

$$- \underbrace{\alpha\left(1-\alpha^N\right)}_{:=\beta_3} \boldsymbol{B}.$$

Proof of lemma C.3. Recalling the loss:

$$\mathcal{L}(\boldsymbol{\theta}) = \frac{1}{2}\mathbb{E}\left[\left((1-\alpha)(\boldsymbol{C}\boldsymbol{x}_q + \boldsymbol{b}_C)^\top \sum_{i=0}^{N-1} \alpha^{i+1} y_{N-i}(\boldsymbol{B}\boldsymbol{x}_{N-i} + y_{N-i}\boldsymbol{b} + \boldsymbol{b}_B) - \boldsymbol{w}^\top \boldsymbol{x}_q\right)^2\right]$$

We will compute the gradient of $\{\boldsymbol{B}, \boldsymbol{C}, \boldsymbol{b}, \boldsymbol{b}_B, \boldsymbol{b}_C\}$ with respect to $\mathcal{L}(\boldsymbol{\theta})$. Some expectation calculation are detailed in Section C.3.1.

$$\nabla_{\boldsymbol{b}_C}\mathcal{L}(\boldsymbol{\theta}) = \mathbb{E}\Bigg[(1-\alpha)\underbrace{\left((1-\alpha)(\boldsymbol{C}\boldsymbol{x}_q + \boldsymbol{b}_C)^\top \sum_{i=0}^{N-1} \alpha^{i+1} y_{N-i}(\boldsymbol{B}\boldsymbol{x}_{N-i} + y_{N-i}\boldsymbol{b} + \boldsymbol{b}_B) - \boldsymbol{w}^\top \boldsymbol{x}_q\right)}_{:=v}$$

$$\cdot \underbrace{\sum_{i=0}^{N-1} \alpha^{i+1} y_{N-i}(\boldsymbol{B}\boldsymbol{x}_{N-i} + y_{N-i}\boldsymbol{b} + \boldsymbol{b}_B)}_{:=v}\Bigg]$$

$$= (1-\alpha)^2 \mathbb{E}\left[\boldsymbol{v}\boldsymbol{v}^\top(\boldsymbol{C}\boldsymbol{x}_q + \boldsymbol{b}_C)\right] - (1-\alpha)\mathbb{E}\left[\boldsymbol{v}\boldsymbol{w}^\top \boldsymbol{x}_q\right]$$

$$= (1-\alpha)^2 \mathbb{E}\left[\boldsymbol{v}\boldsymbol{v}^\top \boldsymbol{C}\right]\mathbb{E}\left[\boldsymbol{x}_q\right] + (1-\alpha)^2 \mathbb{E}\left[\boldsymbol{v}\boldsymbol{v}^\top\right]\boldsymbol{b}_C - (1-\alpha)\mathbb{E}\left[\boldsymbol{v}\boldsymbol{w}^\top\right]\mathbb{E}\left[\boldsymbol{x}_q\right]$$

It is clear that $\mathbb{E}\left[\boldsymbol{x}_q\right] = \boldsymbol{0}$. Thus, if $\boldsymbol{b}_C = \boldsymbol{0}$, then $\nabla_{\boldsymbol{b}_C}\mathcal{L}(\boldsymbol{\theta}) = \boldsymbol{0}$. Notice that we assume $\boldsymbol{b}_C(0) = \boldsymbol{0}$ at initialization, so by induction, $\boldsymbol{b}_C(t) = \boldsymbol{0}$ and $\nabla_{\boldsymbol{b}_C}\mathcal{L}(\boldsymbol{\theta}(t)) = \boldsymbol{0}$ for $t \geq 0$. We will consider $\boldsymbol{b}_C = \boldsymbol{0}$ when computing other gradients.

$$\nabla_{\boldsymbol{b}_B}\mathcal{L}(\boldsymbol{\theta}) = \mathbb{E}\Bigg[(1-\alpha)\left((1-\alpha)(\boldsymbol{C}\boldsymbol{x}_q + \boldsymbol{b}_C)^\top \sum_{i=0}^{N-1} \alpha^{i+1} y_{N-i}(\boldsymbol{B}\boldsymbol{x}_{N-i} + y_{N-i}\boldsymbol{b} + \boldsymbol{b}_B) - \boldsymbol{w}^\top \boldsymbol{x}_q\right)$$

$$\cdot (\boldsymbol{C}\boldsymbol{x}_q + \boldsymbol{b}_C) \cdot \sum_{i=0}^{N-1} \alpha^{i+1} y_{N-i}\Bigg]$$

$$= \mathbb{E}\Bigg[(1-\alpha)\left((1-\alpha)(\boldsymbol{C}\boldsymbol{x}_q)^\top \sum_{i=0}^{N-1} \alpha^{i+1} y_{N-i}(\boldsymbol{B}\boldsymbol{x}_{N-i} + y_{N-i}\boldsymbol{b} + \boldsymbol{b}_B) - \boldsymbol{w}^\top \boldsymbol{x}_q\right)$$

$$\cdot \boldsymbol{C}\boldsymbol{x}_q \cdot \sum_{i=0}^{N-1} \alpha^{i+1} y_{N-i}\Bigg]$$

$$= (1-\alpha)^2 \mathbb{E}\left[\boldsymbol{x}_q^\top \boldsymbol{C}^\top \left(\sum_{i=0}^{N-1} \alpha^{i+1} y_{N-i}(\boldsymbol{B}\boldsymbol{x}_{N-i} + y_{N-i}\boldsymbol{b} + \boldsymbol{b}_B)\right) \cdot \boldsymbol{C}\boldsymbol{x}_q \cdot \sum_{i=0}^{N-1} \alpha^{i+1} y_{N-i}\right]$$

$$- (1-\alpha)\mathbb{E}\left[\boldsymbol{w}^\top \boldsymbol{x}_q \cdot \boldsymbol{C}\boldsymbol{x}_q \cdot \sum_{i=0}^{N-1} \alpha^{i+1} y_{N-i}\right]$$

$$= \frac{d\alpha^2(1-\alpha)\left(1-\alpha^{2N}\right)}{(1+\alpha)}\boldsymbol{C}\boldsymbol{C}^\top \boldsymbol{b}_B$$

The last equality follows from lemma C.4 where we have: $\mathbb{E}\Big[\boldsymbol{x}_q^\top \boldsymbol{C}^\top \Big( \sum_{i=0}^{N-1} \alpha^{i+1} y_{N-i}(\boldsymbol{B}\boldsymbol{x}_{N-i} +$

$y_{N-i}\boldsymbol{b} + \boldsymbol{b}_B)\Big) \cdot \boldsymbol{C}\boldsymbol{x}_q \cdot \sum_{i=0}^{N-1} \alpha^{i+1} y_{N-i}\Big] = \frac{d\alpha^2\left(1-\alpha^{2N}\right)}{(1-\alpha)(1+\alpha)}\boldsymbol{C}\boldsymbol{C}^\top \boldsymbol{b}_B$, and $\mathbb{E}\Big[\boldsymbol{w}^\top \boldsymbol{x}_q \cdot \boldsymbol{C}\boldsymbol{x}_q \cdot$

$\sum_{i=0}^{N-1} \alpha^{i+1} y_{N-i}\Big] = \boldsymbol{0}$. Similar to $\boldsymbol{b}_C$, notice that $\boldsymbol{b}_B$ is initialized as $\boldsymbol{0}$, thus by induction, $\boldsymbol{b}_B(t) = \boldsymbol{0}$ and $\nabla_{\boldsymbol{b}_B}\mathcal{L}(\boldsymbol{\theta}(t)) = \boldsymbol{0}$ for $t \geq 0$. We will consider $\boldsymbol{b}_B = \boldsymbol{b}_C = \boldsymbol{0}$ when computing other gradients.

$$\nabla_{\boldsymbol{C}}\mathcal{L}(\boldsymbol{\theta}) = \mathbb{E}\Big[(1-\alpha)\Big((1-\alpha)(\boldsymbol{C}\boldsymbol{x}_q + \boldsymbol{b}_C)^\top \sum_{i=0}^{N-1} \alpha^{i+1} y_{N-i}(\boldsymbol{B}\boldsymbol{x}_{N-i} + y_{N-i}\boldsymbol{b} + \boldsymbol{b}_B) - \boldsymbol{w}^\top \boldsymbol{x}_q\Big)$$

$$\cdot \sum_{i=0}^{N-1} \alpha^{i+1} y_{N-i}(\boldsymbol{B}\boldsymbol{x}_{N-i} + y_{N-i}\boldsymbol{b} + \boldsymbol{b}_B)\boldsymbol{x}_q^\top\Big]$$

$$= (1-\alpha)^2 \mathbb{E}\Big[\boldsymbol{x}_q^\top \boldsymbol{C}^\top \Big( \sum_{i=0}^{N-1} \alpha^{i+1} y_{N-i}(\boldsymbol{B}\boldsymbol{x}_{N-i} + y_{N-i}\boldsymbol{b})\Big)$$

$$\cdot \Big( \sum_{i=0}^{N-1} \alpha^{i+1} y_{N-i}(\boldsymbol{B}\boldsymbol{x}_{N-i} + y_{N-i}\boldsymbol{b})\Big)\boldsymbol{x}_q^\top\Big]$$

$$- (1-\alpha)\mathbb{E}\Big[\boldsymbol{w}^\top \boldsymbol{x}_q \cdot \sum_{i=0}^{N-1} \alpha^{i+1} y_{N-i}(\boldsymbol{B}\boldsymbol{x}_{N-i} + y_{N-i}\boldsymbol{b})\boldsymbol{x}_q^\top\Big]$$

$$= \underbrace{\Big(\alpha^2\big(1-\alpha^N\big)^2 + \frac{(d+1)\alpha^2(1-\alpha)\big(1-\alpha^{2N}\big)}{(1+\alpha)}\Big)}_{:=\beta_1} \boldsymbol{B}\boldsymbol{B}^\top \boldsymbol{C}$$

$$+ \underbrace{\Big(d^2\alpha^2\big(1-\alpha^N\big)^2 + \frac{(2d^2 + 6d)\alpha^2(1-\alpha)\big(1-\alpha^{2N}\big)}{(1+\alpha)}\Big)}_{:=\beta_2} \boldsymbol{b}\boldsymbol{b}^\top \boldsymbol{C}$$

$$- \underbrace{\alpha\big(1-\alpha^N\big)}_{:=\beta_3} \boldsymbol{B}$$

The last equality follows from lemma C.4, where we have:

$$\mathbb{E}\Big[\boldsymbol{x}_q^\top \boldsymbol{C}^\top \Big( \sum_{i=0}^{N-1} \alpha^{i+1} y_{N-i}(\boldsymbol{B}\boldsymbol{x}_{N-i} + y_{N-i}\boldsymbol{b})\Big) \cdot \Big( \sum_{i=0}^{N-1} \alpha^{i+1} y_{N-i}(\boldsymbol{B}\boldsymbol{x}_{N-i} + y_{N-i}\boldsymbol{b})\Big)\boldsymbol{x}_q^\top\Big]$$

$$= \Big(\frac{\alpha^2\big(1-\alpha^N\big)^2}{(1-\alpha)^2} + \frac{(d+1)\alpha^2\big(1-\alpha^{2N}\big)}{(1-\alpha)(1+\alpha)}\Big)\boldsymbol{B}\boldsymbol{B}^\top \boldsymbol{C}$$

$$+ \Big(\frac{d^2\alpha^2\big(1-\alpha^N\big)^2}{(1-\alpha)^2} + \frac{(2d^2 + 6d)\alpha^2\big(1-\alpha^{2N}\big)}{(1-\alpha)(1+\alpha)}\Big)\boldsymbol{b}\boldsymbol{b}^\top \boldsymbol{C}$$

and $\mathbb{E}\Big[\boldsymbol{w}^\top \boldsymbol{x}_q \cdot \sum_{i=0}^{N-1} \alpha^{i+1} y_{N-i}(\boldsymbol{B}\boldsymbol{x}_{N-i} + y_{N-i}\boldsymbol{b})\boldsymbol{x}_q^\top\Big] = \alpha\Big(\frac{1-\alpha^N}{1-\alpha}\Big)\boldsymbol{B}$

$$\nabla_{\boldsymbol{B}}\mathcal{L}(\boldsymbol{\theta}) = \mathbb{E}\Big[(1-\alpha)\Big((1-\alpha)(\boldsymbol{C}\boldsymbol{x}_q + \boldsymbol{b}_C)^\top \sum_{i=0}^{N-1} \alpha^{i+1} y_{N-i}(\boldsymbol{B}\boldsymbol{x}_{N-i} + y_{N-i}\boldsymbol{b} + \boldsymbol{b}_B) - \boldsymbol{w}^\top \boldsymbol{x}_q\Big)$$

$$\cdot (\boldsymbol{C}\boldsymbol{x}_q + \boldsymbol{b}_C) \sum_{i=0}^{N-1} \alpha^{i+1} y_{N-i}\boldsymbol{x}_{N-i}^\top\Big]$$

$$= (1-\alpha)^2 \mathbb{E}\Big[\boldsymbol{x}_q^\top \boldsymbol{C}^\top \sum_{i=0}^{N-1} \alpha^{i+1} y_{N-i}(\boldsymbol{B}\boldsymbol{x}_{N-i} + y_{N-i}\boldsymbol{b}) \cdot \boldsymbol{C}\boldsymbol{x}_q \sum_{i=0}^{N-1} \alpha^{i+1} y_{N-i}\boldsymbol{x}_{N-i}^\top\Big]$$

$$- (1-\alpha)\mathbb{E}\Big[\boldsymbol{w}^\top \boldsymbol{x}_q \cdot \boldsymbol{C}\boldsymbol{x}_q \sum_{i=0}^{N-1} \alpha^{i+1} y_{N-i}\boldsymbol{x}_{N-i}^\top\Big]$$

$$= \underbrace{\Big(\alpha^2\big(1-\alpha^N\big)^2 + \frac{(d+1)\alpha^2(1-\alpha)\big(1-\alpha^{2N}\big)}{(1+\alpha)}\Big)}_{:=\beta_1} \boldsymbol{C}\boldsymbol{C}^\top \boldsymbol{B}$$

$$- \underbrace{\alpha\big(1-\alpha^N\big)}_{:=\beta_3} \boldsymbol{C}$$

The last equality follows from lemma C.4, where we have:

$$\mathbb{E}\Big[\boldsymbol{x}_q^\top \boldsymbol{C}^\top \sum_{i=0}^{N-1} \alpha^{i+1} y_{N-i}(\boldsymbol{B}\boldsymbol{x}_{N-i} + y_{N-i}\boldsymbol{b}) \cdot \boldsymbol{C}\boldsymbol{x}_q \sum_{i=0}^{N-1} \alpha^{i+1} y_{N-i}\boldsymbol{x}_{N-i}^\top\Big]$$

$$= \Big(\frac{\alpha^2\big(1-\alpha^N\big)^2}{(1-\alpha)^2} + \frac{(d+1)\alpha^2\big(1-\alpha^{2N}\big)}{(1-\alpha)(1+\alpha)}\Big)\boldsymbol{C}\boldsymbol{C}^\top \boldsymbol{B}$$

and $\mathbb{E}\Big[\boldsymbol{w}^\top \boldsymbol{x}_q \cdot \boldsymbol{C}\boldsymbol{x}_q \sum_{i=0}^{N-1} \alpha^{i+1} y_{N-i}\boldsymbol{x}_{N-i}^\top\Big] = \alpha\Big(\frac{1-\alpha^N}{1-\alpha}\Big)\boldsymbol{C}$.

$$\nabla_{\boldsymbol{b}}\mathcal{L}(\boldsymbol{\theta}) = \mathbb{E}\Big[(1-\alpha)\Big((1-\alpha)(\boldsymbol{C}\boldsymbol{x}_q + \boldsymbol{b}_C)^\top \sum_{i=0}^{N-1} \alpha^{i+1} y_{N-i}(\boldsymbol{B}\boldsymbol{x}_{N-i} + y_{N-i}\boldsymbol{b} + \boldsymbol{b}_B) - \boldsymbol{w}^\top \boldsymbol{x}_q\Big)$$

$$\cdot (\boldsymbol{C}\boldsymbol{x}_q + \boldsymbol{b}_C) \cdot \sum_{i=0}^{N-1} \alpha^{i+1} y_{N-i}^2\Big]$$

$$= \mathbb{E}\Big[(1-\alpha)\Big((1-\alpha)(\boldsymbol{C}\boldsymbol{x}_q)^\top \sum_{i=0}^{N-1} \alpha^{i+1} y_{N-i}(\boldsymbol{B}\boldsymbol{x}_{N-i} + y_{N-i}\boldsymbol{b}) - \boldsymbol{w}^\top \boldsymbol{x}_q\Big)$$

$$\cdot \boldsymbol{C}\boldsymbol{x}_q \cdot \sum_{i=0}^{N-1} \alpha^{i+1} y_{N-i}^2\Big]$$

$$= (1-\alpha)^2\mathbb{E}\Big[\Big(\boldsymbol{x}_q^\top \boldsymbol{C}^\top \sum_{i=0}^{N-1} \alpha^{i+1} y_{N-i}(\boldsymbol{B}\boldsymbol{x}_{N-i} + y_{N-i}\boldsymbol{b})\Big) \cdot \boldsymbol{C}\boldsymbol{x}_q \cdot \sum_{i=0}^{N-1} \alpha^{i+1} y_{N-i}^2\Big]$$

$$- (1-\alpha)\mathbb{E}\Big[\boldsymbol{w}^\top \boldsymbol{x}_q \cdot \boldsymbol{C}\boldsymbol{x}_q \cdot \sum_{i=0}^{N-1} \alpha^{i+1} y_{N-i}^2\Big]$$

$$= \underbrace{\Big(d^2\alpha^2\big(1-\alpha^N\big)^2 + \frac{(2d^2 + 6d)\alpha^2(1-\alpha)\big(1-\alpha^{2N}\big)}{(1+\alpha)}\Big)}_{:=\beta_2} \boldsymbol{C}\boldsymbol{C}^\top \boldsymbol{b}$$

The last equality follows from lemma C.4, where we have:

$$\mathbb{E}\Big[\Big(\boldsymbol{x}_q^\top \boldsymbol{C}^\top \sum_{i=0}^{N-1} \alpha^{i+1} y_{N-i}(\boldsymbol{B}\boldsymbol{x}_{N-i} + y_{N-i}\boldsymbol{b})\Big) \cdot \boldsymbol{C}\boldsymbol{x}_q \cdot \sum_{i=0}^{N-1} \alpha^{i+1} y_{N-i}^2\Big]$$

$$= \Big(\frac{d^2\alpha^2\big(1-\alpha^N\big)^2}{(1-\alpha)^2} + \frac{(2d^2 + 6d)\alpha^2\big(1-\alpha^{2N}\big)}{(1-\alpha)(1+\alpha)}\Big)\boldsymbol{C}\boldsymbol{C}^\top \boldsymbol{b}$$

and $\mathbb{E}\Big[\boldsymbol{w}^\top \boldsymbol{x}_q \cdot \boldsymbol{C}\boldsymbol{x}_q \cdot \sum_{i=0}^{N-1} \alpha^{i+1} y_{N-i}^2\Big] = \boldsymbol{0}$.

### C.3.1 Auxiliary Lemma for Lemma A.4

**Lemma C.4** *If vectors $\boldsymbol{x}_i$, $\boldsymbol{x}_q$ and $\boldsymbol{w}$ are iid generated from $\mathcal{N}(0, \boldsymbol{I}_d)$, $y = \boldsymbol{x}_i^\top \boldsymbol{w}$ we have the following expectations:*

$$\mathbb{E}\Big[\boldsymbol{x}_q^\top \boldsymbol{C}^\top \Big(\sum_{i=0}^{N-1} \alpha^{i+1} y_{N-i}(\boldsymbol{B}\boldsymbol{x}_{N-i} + y_{N-i}\boldsymbol{b})\Big) \cdot \Big(\sum_{i=0}^{N-1} \alpha^{i+1} y_{N-i}(\boldsymbol{B}\boldsymbol{x}_{N-i} + y_{N-i}\boldsymbol{b})\Big)\boldsymbol{x}_q^\top\Big]$$

$$= \Big(\frac{\alpha^2(1-\alpha^N)^2}{(1-\alpha)^2} + \frac{(d+1)\alpha^2(1-\alpha^{2N})}{(1-\alpha)(1+\alpha)}\Big)\boldsymbol{B}\boldsymbol{B}^\top \boldsymbol{C}$$

$$+ \Big(\frac{d^2\alpha^2(1-\alpha^N)^2}{(1-\alpha)^2} + \frac{(2d^2+6d)\alpha^2(1-\alpha^{2N})}{(1-\alpha)(1+\alpha)}\Big)\boldsymbol{b}\boldsymbol{b}^\top \boldsymbol{C}$$

$$\mathbb{E}\Big[\boldsymbol{w}^\top \boldsymbol{x}_q \cdot \sum_{i=0}^{N-1} \alpha^{i+1} y_{N-i}(\boldsymbol{B}\boldsymbol{x}_{N-i} + y_{N-i}\boldsymbol{b})\boldsymbol{x}_q^\top\Big] = \alpha\Big(\frac{1-\alpha^N}{1-\alpha}\Big)\boldsymbol{B}$$

$$\mathbb{E}\Big[\boldsymbol{x}_q^\top \boldsymbol{C}^\top \sum_{i=0}^{N-1} \alpha^{i+1} y_{N-i}(\boldsymbol{B}\boldsymbol{x}_{N-i} + y_{N-i}\boldsymbol{b}) \cdot \boldsymbol{C}\boldsymbol{x}_q \sum_{i=0}^{N-1} \alpha^{i+1} y_{N-i}\boldsymbol{x}_{N-i}^\top\Big]$$

$$= \Big(\frac{\alpha^2(1-\alpha^N)^2}{(1-\alpha)^2} + \frac{(d+1)\alpha^2(1-\alpha^{2N})}{(1-\alpha)(1+\alpha)}\Big)\boldsymbol{C}\boldsymbol{C}^\top \boldsymbol{B}$$

$$\mathbb{E}\Big[\boldsymbol{w}^\top \boldsymbol{x}_q \cdot \boldsymbol{C}\boldsymbol{x}_q \sum_{i=0}^{N-1} \alpha^{i+1} y_{N-i}\boldsymbol{x}_{N-i}^\top\Big] = \alpha\Big(\frac{1-\alpha^N}{1-\alpha}\Big)\boldsymbol{C}$$

$$\mathbb{E}\Big[\Big(\boldsymbol{x}_q^\top \boldsymbol{C}^\top \sum_{i=0}^{N-1} \alpha^{i+1} y_{N-i}(\boldsymbol{B}\boldsymbol{x}_{N-i} + y_{N-i}\boldsymbol{b})\Big) \cdot \boldsymbol{C}\boldsymbol{x}_q \cdot \sum_{i=0}^{N-1} \alpha^{i+1} y_{N-i}^2\Big]$$

$$= \Big(\frac{d^2\alpha^2(1-\alpha^N)^2}{(1-\alpha)^2} + \frac{(2d^2+6d)\alpha^2(1-\alpha^{2N})}{(1-\alpha)(1+\alpha)}\Big)\boldsymbol{C}\boldsymbol{C}^\top \boldsymbol{b}$$

$$\mathbb{E}\Big[\boldsymbol{w}^\top \boldsymbol{x}_q \cdot \boldsymbol{C}\boldsymbol{x}_q \cdot \sum_{i=0}^{N-1} \alpha^{i+1} y_{N-i}^2\Big] = \boldsymbol{0}$$

$$\mathbb{E}\Big[\boldsymbol{x}_q^\top \boldsymbol{C}^\top \Big(\sum_{i=0}^{N-1} \alpha^{i+1} y_{N-i}(\boldsymbol{B}\boldsymbol{x}_{N-i} + y_{N-i}\boldsymbol{b} + \boldsymbol{b}_B)\Big) \cdot \boldsymbol{C}\boldsymbol{x}_q \cdot \sum_{i=0}^{N-1} \alpha^{i+1} y_{N-i}\Big]$$

$$= \frac{d\alpha^2(1-\alpha^{2N})}{(1-\alpha)(1+\alpha)}\boldsymbol{C}\boldsymbol{C}^\top \boldsymbol{b}_B$$

$$\mathbb{E}\Big[\boldsymbol{w}^\top \boldsymbol{x}_q \cdot \boldsymbol{C}\boldsymbol{x}_q \cdot \sum_{i=0}^{N-1} \alpha^{i+1} y_{N-i}\Big] = \boldsymbol{0}$$

Proof of lemma C.4. We will use the results of lemma A.3 to prove the above equation.

$$\mathbb{E}\Big[\boldsymbol{x}_q^\top \boldsymbol{C}^\top \underbrace{\Big(\sum_{i=0}^{N-1}\alpha^{i+1}y_{N-i}(\boldsymbol{B}\boldsymbol{x}_{N-i}+y_{N-i}\boldsymbol{b})\Big)}_{\spadesuit}$$

$$\cdot\underbrace{\Big(\sum_{i=0}^{N-1}\alpha^{i+1}y_{N-i}(\boldsymbol{B}\boldsymbol{x}_{N-i}+y_{N-i}\boldsymbol{b})\Big)\boldsymbol{x}_q^\top\Big]}$$

$$=\mathbb{E}\Big[\sum_{i=0}^{N-1}\alpha^{i+1}y_{N-i}\big(\boldsymbol{B}\boldsymbol{x}_{N-i}+y_{N-i}\boldsymbol{b}\big)$$

$$\underbrace{\Big(\boldsymbol{x}_q^\top \boldsymbol{C}^\top \sum_{i=0}^{N-1}\alpha^{i+1}y_{N-i}(\boldsymbol{B}\boldsymbol{x}_{N-i}+y_{N-i}\boldsymbol{b})\Big)^\top \boldsymbol{x}_q^\top\Big]}_{\spadesuit}$$

$$=\mathbb{E}\Big[\sum_{i=0}^{N-1}\sum_{j=0}^{N-1}\alpha^{i+j+2}\big(\big(\boldsymbol{B}y_{N-i}\boldsymbol{x}_{N-i}+y_{N-i}^2\boldsymbol{b}\big)$$

$$\big(y_{N-j}\boldsymbol{x}_{N-j}^\top \boldsymbol{B}^\top + y_{N-j}^2\boldsymbol{b}^\top\big)\big)\Big]\boldsymbol{C}\mathbb{E}\Big[\boldsymbol{x}_q\boldsymbol{x}_q^\top\Big]$$

$$=\boldsymbol{B}\mathbb{E}\Big[\sum_{i=0}^{N-1}\sum_{j=0}^{N-1}\alpha^{i+j+2}y_{N-i}y_{N-j}\boldsymbol{x}_{N-i}\boldsymbol{x}_{N-j}^\top\Big]\boldsymbol{B}^\top \boldsymbol{C}$$

$$+\boldsymbol{B}\mathbb{E}\Big[\sum_{i=0}^{N-1}\sum_{j=0}^{N-1}\alpha^{i+j+2}y_{N-i}y_{N-j}^2\boldsymbol{x}_{N-i}\Big]\boldsymbol{b}^\top \boldsymbol{C}$$

$$+\boldsymbol{b}\mathbb{E}\Big[\sum_{i=0}^{N-1}\sum_{j=0}^{N-1}\alpha^{i+j+2}y_{N-j}y_{N-i}^2\boldsymbol{x}_{N-j}\Big]\boldsymbol{B}^\top \boldsymbol{C}$$

$$+\mathbb{E}\Big[\sum_{i=0}^{N-1}\sum_{j=0}^{N-1}\alpha^{i+j+2}y_{N-i}^2 y_{N-j}^2\Big]\boldsymbol{b}\boldsymbol{b}^\top \boldsymbol{C}$$

$$=\Big(\frac{\alpha^2\big(1-\alpha^N\big)^2}{(1-\alpha)^2}+\frac{(d+1)\alpha^2\big(1-\alpha^{2N}\big)}{(1-\alpha)(1+\alpha)}\Big)\boldsymbol{B}\boldsymbol{B}^\top \boldsymbol{C}$$

$$+\Big(\frac{d^2\alpha^2\big(1-\alpha^N\big)^2}{(1-\alpha)^2}+\frac{(2d^2+6d)\alpha^2\big(1-\alpha^{2N}\big)}{(1-\alpha)(1+\alpha)}\Big)\boldsymbol{b}\boldsymbol{b}^\top \boldsymbol{C}$$

The last equality follows from lemma A.3, where we have:
$\mathbb{E}\Big[\sum_{i=0}^{N-1}\sum_{j=0}^{N-1}\alpha^{i+j+2}y_{N-i}y_{N-j}\boldsymbol{x}_{N-i}\boldsymbol{x}_{N-j}^\top\Big] = \Big(\frac{\alpha^2\big(1-\alpha^N\big)^2}{(1-\alpha)^2}+\frac{(d+1)\alpha^2\big(1-\alpha^{2N}\big)}{(1-\alpha)(1+\alpha)}\Big)\boldsymbol{I}$,
$\mathbb{E}\Big[\sum_{i=0}^{N-1}\sum_{j=0}^{N-1}\alpha^{i+j+2}y_{N-i}y_{N-j}^2\boldsymbol{x}_{N-i}\Big] = \boldsymbol{0}$ and $\mathbb{E}\Big[\sum_{i=0}^{N-1}\sum_{j=0}^{N-1}\alpha^{i+j+2}y_{N-i}^2 y_{N-j}^2\Big] =$
$\Big(\frac{d^2\alpha^2\big(1-\alpha^N\big)^2}{(1-\alpha)^2}+\frac{(2d^2+6d)\alpha^2\big(1-\alpha^{2N}\big)}{(1-\alpha)(1+\alpha)}\Big)$.

$$\mathbb{E}\Big[\underbrace{\boldsymbol{w}^\top \boldsymbol{x}_q}_{\spadesuit}\cdot\sum_{i=0}^{N-1}\alpha^{i+1}y_{N-i}(\boldsymbol{B}\boldsymbol{x}_{N-i}+y_{N-i}\boldsymbol{b})\boldsymbol{x}_q^\top\Big]$$

$$=\mathbb{E}\Big[\sum_{i=0}^{N-1}\alpha^{i+1}y_{N-i}(\boldsymbol{B}\boldsymbol{x}_{N-i}+y_{N-i}\boldsymbol{b})\underbrace{\boldsymbol{w}^\top \boldsymbol{x}_q}_{\spadesuit}\boldsymbol{x}_q^\top\Big]$$

$$= B\mathbb{E}\Big[\sum_{i=0}^{N-1}\alpha^{i+1}y_{N-i}\boldsymbol{x}_{N-i}\boldsymbol{w}^\top\Big]\mathbb{E}\Big[\boldsymbol{x}_q\boldsymbol{x}_q^\top\Big]$$

$$+\,\boldsymbol{b}\mathbb{E}\Big[\sum_{i=0}^{N-1}\alpha^{i+1}y_{N-i}^2\boldsymbol{w}^\top\Big]\mathbb{E}\Big[\boldsymbol{x}_q\boldsymbol{x}_q^\top\Big]$$

$$= \alpha\Big(\frac{1-\alpha^N}{1-\alpha}\Big)\boldsymbol{B}$$

The last equality follows from lemma A.3, where we have: $\mathbb{E}\Big[\sum_{i=0}^{N-1}\alpha^{i+1}y_{N-i}\boldsymbol{x}_{N-i}\boldsymbol{w}^\top\Big] = \alpha\Big(\frac{1-\alpha^N}{1-\alpha}\Big)\boldsymbol{I}$, $\mathbb{E}\Big[\sum_{i=0}^{N-1}\alpha^{i+1}y_{N-i}^2\boldsymbol{w}\Big] = \boldsymbol{0}$, and $\mathbb{E}\Big[\boldsymbol{x}_q\boldsymbol{x}_q^\top\Big] = \boldsymbol{I}$.

$$\mathbb{E}\Big[\underbrace{\boldsymbol{x}_q^\top\boldsymbol{C}^\top\sum_{i=0}^{N-1}\alpha^{i+1}y_{N-i}(\boldsymbol{B}\boldsymbol{x}_{N-i}+y_{N-i}\boldsymbol{b})}_{\spadesuit}\cdot\boldsymbol{C}\boldsymbol{x}_q\sum_{i=0}^{N-1}\alpha^{i+1}y_{N-i}\boldsymbol{x}_{N-i}^\top\Big]$$

$$= \mathbb{E}\Big[\boldsymbol{C}\boldsymbol{x}_q\underbrace{\Big(\boldsymbol{x}_q^\top\boldsymbol{C}^\top\sum_{i=0}^{N-1}\alpha^{i+1}y_{N-i}(\boldsymbol{B}\boldsymbol{x}_{N-i}+y_{N-i}\boldsymbol{b})\Big)}_{\spadesuit}\sum_{i=0}^{N-1}\alpha^{i+1}y_{N-i}\boldsymbol{x}_{N-i}^\top\Big]$$

$$= \boldsymbol{C}\mathbb{E}\Big[\boldsymbol{x}_q\boldsymbol{x}_q^\top\Big]\boldsymbol{C}^\top\boldsymbol{B}\mathbb{E}\Big[\sum_{i=0}^{N-1}\sum_{j=0}^{N-1}\alpha^{i+j+2}y_{N-i}y_{N-j}\boldsymbol{x}_{N-i}\boldsymbol{x}_{N-j}^\top\Big]$$

$$= \Big(\frac{\alpha^2\big(1-\alpha^N\big)^2}{(1-\alpha)^2}+\frac{(d+1)\alpha^2\big(1-\alpha^{2N}\big)}{(1-\alpha)(1+\alpha)}\Big)\boldsymbol{C}\boldsymbol{C}^\top\boldsymbol{B}$$

The last equality follows from lemma A.3, where we have: $\mathbb{E}\Big[\sum_{i=0}^{N-1}\sum_{j=0}^{N-1}\alpha^{i+j+2}y_{N-i}y_{N-j}\boldsymbol{x}_{N-i}\boldsymbol{x}_{N-j}^\top\Big] = \Big(\frac{\alpha^2\big(1-\alpha^N\big)^2}{(1-\alpha)^2}+\frac{(d+1)\alpha^2\big(1-\alpha^{2N}\big)}{(1-\alpha)(1+\alpha)}\Big)\boldsymbol{I}$, and $\mathbb{E}\Big[\boldsymbol{x}_q\boldsymbol{x}_q^\top\Big] = \boldsymbol{I}$.

$$\mathbb{E}\Big[\underbrace{\boldsymbol{w}^\top\boldsymbol{x}_q}_{\spadesuit}\cdot\boldsymbol{C}\boldsymbol{x}_q\sum_{i=0}^{N-1}\alpha^{i+1}y_{N-i}\boldsymbol{x}_{N-i}^\top\Big]$$

$$= \boldsymbol{C}\mathbb{E}\Big[\boldsymbol{x}_q\underbrace{\boldsymbol{x}_q^\top\boldsymbol{w}}_{\spadesuit}\sum_{i=0}^{N-1}\alpha^{i+1}y_{N-i}\boldsymbol{x}_{N-i}^\top\Big]$$

$$= \boldsymbol{C}\mathbb{E}\Big[\boldsymbol{x}_q\boldsymbol{x}_q^\top\Big]\mathbb{E}\Big[\sum_{i=0}^{N-1}\alpha^{i+1}y_{N-i}\boldsymbol{w}\boldsymbol{x}_{N-i}^\top\Big]$$

$$= \alpha\Big(\frac{1-\alpha^N}{1-\alpha}\Big)\boldsymbol{C}$$

The last equality follows from lemma A.3, where we have: $\mathbb{E}\Big[\sum_{i=0}^{N-1}\alpha^{i+1}y_{N-i}\boldsymbol{x}_{N-i}\boldsymbol{w}^\top\Big] = \alpha\Big(\frac{1-\alpha^N}{1-\alpha}\Big)\boldsymbol{I}$, and $\mathbb{E}\Big[\boldsymbol{x}_q\boldsymbol{x}_q^\top\Big] = \boldsymbol{I}$.

$$\mathbb{E}\Big[\underbrace{\Big(\boldsymbol{x}_q^\top\boldsymbol{C}^\top\sum_{i=0}^{N-1}\alpha^{i+1}y_{N-i}(\boldsymbol{B}\boldsymbol{x}_{N-i}+y_{N-i}\boldsymbol{b})\Big)}_{\spadesuit}\cdot\boldsymbol{C}\boldsymbol{x}_q\cdot\sum_{i=0}^{N-1}\alpha^{i+1}y_{N-i}^2\Big]$$

$$= \mathbb{E}\Big[ C\boldsymbol{x}_q \underbrace{\Big(\boldsymbol{x}_q^\top C^\top \sum_{i=0}^{N-1} \alpha^{i+1} y_{N-i}(B\boldsymbol{x}_{N-i} + y_{N-i}\boldsymbol{b})\Big)}_{\spadesuit} \cdot \sum_{i=0}^{N-1} \alpha^{i+1} y_{N-i}^2 \Big]$$

$$= C\mathbb{E}\Big[\boldsymbol{x}_q \boldsymbol{x}_q^\top\Big] C^\top B \mathbb{E}\Big[ \sum_{i=0}^{N-1}\sum_{j=0}^{N-1} \alpha^{i+j+2} y_{N-i} y_{N-j}^2 \boldsymbol{x}_{N-i}\Big]$$

$$+ C\mathbb{E}\Big[\boldsymbol{x}_q \boldsymbol{x}_q^\top\Big] C^\top \boldsymbol{b}\, \mathbb{E}\Big[ \sum_{i=0}^{N-1}\sum_{j=0}^{N-1} \alpha^{i+j+2} y_{N-i}^2 y_{N-j}^2 \Big]$$

$$= \Big(\frac{d^2\alpha^2\Big(1-\alpha^N\Big)^2}{(1-\alpha)^2} + \frac{(2d^2+6d)\alpha^2\Big(1-\alpha^{2N}\Big)}{(1-\alpha)(1+\alpha)}\Big) CC^\top \boldsymbol{b}$$

The last equality follows from lemma A.3, where we have: $\mathbb{E}\Big[ \sum_{i=0}^{N-1}\sum_{j=0}^{N-1} \alpha^{i+j+2} y_{N-i} y_{N-j}^2 \boldsymbol{x}_{N-i}\Big] = \mathbf{0}$, $\mathbb{E}\Big[ \sum_{i=0}^{N-1}\sum_{j=0}^{N-1} \alpha^{i+j+2} y_{N-i}^2 y_{N-j}^2\Big] = \Big(\frac{d^2\alpha^2\Big(1-\alpha^N\Big)^2}{(1-\alpha)^2} + \frac{(2d^2+6d)\alpha^2\Big(1-\alpha^{2N}\Big)}{(1-\alpha)(1+\alpha)}\Big)$, and $\mathbb{E}\Big[\boldsymbol{x}_q \boldsymbol{x}_q^\top\Big] = I$.

$$\mathbb{E}\Big[ \underbrace{\boldsymbol{w}^\top \boldsymbol{x}_q}_{\spadesuit} \cdot C\boldsymbol{x}_q \cdot \sum_{i=0}^{N-1} \alpha^{i+1} y_{N-i}^2 \Big]$$

$$= \mathbb{E}\Big[ C\boldsymbol{x}_q \underbrace{\boldsymbol{x}_q^\top \boldsymbol{w}}_{\spadesuit} \cdot \sum_{i=0}^{N-1} \alpha^{i+1} y_{N-i}^2 \Big]$$

$$= C\mathbb{E}\Big[\boldsymbol{x}_q \boldsymbol{x}_q^\top\Big] \mathbb{E}\Big[ \sum_{i=0}^{N-1} \alpha^{i+1} y_{N-i}^2 \boldsymbol{w}\Big]$$

$$= \mathbf{0}$$

The last equality follows from lemma A.3, where we have $\mathbb{E}\Big[ \sum_{i=0}^{N-1} \alpha^{i+1} y_{N-i}^2 \boldsymbol{w}\Big] = \mathbf{0}$.

$$\mathbb{E}\Big[ \boldsymbol{x}_q^\top C^\top \underbrace{\Big( \sum_{i=0}^{N-1} \alpha^{i+1} y_{N-i}(B\boldsymbol{x}_{N-i} + y_{N-i}\boldsymbol{b} + \boldsymbol{b}_B)\Big)}_{\spadesuit} \cdot C\boldsymbol{x}_q \cdot \sum_{i=0}^{N-1} \alpha^{i+1} y_{N-i} \Big]$$

$$= \mathbb{E}\Big[ C\boldsymbol{x}_q \cdot \underbrace{\boldsymbol{x}_q^\top C^\top \Big( \sum_{i=0}^{N-1} \alpha^{i+1} y_{N-i}(B\boldsymbol{x}_{N-i} + y_{N-i}\boldsymbol{b} + \boldsymbol{b}_B)\Big)}_{\spadesuit} \cdot \sum_{j=0}^{N-1} \alpha^{j+1} y_{N-j} \Big]$$

$$= C\mathbb{E}\Big[\boldsymbol{x}_q \boldsymbol{x}_q^\top\Big] C^\top B \mathbb{E}\Big[ \sum_{i=0}^{N-1}\sum_{j=0}^{N-1} \alpha^{i+j+2} y_{N-i} y_{N-j} \boldsymbol{x}_{N-i}\Big]$$

$$+ C\mathbb{E}\Big[\boldsymbol{x}_q \boldsymbol{x}_q^\top\Big] C^\top \boldsymbol{b}\, \mathbb{E}\Big[ \sum_{i=0}^{N-1}\sum_{j=0}^{N-1} \alpha^{i+j+2} y_{N-i}^2 y_{N-j} \Big]$$

$$+ C\mathbb{E}\Big[\boldsymbol{x}_q \boldsymbol{x}_q^\top\Big] C^\top \boldsymbol{b}_B\, \mathbb{E}\Big[ \sum_{i=0}^{N-1}\sum_{j=0}^{N-1} \alpha^{i+j+2} y_{N-i} y_{N-j} \Big]$$

$$= \frac{d\alpha^2\Big(1-\alpha^{2N}\Big)}{(1-\alpha)(1+\alpha)} CC^\top \boldsymbol{b}_B$$

The last equality follows from lemma A.3, where we have: $\mathbb{E}\left[\sum_{i=0}^{N-1}\sum_{j=0}^{N-1}\alpha^{i+j+2}\boldsymbol{x}_{N-i}\underbrace{\boldsymbol{x}_{N-i}^{\top}\boldsymbol{w}}_{y_{N-i}}\underbrace{\boldsymbol{x}_{N-j}^{\top}\boldsymbol{w}}_{y_{N-j}}\right]=\boldsymbol{0}$, $\mathbb{E}\left[\sum_{i=0}^{N-1}\sum_{j=0}^{N-1}\alpha^{i+j+2}y_{N-i}^{2}y_{N-j}\right]=0$,

$\mathbb{E}\left[\sum_{i=0}^{N-1}\sum_{j=0}^{N-1}\alpha^{i+j+2}y_{N-i}y_{N-j}\right]=\dfrac{d\alpha^{2}\left(1-\alpha^{2N}\right)}{(1-\alpha)(1+\alpha)}$, and $\mathbb{E}\left[\boldsymbol{x}_{q}\boldsymbol{x}_{q}^{\top}\right]=\boldsymbol{I}$.

$$\mathbb{E}\left[\underbrace{\boldsymbol{w}^{\top}\boldsymbol{x}_{q}}_{\spadesuit}\cdot\boldsymbol{C}\boldsymbol{x}_{q}\cdot\sum_{i=0}^{N-1}\alpha^{i+1}y_{N-i}\right]$$

$$=\mathbb{E}\left[\boldsymbol{C}\boldsymbol{x}_{q}\underbrace{\boldsymbol{x}_{q}^{\top}\boldsymbol{w}}_{\spadesuit}\cdot\sum_{i=0}^{N-1}\alpha^{i+1}y_{N-i}\right]$$

$$=\boldsymbol{C}\mathbb{E}\left[\boldsymbol{x}_{q}\boldsymbol{x}_{q}^{\top}\right]\mathbb{E}\left[\sum_{i=0}^{N-1}\alpha^{i+1}y_{N-i}\boldsymbol{w}\right]$$

$$=\boldsymbol{0}$$

The last equality follows from lemma A.3, where we have $\mathbb{E}\left[\sum_{i=0}^{N-1}\alpha^{i+1}y_{N-i}\boldsymbol{w}\right]=\boldsymbol{0}$.

## C.4 Proof of claim B.1

This Section presents the bounds for terms $\boldsymbol{b}_{i}^{\top}(T+1)\boldsymbol{b}_{i}(T+1)$, $\boldsymbol{c}_{i}^{\top}(Tt+1)\boldsymbol{c}_{i}(T+1)$ and $\boldsymbol{b}^{\top}(T+1)\boldsymbol{b}(T+1)$, establishing the property $\mathcal{A}(T+1)$.

Recurring the *Vector-coupled Dynamics* equations of $\boldsymbol{b}_{i}^{\top}(t+1)\boldsymbol{b}_{i}(t+1)$, $\boldsymbol{c}_{i}^{\top}(t+1)\boldsymbol{c}_{i}(t+1)$ and $\boldsymbol{b}^{\top}(t+1)\boldsymbol{b}(t+1)$ in lemma A.7, we have:

$$\boldsymbol{b}_{i}^{\top}(T+1)\boldsymbol{b}_{i}(T+1)=\boldsymbol{b}_{i}^{\top}(T)\boldsymbol{b}_{i}(T)+2\eta\Big(\big(\beta_{3}-\beta_{1}\boldsymbol{c}_{i}^{\top}(T)\boldsymbol{b}_{i}(T)\big)\boldsymbol{c}_{i}^{\top}(T)\boldsymbol{b}_{i}(T)$$

$$-\beta_{1}\sum_{k\neq i}^{d}\big(\boldsymbol{c}_{k}^{\top}(T)\boldsymbol{b}_{i}(T)\big)^{2}\Big)+\eta^{2}\big\|\bar{\boldsymbol{b}}_{i}(T)\big\|_{2}^{2}$$

$$=\boldsymbol{b}_{i}^{\top}(0)\boldsymbol{b}_{i}(0)+\sum_{s=0}^{T}\Big(2\eta\big(\big(\beta_{3}-\beta_{1}\boldsymbol{c}_{i}^{\top}(s)\boldsymbol{b}_{i}(s)\big)\boldsymbol{c}_{i}^{\top}(s)\boldsymbol{b}_{i}(s)-\beta_{1}\sum_{k\neq i}^{d}\big(\boldsymbol{c}_{k}^{\top}(s)\boldsymbol{b}_{i}(s)\big)^{2}\big)$$

$$+\eta^{2}\big\|\bar{\boldsymbol{b}}_{i}(s)\big\|_{2}^{2}\Big)$$

$$=\boldsymbol{b}_{i}^{\top}(0)\boldsymbol{b}_{i}(0)+2\eta\underbrace{\sum_{s=0}^{T}\big(\beta_{3}-\beta_{1}\boldsymbol{c}_{i}^{\top}(s)\boldsymbol{b}_{i}(s)\big)\boldsymbol{c}_{i}^{\top}(s)\boldsymbol{b}_{i}(s)}_{\text{term I}}-2\eta\beta_{1}\sum_{k\neq i}^{d}\underbrace{\sum_{s=0}^{T}\big(\boldsymbol{c}_{k}^{\top}(s)\boldsymbol{b}_{i}(s)\big)^{2}}_{\text{term II}}$$

$$+\eta^{2}\underbrace{\sum_{s=0}^{T}\big\|\bar{\boldsymbol{b}}_{i}(s)\big\|_{2}^{2}}_{\text{term III}}$$

$$\boldsymbol{c}_{i}^{\top}(T+1)\boldsymbol{c}_{i}(T+1)=\boldsymbol{c}_{i}^{\top}(T)\boldsymbol{c}_{i}(T)+2\eta\Big(\big(\beta_{3}-\beta_{1}\boldsymbol{c}_{i}^{\top}(T)\boldsymbol{b}_{i}(T)\big)\boldsymbol{c}_{i}^{\top}(T)\boldsymbol{b}_{i}(T)$$

$$-\beta_{1}\sum_{k\neq i}^{d}\big(\boldsymbol{c}_{i}^{\top}(T)\boldsymbol{b}_{k}(T)\big)^{2}-\beta_{2}\big(\boldsymbol{c}_{i}^{\top}(T)\boldsymbol{b}(T)\big)^{2}\Big)+\eta^{2}\big\|\bar{\boldsymbol{c}}_{i}(T)\big\|_{2}^{2}$$

$$=\boldsymbol{c}_{i}^{\top}(T)\boldsymbol{c}_{i}(T)+\sum_{s=0}^{T}\Big(2\eta\big(\big(\beta_{3}-\beta_{1}\boldsymbol{c}_{i}^{\top}(s)\boldsymbol{b}_{i}(s)\big)\boldsymbol{c}_{i}^{\top}(s)\boldsymbol{b}_{i}(s)-\beta_{1}\sum_{k\neq i}^{d}\big(\boldsymbol{c}_{i}^{\top}(s)\boldsymbol{b}_{k}(s)\big)^{2}$$

$$- \beta_2 \big(\boldsymbol{c}_i^\top(s)\boldsymbol{b}(s)\big)^2\big) + \eta^2 \big\|\bar{\boldsymbol{c}}_i(s)\big\|_2^2\Big)$$

$$= \boldsymbol{c}_i^\top(0)\boldsymbol{c}_i(0) + 2\eta \underbrace{\sum_{s=0}^{T} \big(\beta_3 - \beta_1 \boldsymbol{c}_i^\top(s)\boldsymbol{b}_i(s)\big)\boldsymbol{c}_i^\top(s)\boldsymbol{b}_i(s)}_{\text{term I}} - 2\eta\beta_1 \sum_{k\neq i}^{d} \underbrace{\sum_{s=0}^{T} \big(\boldsymbol{c}_i^\top(s)\boldsymbol{b}_k(s)\big)^2}_{\text{=term II}}$$

$$- 2\eta\beta_2 \underbrace{\sum_{s=0}^{T} \big(\boldsymbol{c}_i^\top(s)\boldsymbol{b}(s)\big)^2}_{\text{term IV}} + \eta^2 \underbrace{\sum_{s=0}^{T} \big\|\bar{\boldsymbol{c}}_i(s)\big\|_2^2}_{\text{term V}}$$

$$\boldsymbol{b}^\top(T+1)\boldsymbol{b}(T+1) = \boldsymbol{b}^\top(T)\boldsymbol{b}(T) - 2\eta\Big(\beta_2 \sum_{k=1}^{d} \big(\boldsymbol{c}_k^\top(T)\boldsymbol{b}(T)\big)^2\Big) + \eta^2 \big\|\bar{\boldsymbol{b}}(T)\big\|_2^2$$

$$= \boldsymbol{b}^\top(0)\boldsymbol{b}(0) + 2\eta\beta_2 \sum_{k=1}^{d} \underbrace{\sum_{s=0}^{T} \big(\boldsymbol{c}_k^\top(s)\boldsymbol{b}(s)\big)^2}_{\text{=term IV}} + \eta^2 \underbrace{\sum_{s=0}^{T} \big\|\bar{\boldsymbol{b}}(s)\big\|_2^2}_{\text{term VI}}$$

To bound terms I - VI, we will use some inequalities from property $\mathcal{B}(t)$ and lemma C.7 as following with $i, j, k \in [1, d], i \neq j$:

$$|\beta_3 - \beta_1 \boldsymbol{c}_i^\top(s)\boldsymbol{b}_i(s)| \leq \delta(t) \exp(-\eta\beta_1\gamma t)$$

$$|\boldsymbol{c}_i^\top(t)\boldsymbol{b}_j(t)| \leq 2\delta(t) \exp(-\eta\beta_1\gamma t)$$

$$|\boldsymbol{c}_i^\top(t)\boldsymbol{b}(t)| \leq 2\delta(t) \exp(-\eta\beta_2\gamma t) + \frac{\delta(t)}{\beta_2} \exp(-\eta\beta_1\gamma t)$$

$$\left|\bar{\boldsymbol{b}}_i(t)^\top\bar{\boldsymbol{b}}_k(t)\right| \leq 8d_h d^2\delta(t)^2 \exp(-\eta\beta_1\gamma t),$$

$$\left|\bar{\boldsymbol{c}}_i(t)^\top\bar{\boldsymbol{c}}_k(t)\right| \leq 8d_h d^2\delta(t)^2 \exp(-\eta\beta_1\gamma t) + 40\beta_2^2 d_h \delta(t)^2 \exp(-\eta\beta_2\gamma t),$$

$$\left\|\bar{\boldsymbol{b}}(t)\right\|_2^2 \leq 16d_h\beta_2^2 d^2\delta(t)^2 \exp(-\eta\beta_2\gamma t) + 2d_h d^2\delta(t)^2 \exp(-\eta\beta_1\gamma t),$$

Next we begin bounding terms I - VI.

**Bound of term I:**   By $\left|\beta_3 - \beta_1 \boldsymbol{c}_i^\top(s)\boldsymbol{b}_i(s)\right| \leq \delta(s) \exp(-\eta\beta_1\gamma s)$, we have:

$$\left|\boldsymbol{c}_i^\top(s)\boldsymbol{b}_i(s)\right| \leq \frac{\delta(s) \exp(-\eta\beta_1\gamma s) + \beta_3}{\beta_1}$$

$$\leq \frac{4\delta(s) + 2\alpha}{\alpha^2}$$

$$\leq \frac{5\delta(s)}{\alpha^2}$$

$$\leq 6\delta(s)$$

The third inequality is by $\delta(s) \geq 2\sqrt{d_h \log(4d(2d+1)/\delta)} \geq 2\alpha = 2\alpha = 2\exp((-\ln 2)/N)$. For the last inequality, as long as $N \geq \frac{2\ln 2}{\ln 6 - \ln 5}$, we have $\frac{5}{\alpha^2} \leq 6$.

$$\left|\sum_{s=0}^{T} \big(\beta_3 - \beta_1 \boldsymbol{c}_i^\top(s)\boldsymbol{b}_i(s)\big)\boldsymbol{c}_i^\top(s)\boldsymbol{b}_i(s)\right|$$

$$\leq \sum_{s=0}^{T} 6\delta(s)^2 \exp(-\eta\beta_1\gamma s)$$

$$\leq 6\delta_{\max}^2 \int_{-1}^{\infty} \exp(-\eta\beta_1\gamma s)ds$$

$$\leq \frac{6\delta_{\max}^2 \exp(\eta\beta_1\gamma)}{\eta\beta_1\gamma}$$

The second inequality is due to $\exp(-\eta\beta_1\gamma s)$ is monotone decreasing.

**Bound of term II:**

$$\sum_{s=0}^{T} \left(\boldsymbol{c}_k^\top(s)\boldsymbol{b}_i(s)\right)^2$$

$$\leq \sum_{s=0}^{T} \left(2\delta(s)\exp(-\eta\beta_1\gamma s)\right)^2$$

$$\leq 4\delta_{\max}^2 \int_{-1}^{\infty} \exp(-2\eta\beta_1\gamma s)ds$$

$$\leq \frac{2\delta_{\max}^2 \exp(2\eta\beta_1\gamma)}{\eta\beta_1\gamma}$$

The second inequality is due to $\exp(-2\eta\beta_1\gamma s)$ is monotone decreasing.

**Bound of term III:**

$$\sum_{s=0}^{T} \left\|\bar{\boldsymbol{b}}_i(s)\right\|_2^2$$

$$\leq \sum_{s=0}^{T} 8d_h d^2 \delta(t)^2 \exp(-\eta\beta_1\gamma t)$$

$$\leq 8d_h d^2 \delta_{\max}^2 \int_{-1}^{\infty} \exp(-\eta\beta_1\gamma s)ds$$

$$\leq \frac{8d_h d^2 \delta_{\max}^2 \exp(\eta\beta_1\gamma)}{\eta\beta_1\gamma}$$

The second inequality is due to $\exp(-\eta\beta_1\gamma s)$ is monotone decreasing.

**Bound of term IV:**

$$\sum_{s=0}^{T} \left(\boldsymbol{c}_i^\top(s)\boldsymbol{b}(s)\right)^2$$

$$= \sum_{s=0}^{T} \left(2\delta(s)\exp(-\eta\beta_2\gamma s) + \frac{\delta(s)}{\beta_2}\exp(-\eta\beta_1\gamma s)\right)^2$$

$$\leq \delta_{\max}^2 \cdot \left(4\sum_{s=0}^{T}\exp(-2\eta\beta_2\gamma s) + \frac{4}{\beta_2}\sum_{s=0}^{T}\exp(-\eta(\beta_1+\beta_2)\gamma s) + \frac{1}{\beta_2^2}\sum_{s=0}^{T}\exp(-2\eta\beta_1\gamma s)\right)$$

$$\leq \delta_{\max}^2 \cdot \left(4\int_{-1}^{\infty}\exp(-2\eta\beta_2\gamma s)ds + \frac{4}{\beta_2}\int_{-1}^{\infty}\exp(-\eta(\beta_1+\beta_2)\gamma s)ds\right.$$

$$\left.+ \frac{1}{\beta_2^2}\int_{-1}^{\infty}\exp(-2\eta\beta_1\gamma s)ds\right)$$

$$= \frac{2\delta_{\max}^2 \exp(2\eta\beta_2\gamma)}{\eta\beta_2\gamma} + \frac{4\delta_{\max}^2 \exp(\eta(\beta_1+\beta_2)\gamma)}{\eta\beta_2(\beta_1+\beta_2)\gamma} + \frac{\delta_{\max}^2 \exp(2\eta\beta_1\gamma)}{2\eta\beta_1\beta_2^2\gamma}$$

$$\leq \frac{17\delta_{\max}^2}{\eta\beta_2\gamma}$$

The second inequality is due to $\exp(-2\eta\beta_2\gamma s)$, $\exp(-\eta(\beta_1 + \beta_2)\gamma s)$ and $\exp(-2\eta\beta_1\gamma s)$ are monotone decreasing. The last inequality is by $\frac{\exp(\eta(\beta_1+\beta_2)\gamma)}{(\beta_1+\beta_2)} \leq 2$ and $\frac{\exp(2\eta\beta_1\gamma)}{2\beta_1\beta_2} \leq 7$ since $\exp(2\eta\beta_1\gamma) \leq \exp(\eta(\beta_1 + \beta_2)\gamma) \leq 2$ and $\beta_1 + \beta_2 \geq 1$, $\beta_1\beta_2 \geq \frac{1}{7}$.

**Bound of term V:**

$$\sum_{s=0}^{T} \left\| \bar{\boldsymbol{c}}_i(s) \right\|_2^2$$

$$\leq \sum_{s=0}^{T} \left( 8d_h d^2 \delta(t)^2 \exp(-\eta\beta_1\gamma t) + 40\beta_2^2 d_h \delta(t)^2 \exp(-\eta\beta_2\gamma t) \right)$$

$$\leq 8d_h d^2 \delta_{\max}^2 \int_{-1}^{\infty} \exp(-\eta\beta_1\gamma s)ds + 40\beta_2^2 d_h \delta_{\max}^2 \int_{-1}^{\infty} \exp(-\eta\beta_2\gamma s)ds$$

$$= \frac{8d_h d^2 \delta_{\max}^2 \exp(\eta\beta_1\gamma)}{\eta\beta_1\gamma} + \frac{40\beta_2 d_h \delta_{\max}^2 \exp(\eta\beta_2\gamma)}{\eta\gamma}$$

$$\leq \frac{16d_h d^2 \delta_{\max}^2 \exp(\eta\beta_1\gamma)}{\eta\beta_1\gamma} + \frac{80\beta_2 d_h \delta_{\max}^2 \exp(\eta\beta_2\gamma)}{\eta\gamma}$$

The second inequality is due to $\exp(-\eta\beta_2\gamma s)$ and $\exp(-\eta\beta_1\gamma s)$ are monotone decreasing.

**Bound of term VI:**

$$\sum_{s=0}^{T} \left\| \bar{\boldsymbol{b}}(s) \right\|_2^2$$

$$\leq \sum_{s=0}^{T} \left( 16d_h \beta_2^2 d^2 \delta(t)^2 \exp(-\eta\beta_2\gamma t) + 2d_h d^2 \delta(t)^2 \exp(-\eta\beta_1\gamma t) \right)$$

$$\leq 16d_h \beta_2^2 d^2 \delta_{\max}^2 \int_{-1}^{\infty} \exp(-\eta\beta_2\gamma s)ds + 2d_h d^2 \delta_{\max}^2 \int_{-1}^{\infty} \exp(-\eta\beta_1\gamma s)ds$$

$$\leq \frac{16d_h \beta_2 d^2 \delta_{\max}^2 \exp(\eta\beta_2\gamma)}{\eta\gamma} + \frac{2d_h d^2 \delta_{\max}^2 \exp(\eta\beta_1\gamma)}{\eta\beta_1\gamma}$$

The second inequality is due to $\exp(-\eta\beta_2\gamma s)$ and $\exp(-\eta\beta_1\gamma s)$ are monotone decreasing.

We next use the bounds of I - VI to bound $\boldsymbol{b}_i^\top(T + 1)\boldsymbol{b}_i(T + 1)$, $\boldsymbol{c}_i^\top(T + 1)\boldsymbol{c}_i(T + 1)$ and $\boldsymbol{b}^\top(T + 1)\boldsymbol{b}(T + 1)$.

**Lower bound of $\boldsymbol{b}_i^\top(T + 1)\boldsymbol{b}_i(T + 1)$**

$$\boldsymbol{b}_i^\top(T + 1)\boldsymbol{b}_i(T + 1)$$

$$= \underbrace{\boldsymbol{b}_i^\top(0)\boldsymbol{b}_i(0)}_{\geq \frac{3d_h}{4}} + \underbrace{2\eta \sum_{s=0}^{T} \left( \beta_3 - \beta_1 \boldsymbol{c}_i^\top(s)\boldsymbol{b}_i(s) \right) \boldsymbol{c}_i^\top(s)\boldsymbol{b}_i(s)}_{\text{term I}} - \underbrace{2\eta\beta_1 \sum_{k \neq i}^{d} \sum_{s=0}^{T} \left( \boldsymbol{c}_k^\top(s)\boldsymbol{b}_i(s) \right)^2}_{\text{term II}}$$

$$+ \underbrace{\eta^2 \sum_{s=0}^{T} \left\| \bar{\boldsymbol{b}}_i(s) \right\|_2^2}_{\geq 0}$$

$$\geq \frac{3d_h}{4} - 2\eta \cdot \frac{6\delta_{\max}^2 \exp(\eta\beta_1\gamma)}{\eta\beta_1\gamma} - 2\eta\beta_1(d - 1) \cdot \frac{2\delta_{\max}^2 \exp(2\eta\beta_1\gamma)}{\eta\beta_1\gamma}$$

$$\geq \frac{3d_h}{4} - \frac{12\delta_{\max}^2 \exp(\eta\beta_1\gamma)}{\beta_1\gamma} - \frac{4(d - 1)\delta_{\max}^2 \exp(2\eta\beta_1\gamma)}{\gamma}$$

$$\geq \frac{3d_h}{4} - \frac{2*12*9d_h \log(4d(2d+1)/\delta)}{\beta_1 \frac{1}{2}d_h} - \frac{2*4(d-1)*9d_h \log(4d(2d+1)/\delta)}{\frac{1}{2}d_h}$$

$$\geq \frac{d_h}{2}$$

The third inequality is by $\delta_{\max} = 3\sqrt{d_h \log(4d(2d+1)/\delta)}$, $\exp(\eta\beta_1\gamma) \leq \exp(2\eta\beta_1\gamma) \leq 2$ and $\gamma = \frac{1}{2}d_h$. The last inequality follows from $d_h = \widetilde{\Omega}(d^2) \geq \big(1728 \log(4d(2d+1)/\delta) + 576(d-1)\beta_1 \log(4d(2d+1)/\delta)\big)/\beta_1$.

**Upper bound of $\boldsymbol{b}_i^\top(T+1)\boldsymbol{b}_i(T+1)$**

$$\boldsymbol{b}_i^\top(T+1)\boldsymbol{b}_i(T+1)$$

$$= \underbrace{\boldsymbol{b}_i^\top(0)\boldsymbol{b}_i(0)}_{\leq \frac{5d_h}{4}} - 2\eta \underbrace{\sum_{s=0}^{T} \big(\beta_3 - \beta_1 \boldsymbol{c}_i^\top(s)\boldsymbol{b}_i(s)\big)\boldsymbol{c}_i^\top(s)\boldsymbol{b}_i(s)}_{\text{term I}} - 2\eta\beta_1 \underbrace{\sum_{k\neq i}^{d} \sum_{s=0}^{T} \big(\boldsymbol{c}_k^\top(s)\boldsymbol{b}_i(s)\big)^2}_{\text{term II}}$$

$$+ \eta^2 \underbrace{\sum_{s=0}^{T} \left\|\bar{\boldsymbol{b}}_i(s)\right\|_2^2}_{\text{term III}}$$

$$\leq \frac{5d_h}{4} + 2\eta \cdot \frac{6\delta_{\max}^2 \exp(\eta\beta_1\gamma)}{\eta\beta_1\gamma} + \eta^2 \cdot \frac{8d_h d^2 \delta_{\max}^2 \exp(\eta\beta_1\gamma)}{\eta\beta_1\gamma}$$

$$\leq \frac{5d_h}{4} + \frac{12\delta_{\max}^2 \exp(\eta\beta_1\gamma)}{\beta_1\gamma} + \frac{8\eta d_h d^2 \delta_{\max}^2 \exp(\eta\beta_1\gamma)}{\beta_1\gamma}$$

$$\leq \frac{5d_h}{4} + \frac{2*12*9d_h \log(4d(2d+1)/\delta)}{\beta_1 \frac{1}{2}d_h} + \frac{2*8\eta d_h d^2 * 9d_h \log(4d(2d+1)/\delta)}{\beta_1 \frac{1}{2}d_h}$$

$$\leq 2d_h$$

The third inequality is by $\delta_{\max} = 3\sqrt{d_h \log(4d(2d+1)/\delta)}$, $\exp(\eta\beta_1\gamma) \leq 2$ and $\gamma = \frac{1}{2}d_h$. The last inequality follows from

$$d_h = \widetilde{\Omega}(d^2)$$
$$\geq \big(576 \log(4d(2d+1)/\delta) + 192 \log(4d(2d+1)/\delta)\big)/\beta_1$$
$$\geq \big(576 \log(4d(2d+1)/\delta) + 384\eta d_h d^2 \log(4d(2d+1)/\delta)\big)/\beta_1$$

**Lower bound of $\boldsymbol{c}_i^\top(T+1)\boldsymbol{c}_i(T+1)$**

$$\boldsymbol{c}_i^\top(T+1)\boldsymbol{c}_i(T+1)$$

$$= \underbrace{\boldsymbol{c}_i^\top(0)\boldsymbol{c}_i(0)}_{\geq \frac{3d_h}{4}} + 2\eta \underbrace{\sum_{s=0}^{T} \big(\beta_3 - \beta_1 \boldsymbol{c}_i^\top(s)\boldsymbol{b}_i(s)\big)\boldsymbol{c}_i^\top(s)\boldsymbol{b}_i(s)}_{\text{term I}} - 2\eta\beta_1 \underbrace{\sum_{k\neq i}^{d} \sum_{s=0}^{T} \big(\boldsymbol{c}_i^\top(s)\boldsymbol{b}_k(s)\big)^2}_{= \text{term II}}$$

$$- 2\eta\beta_2 \underbrace{\sum_{s=0}^{T} \big(\boldsymbol{c}_i^\top(s)\boldsymbol{b}(s)\big)^2}_{\text{term IV}} + \eta^2 \underbrace{\sum_{s=0}^{T} \left\|\bar{\boldsymbol{c}}_i(s)\right\|_2^2}_{\geq 0}$$

$$\geq \frac{3d_h}{4} - 2\eta \cdot \frac{6\delta_{\max}^2 \exp(\eta\beta_1\gamma)}{\eta\beta_1\gamma} - 2\eta\beta_1(d-1) \cdot \frac{2\delta_{\max}^2 \exp(2\eta\beta_1\gamma)}{\eta\beta_1\gamma} - 2\eta\beta_2 \cdot \frac{17\delta_{\max}^2}{\eta\beta_2\gamma}$$

$$\geq \frac{3d_h}{4} - \frac{2*12*9d_h \log(4d(2d+1)/\delta)}{\beta_1 \frac{1}{2}d_h} - \frac{2*4(d-1)*9d_h \log(4d(2d+1)/\delta)}{\frac{1}{2}d_h}$$

$$- \frac{34*9d_h \log(4d(2d+1)/\delta)}{\frac{1}{2}d_h}$$

$$\geq \frac{d_h}{2}$$

The second inequality is by $\delta_{\max} = 3\sqrt{d_h \log(4d(2d+1)/\delta)}$, $\exp(\eta\beta_1\gamma) \leq \exp(2\eta\beta_1\gamma) \leq 2$ and $\gamma = \frac{1}{2}d_h$. The last inequality follows from $d_h = \widetilde{\Omega}(d^2) \geq \big(1728\log(4d(2d+1)/\delta) + (576d + 1872)\beta_1\log(4d(2d+1)/\delta)\big)/\beta_1$.

**Upper bound of $c_i^\top(T+1)c_i(T+1)$**

$$c_i^\top(T+1)c_i(T+1)$$

$$= \underbrace{c_i^\top(0)c_i(0)}_{\leq \frac{5d_h}{4}} + 2\eta \underbrace{\sum_{s=0}^{T} \big(\beta_3 - \beta_1 c_i^\top(s)b_i(s)\big)c_i^\top(s)b_i(s)}_{\text{term I}} - 2\eta\beta_1 \sum_{k\neq i}^{d} \underbrace{\sum_{s=0}^{T} \big(c_i^\top(s)b_k(s)\big)^2}_{=\text{term II}}$$

$$- 2\eta\beta_2 \underbrace{\sum_{s=0}^{T} \big(c_i^\top(s)b(s)\big)^2}_{\geq 0} + \eta^2 \underbrace{\sum_{s=0}^{T} \big\|\bar{c}_i(s)\big\|_2^2}_{\text{term V}}$$

$$\leq \frac{5d_h}{4} + 2\eta \cdot \frac{6\delta_{\max}^2 \exp(\eta\beta_1\gamma)}{\eta\beta_1\gamma} + 2\eta\beta_1(d-1) \cdot \frac{2\delta_{\max}^2 \exp(2\eta\beta_1\gamma)}{\eta\beta_1\gamma}$$

$$+ \eta^2 \cdot \Big(\frac{16d_h d^2 \delta_{\max}^2 \exp(\eta\beta_1\gamma)}{\eta\beta_1\gamma} + \frac{80\beta_2 d_h \delta_{\max}^2 \exp(\eta\beta_2\gamma)}{\eta\gamma}\Big)$$

$$\leq \frac{5d_h}{4} + \frac{2*12*9d_h \log(4d(2d+1)/\delta)}{\beta_1 \frac{1}{2}d_h} + \frac{2*4(d-1)*9d_h \log(4d(2d+1)/\delta)}{\frac{1}{2}d_h}$$

$$+ \frac{2*16\eta d_h d^2 * 9d_h \log(4d(2d+1)/\delta)}{\beta_1 \frac{1}{2}d_h} + \frac{2*80\eta\beta_2 d_h * 9d_h \log(4d(2d+1)/\delta)}{\frac{1}{2}d_h}$$

$$\leq 2d_h$$

The second inequality is by $\delta_{\max} = 3\sqrt{d_h \log(4d(2d+1)/\delta)}$, $\exp(\eta\beta_1\gamma) \leq \exp(2\eta\beta_1\gamma) \leq 2$ and $\gamma = \frac{1}{2}d_h$. The last inequality follows from

$$d_h = \widetilde{\Omega}(d^2)$$
$$\geq 576\log(4d(2d+1)/\delta)/\beta_1 + 192(d-1)\log(4d(2d+1)/\delta)$$
$$+ 384\log(4d(2d+1)/\delta)/\beta_1 + 3840\ln 2\log(4d(2d+1)/\delta)$$
$$\geq 576\log(4d(2d+1)/\delta)/\beta_1 + 192(d-1)\log(4d(2d+1)/\delta)$$
$$+ 768\eta d_h d^2 \log(4d(2d+1)/\delta)/\beta_1 + 3840\eta\beta_2 d_h \log(4d(2d+1)/\delta)$$

**Lower bound of $b^\top(T+1)b(T+1)$**

$$b^\top(T+1)b(T+1) = \underbrace{b^\top(0)b(0)}_{\geq \frac{3d_h}{4}} - 2\eta\beta_2 \sum_{k=1}^{d} \underbrace{\sum_{s=0}^{T} \big(c_k^\top(s)b(s)\big)^2}_{=\text{term IV}} + \eta^2 \underbrace{\sum_{s=0}^{T} \big\|\bar{b}(s)\big\|_2^2}_{\geq 0}$$

$$\geq \frac{3d_h}{4} - 2\eta\beta_2 d \cdot \frac{17\delta_{\max}^2}{\eta\beta_2\gamma}$$

$$\geq \frac{3d_h}{4} - \frac{34d\delta_{\max}^2}{\gamma}$$

$$\geq \frac{3d_h}{4} - \frac{34d*9d_h \log(4d(2d+1)/\delta)}{\frac{1}{2}d_h}$$

$$\geq \frac{d_h}{2}$$

The third inequality is by $\delta_{\max} = 3\sqrt{d_h \log(4d(2d+1)/\delta)}$ and $\gamma = \frac{1}{2}d_h$. The last inequality follows from $d_h = \widetilde{\Omega}(d^2) \geq 2448d\log(4d(2d+1)/\delta)$.

**Upper bound of $\boldsymbol{b}^\top(T+1)\boldsymbol{b}(T+1)$**

$$\boldsymbol{b}^\top(T+1)\boldsymbol{b}(T+1) = \underbrace{\boldsymbol{b}^\top(0)\boldsymbol{b}(0)}_{\leq \frac{5d_h}{4}} - 2\eta\beta_2 \sum_{k=1}^{d}\sum_{s=0}^{T} \underbrace{\left(\boldsymbol{c}_k^\top(s)\boldsymbol{b}(s)\right)^2}_{=\text{term IV}} + \eta^2 \sum_{s=0}^{T} \underbrace{\left\|\bar{\boldsymbol{b}}(s)\right\|_2^2}_{\text{term VI}}$$

$$\leq \frac{5d_h}{4} + 2\eta\beta_2 d \cdot \frac{17\delta_{\max}^2}{\eta\beta_2\gamma} + \eta^2 \cdot \left(\frac{16d_h\beta_2 d^2\delta_{\max}^2 \exp(\eta\beta_2\gamma)}{\eta\gamma} + \frac{2d_h d^2\delta_{\max}^2 \exp(\eta\beta_1\gamma)}{\eta\beta_1\gamma}\right)$$

$$\leq \frac{5d_h}{4} + \frac{34d\delta_{\max}^2}{\gamma} + \frac{2*16\eta d_h\beta_2 d^2\delta_{\max}^2}{\gamma} + \frac{2*2\eta d_h d^2\delta_{\max}^2}{\beta_1\gamma}$$

$$\leq \frac{5d_h}{4} + \frac{34d * 9d_h \log(4d(2d+1)/\delta)}{\frac{1}{2}d_h} + \frac{32\eta d_h\beta_2 d^2 * 9d_h \log(4d(2d+1)/\delta)}{\frac{1}{2}d_h}$$

$$+ \frac{4\eta d_h d^2 * 9d_h \log(4d(2d+1)/\delta)}{\beta_1\frac{1}{2}d_h}$$

$$\leq 2d_h$$

The second inequality is by $\exp(\eta\beta_1\gamma) \leq \exp(\eta\beta_2\gamma) \leq 2$ The third inequality is by $\delta_{\max} = 3\sqrt{d_h \log(4d(2d+1)/\delta)}$ and $\gamma = \frac{1}{2}d_h$. The last inequality follows from

$$d_h = \widetilde{\Omega}(d^2)$$

$$\geq 816d \log(4d(2d+1)/\delta) + 768 \ln 2 d^2 \log(4d(2d+1)/\delta) + 48 \log(4d(2d+1)/\delta)/\beta_1$$

$$\geq 816d \log(4d(2d+1)/\delta) + 768\eta\beta_2 d_h d^2 \log(4d(2d+1)/\delta) + 96\eta d_h d^2 \log(4d(2d+1)/\delta)/\beta_1$$

## C.5   Proof of claim B.2

This Section presents the exponential decay bounds for terms $\left(\beta_3 - \beta_1\boldsymbol{c}_i^\top(T+1)\boldsymbol{b}_i(T+1)\right)$, $\boldsymbol{c}_i^\top(T+1)\boldsymbol{b}_j(T+1)$ and $\boldsymbol{c}_i^\top(T+1)\boldsymbol{b}(T+1)$, establishing the property $\mathcal{B}(T+1)$.

**Bound of $\left(\beta_3 - \beta_1\boldsymbol{c}_i^\top(T+1)\boldsymbol{b}_i(T+1)\right)$**

Recall the following equation from lemma A.7.

$$\boldsymbol{c}_i^\top(t+1)\boldsymbol{b}_i(t+1) = \boldsymbol{c}_i^\top(t)\boldsymbol{b}_i(t) + \eta\Big(\left(\beta_3 - \beta_1\boldsymbol{c}_i^\top(t)\boldsymbol{b}_i(t)\right)\boldsymbol{b}_i^\top(t)\boldsymbol{b}_i(t) - \beta_1 \sum_{k\neq i}^{d} \boldsymbol{c}_i^\top(t)\boldsymbol{b}_k(t) \cdot \boldsymbol{b}_k^\top(t)\boldsymbol{b}_i(t)$$

$$- \beta_2\boldsymbol{c}_i^\top(t)\boldsymbol{b}(t) \cdot \boldsymbol{b}_i^\top(t)\boldsymbol{b}(t) + \left(\beta_3 - \beta_1\boldsymbol{c}_i^\top(t)\boldsymbol{b}_i(t)\right)\boldsymbol{c}_i^\top(t)\boldsymbol{c}_i(t)$$

$$- \beta_1 \sum_{k\neq i}^{d} \boldsymbol{c}_k^\top(t)\boldsymbol{b}_i(t) \cdot \boldsymbol{c}_i^\top(t)\boldsymbol{c}_k(t)\Big) + \eta^2\bar{\boldsymbol{c}}_i^\top(t)\bar{\boldsymbol{b}}_i(t)$$

Based on the above equation, we have:

$$\left|\left(\beta_3 - \beta_1\boldsymbol{c}_i^\top(T+1)\boldsymbol{b}_i(T+1)\right)\right| = \Big|\beta_3 - \beta_1\boldsymbol{c}_i^\top(T)\boldsymbol{b}_i(T)$$

$$- \eta\beta_1\Big(\left(\beta_3 - \beta_1\boldsymbol{c}_i^\top(T)\boldsymbol{b}_i(T)\right)\boldsymbol{b}_i^\top(T)\boldsymbol{b}_i(T) - \beta_1 \sum_{k\neq i}^{d} \boldsymbol{c}_i^\top(T)\boldsymbol{b}_k(T) \cdot \boldsymbol{b}_k^\top(T)\boldsymbol{b}_i(T)$$

$$- \beta_2\boldsymbol{c}_i^\top(T)\boldsymbol{b}(T) \cdot \boldsymbol{b}_i^\top(T)\boldsymbol{b}(T) + \left(\beta_3 - \beta_1\boldsymbol{c}_i^\top(T)\boldsymbol{b}_i(T)\right)\boldsymbol{c}_i^\top(T)\boldsymbol{c}_i(T)$$

$$- \beta_1 \sum_{k\neq i}^{d} \boldsymbol{c}_k^\top(T)\boldsymbol{b}_i(T) \cdot \boldsymbol{c}_i^\top(T)\boldsymbol{c}_k(T)\Big) - \eta^2\beta_1\bar{\boldsymbol{c}}_i^\top(T)\bar{\boldsymbol{b}}_i(T)\Big| \qquad (31)$$

$$= \Big|\left(1 - \eta\beta_1\left(\boldsymbol{b}_i^\top(T)\boldsymbol{b}_i(T) + \boldsymbol{c}_i^\top(T)\boldsymbol{c}_i(T)\right)\right)\left(\beta_3 - \beta_1\boldsymbol{c}_i^\top(T)\boldsymbol{b}_i(T)\right)$$

$$+ \eta\beta_1^2 \sum_{k\neq i}^{d} \boldsymbol{c}_i^\top(T)\boldsymbol{b}_k(T) \cdot \boldsymbol{b}_k^\top(T)\boldsymbol{b}_i(T) + \eta\beta_1^2 \sum_{k\neq i}^{d} \boldsymbol{c}_k^\top(T)\boldsymbol{b}_i(T) \cdot \boldsymbol{c}_i^\top(T)\boldsymbol{c}_k(T)$$

$$+ \eta\beta_1\beta_2\boldsymbol{c}_i^\top(T)\boldsymbol{b}(T) \cdot \boldsymbol{b}_i^\top(T)\boldsymbol{b}(T) - \eta^2\beta_1\bar{\boldsymbol{c}}_i^\top(T)\bar{\boldsymbol{b}}_i(T)\Big|$$

The term $\beta_3 - \beta_1 \boldsymbol{c}_i^\top(T)\boldsymbol{b}_i(T)$ is highlighted with underline, and we collect its *negative feedback* terms together. The factor $\left(1 - \eta\beta_1\left(\boldsymbol{b}_i^\top(T)\boldsymbol{b}_i(T) + \boldsymbol{c}_i^\top(T)\boldsymbol{c}_i(T)\right)\right) \leq 1$ will drive $\left(\beta_3 - \beta_1\boldsymbol{c}_i^\top(T + 1)\boldsymbol{b}_i(T + 1)\right)$ to converge to zero.

By Recurring (Eq. (31)) from $0$ to $T$, we have:

$$
\begin{aligned}
&\left|\left(\beta_3 - \beta_1\boldsymbol{c}_i^\top(T + 1)\boldsymbol{b}_i(T + 1)\right)\right| \\
&= \Big| \prod_{s=0}^{T} \left(1 - \eta\beta_1\left(\boldsymbol{b}_i^\top(s)\boldsymbol{b}_i(s) + \boldsymbol{c}_i^\top(s)\boldsymbol{c}_i(s)\right)\right)\left(\beta_3 - \beta_1\boldsymbol{c}_i^\top(0)\boldsymbol{b}_i(0)\right) \\
&\quad + \sum_{s=0}^{T}\prod_{s'=s+1}^{T} \left(1 - \eta\beta_1\left(\boldsymbol{b}_i^\top(s')\boldsymbol{b}_i(s') + \boldsymbol{c}_i^\top(s')\boldsymbol{c}_i(s')\right)\right) \\
&\quad \cdot \Big( \underbrace{\eta\beta_1^2\sum_{k\neq i}^{d}\boldsymbol{c}_i^\top(s)\boldsymbol{b}_k(s)\cdot\boldsymbol{b}_k^\top(s)\boldsymbol{b}_i(s) + \eta\beta_1^2\sum_{k\neq i}^{d}\boldsymbol{c}_k^\top(s)\boldsymbol{b}_i(s)\cdot\boldsymbol{c}_i^\top(s)\boldsymbol{c}_k(s)}_{\spadesuit} \\
&\quad + \underbrace{\eta\beta_1\beta_2\boldsymbol{c}_i^\top(s)\boldsymbol{b}(s)\cdot\boldsymbol{b}_i^\top(s)\boldsymbol{b}(s)}_{\clubsuit} - \underbrace{\eta^2\beta_1\bar{\boldsymbol{c}}_i^\top(s)\bar{\boldsymbol{b}}_i(s)}_{\diamondsuit}\Big)\Big|
\end{aligned}
\tag{32}
$$

Here $\prod_{s=0}^{T}\left(1 - \eta\beta_1\left(\boldsymbol{b}_i^\top(s)\boldsymbol{b}_i(s) + \boldsymbol{c}_i^\top(s)\boldsymbol{c}_i(s)\right)\right) \leq (1 - 2\eta\beta_1\gamma)^{T+1}$ since $\gamma \leq \boldsymbol{b}_i^\top(s)\boldsymbol{b}_i(s), \boldsymbol{c}_i^\top(s)\boldsymbol{c}_i(s)$. Besides, from property $\mathcal{B}(0),\ldots,\mathcal{B}(T)$ and lemma C.7 we know that $\boldsymbol{c}_i^\top(s)\boldsymbol{b}_k(s)$, $\boldsymbol{c}_i^\top(s)\boldsymbol{b}(s)$ and $\bar{\boldsymbol{c}}_i^\top(s)\bar{\boldsymbol{b}}_i(s)$ have bounds with exponential decreasing rate. Therefore, it is easy to derive an exponential decreasing upper bound for $\left|\left(\beta_3 - \beta_1\boldsymbol{c}_i^\top(T + 1)\boldsymbol{b}_i(T + 1)\right)\right|$.

By substituting the bounds of $\boldsymbol{c}_i^\top(s)\boldsymbol{b}_k(s)$, $\boldsymbol{c}_i^\top(s)\boldsymbol{b}(s)$, $\bar{\boldsymbol{c}}_i^\top(s)\bar{\boldsymbol{b}}_i(s)$, $\boldsymbol{b}_k^\top(s)\boldsymbol{b}_i(s)$, $\boldsymbol{c}_i^\top(s)\boldsymbol{c}_k(s)$ and $\boldsymbol{b}_i^\top(s)\boldsymbol{b}(s)$, we have:

$$
\begin{aligned}
&\left|\left(\beta_3 - \beta_1\boldsymbol{c}_i^\top(T + 1)\boldsymbol{b}_i(T + 1)\right)\right| \\
&\leq (1 - 2\eta\beta_1\gamma)^{T+1}\left|\beta_3 - \beta_1\boldsymbol{c}_i^\top(0)\boldsymbol{b}_i(0)\right| \\
&\quad + \sum_{s=0}^{T}(1 - 2\eta\beta_1\gamma)^{T-s}\cdot\Big( \underbrace{2\eta\beta_1^2(d - 1)\cdot 2\delta(s)^2\exp(-\eta\beta_1\gamma s)}_{\spadesuit} \\
&\quad + \underbrace{\eta\beta_1\beta_2\cdot\left(2\delta(s)^2\exp(-\eta\beta_2\gamma s) + \frac{\delta(s)^2}{\beta_2}\exp(-\eta\beta_1\gamma s)\right)}_{\clubsuit} \\
&\quad + \underbrace{\eta^2\beta_1\left(8d_h d^2\delta(t)^2\exp(-\eta\beta_1\gamma t) + 8\beta_2 d_h d\delta(t)^2\exp(-\eta\beta_2\gamma t)\right)}_{\diamondsuit}\Big)
\end{aligned}
\tag{33}
$$

The notations $\spadesuit$, $\clubsuit$ and $\diamondsuit$ highlight the corresponding terms between (Eq. (32)) and (Eq. (33)) for refference.

We further have the following:

$$\left| \left( \beta_3 - \beta_1 \boldsymbol{c}_i^\top (T+1) \boldsymbol{b}_i(T+1) \right) \right|$$

$$\leq (1 - 2\eta\beta_1\gamma)^{T+1} \left| \beta_3 - \beta_1 \boldsymbol{c}_i^\top(0)\boldsymbol{b}_i(0) \right|$$

$$+ \sum_{s=0}^{T} (1 - 2\eta\beta_1\gamma)^{T-s} \cdot \Big( \underbrace{2\eta\beta_1^2(d-1) \cdot 2\delta(s)^2 \exp(-\eta\beta_1\gamma s)}_{\spadesuit}$$

$$+ \underbrace{\eta\beta_1\beta_2 \cdot \left( 2\delta(s)^2 \exp(-\eta\beta_2\gamma s) + \frac{\delta(s)^2}{\beta_2} \exp(-\eta\beta_1\gamma s) \right)}_{\clubsuit}$$

$$+ \underbrace{\eta^2\beta_1 \left( 8d_h d^2 \delta(t)^2 \exp(-\eta\beta_1\gamma t) + 8\beta_2 d_h d\delta(t)^2 \exp(-\eta\beta_2\gamma t) \right)}_{\diamond} \Big)$$

$$\leq \exp(-2\eta\beta_1\gamma(T+1)) \left| \beta_3 - \beta_1 \boldsymbol{c}_i^\top(0)\boldsymbol{b}_i(0) \right|$$

$$+ \sum_{s=0}^{T} \exp(2\eta\beta_1\gamma(s-T)) \cdot \Big( \left( 4\eta\beta_1^2(d-1)\delta(s)^2 + \eta\beta_1\delta(s)^2 + 8\eta^2\beta_1 d_h d^2 \delta(t)^2 \right) \exp(-\eta\beta_1\gamma s)$$

$$+ \left( 2\eta\beta_1\beta_2\delta(s)^2 + 8\eta^2\beta_1\beta_2 d_h d\delta(s)^2 \right) \exp(-\eta\beta_2\gamma s) \Big)$$

$$\leq (\beta_3 + \beta_1\delta(0)) \exp(-2\eta\beta_1\gamma(T+1))$$

$$+ \left( 4\eta\beta_1^2(d-1)\delta(T)^2 + \eta\beta_1\delta(T)^2 + 8\eta^2\beta_1 d_h d^2 \delta(T)^2 \right) \cdot \frac{2}{\eta\beta_1\gamma} \exp(-\eta\beta_1\gamma(T+1))$$

$$+ \left( 2\eta\beta_1\beta_2\delta(T)^2 + 8\eta^2\beta_1\beta_2 d_h d\delta(T)^2 \right) \cdot \frac{3}{\eta\beta_2\gamma} \exp(-\eta\beta_1\gamma(T+1))$$

$$\leq \left( \frac{\beta_3}{\delta(0)} + \beta_1 + \frac{8\beta_1(d-1)\delta(T)}{\gamma} + \frac{2\delta(T)}{\gamma} + \frac{16\eta d_h d^2 \delta(T)}{\gamma} + \frac{6\beta_1\delta(T)}{\gamma} + \frac{24\eta\beta_1 d_h d\delta(T)}{\gamma} \right)$$

$$\cdot \delta(T) \cdot \exp(-\eta\beta_1\gamma(T+1))$$

$$\leq \delta(T) \exp(-\eta\beta_1\gamma(T+1))$$

$$(34)$$

The second inequality is derived by factoring out the factors $\exp(-\eta\beta_1\gamma s)$ and $\exp(-\eta\beta_2\gamma s)$. The third inequality is due to $\sum_{s=0}^{T} \exp(2\eta\beta_1\gamma(s-T)) \cdot \exp(-\eta\beta_1\gamma s) \leq \frac{2}{\eta\beta_1\gamma} \exp(-\eta\beta_1\gamma(T+1))$ and $\sum_{s=0}^{T} \exp(2\eta\beta_1\gamma(s-T)) \cdot \exp(-\eta\beta_2\gamma s) \leq \frac{3}{\eta\beta_2\gamma} \exp(-\eta\beta_1\gamma(T+1))$ in lemma C.5. The fourth inequality is by $\delta(0) \leq \delta(T)$, $\exp(-2\eta\beta_1\gamma(T+1)) \leq \exp(-\eta\beta_1\gamma(T+1))$, and we consider $\beta_3 = \frac{\beta_3}{\delta(0)} \cdot \delta(0) \leq \frac{\beta_3}{\delta(0)} \cdot \delta(T)$. The fifth inequality is by proving $\left( \frac{\beta_3}{\delta(0)} + \beta_1 + \frac{8\beta_1(d-1)\delta(T)}{\gamma} + \frac{2\delta(T)}{\gamma} + \frac{16\eta d_h d^2 \delta(T)}{\gamma} + \frac{6\beta_1\delta(T)}{\gamma} + \frac{24\eta\beta_1 d_h d\delta(T)}{\gamma} \right) \leq 1$ as follows:

$$\frac{\beta_3}{\delta(0)} + \beta_1 + \frac{8\beta_1(d-1)\delta(T)}{\gamma} + \frac{2\delta(T)}{\gamma} + \frac{16\eta d_h d^2 \delta(T)}{\gamma} + \frac{6\beta_1\delta(T)}{\gamma} + \frac{24\eta\beta_1 d_h d\delta(T)}{\gamma}$$

$$\leq \frac{\beta_3}{\delta(0)} + \frac{3}{4} + \frac{\delta(T)}{\gamma} \cdot \left( 8\beta_1(d-1) + 2 + 16\eta d_h d^2 + 6\beta_1 + 24\eta\beta_1 d_h d \right)$$

$$\leq \frac{\beta_3}{2\sqrt{d_h \log(4d(2d+1)/\delta)}} + \frac{3}{4}$$

$$+ \frac{3\sqrt{d_h \log(4d(2d+1)/\delta)}}{\frac{1}{2}d_h} \cdot \left( 8\beta_1(d-1) + 2 + 16\eta d_h d^2 + 6\beta_1 + 24\eta\beta_1 d_h d \right)$$

$$\leq 1$$

The first inequality is by $\beta_1 \leq \frac{3}{4}$. The second inequality is by $\delta(0) \geq 2\sqrt{d_h \log(4d(2d+1)/\delta)}$, $\delta(T) \leq 3\sqrt{d_h \log(4d(2d+1)/\delta)}$ and $\gamma = \frac{1}{2}d_h$. The last inequality hold as long as $d_h =$

$$\widetilde{\Omega}(d^2) \geq \left(\frac{1}{\sqrt{\log(4d(2d+1)/\delta)}} + 24\sqrt{\log(4d(2d+1)/\delta)}\big(8\beta_1(d-1)+2+8+6\beta_1+12\eta\beta_1/d\big)\right)^2 \geq$$

$$\left(\frac{1}{\sqrt{\log(4d(2d+1)/\delta)}} + 24\sqrt{\log(4d(2d+1)/\delta)}\big(8\beta_1(d-1)+2+16\eta d_h d^2+6\beta_1+24\eta\beta_1 d_h d\big)\right)^2.$$

Therefore, we have

$$\left|\big(\beta_3 - \beta_1 \boldsymbol{c}_i^\top(T+1)\boldsymbol{b}_i(T+1)\big)\right| \leq \delta(T)\exp(-\eta\beta_1\gamma(T+1)) \leq \delta(T+1)\exp(-\eta\beta_1\gamma(T+1)) \quad (35)$$

where the last inequality is by $\delta(T) \leq \delta(T+1)$.

The proof for the bounds of $\boldsymbol{c}_i^\top(T+1)\boldsymbol{b}_j(T+1)$ and $\boldsymbol{c}_i^\top(T+1)\boldsymbol{b}(T+1)$ are similar to that of $\big(\beta_3 - \beta_1\boldsymbol{c}_i^\top(T+1)\boldsymbol{b}_i(T+1)\big)$. We presents the calculation as follows.

**Bound of $\boldsymbol{c}_i^\top(T+1)\boldsymbol{b}_j(T+1)$**

$$\left|\boldsymbol{c}_i^\top(T+1)\boldsymbol{b}_j(T+1)\right|$$

$$= \left|\boldsymbol{c}_i^\top(T)\boldsymbol{b}_j(T) + \eta\Big(\big(\beta_3 - \beta_1\boldsymbol{c}_i^\top(T)\boldsymbol{b}_i(T)\big)\boldsymbol{b}_i^\top(T)\boldsymbol{b}_j(T) - \beta_1\sum_{k\neq i}^d \boldsymbol{c}_i^\top(T)\boldsymbol{b}_k(T)\cdot\boldsymbol{b}_k^\top(T)\boldsymbol{b}_j(T)\right.$$

$$- \beta_2\boldsymbol{c}_i^\top(T)\boldsymbol{b}(T)\cdot\boldsymbol{b}_j^\top(T)\boldsymbol{b}(T) + \big(\beta_3 - \beta_1\boldsymbol{c}_j^\top(T)\boldsymbol{b}_j(T)\big)\boldsymbol{c}_i^\top(T)\boldsymbol{c}_j(T)$$

$$\left.- \beta_1\sum_{k\neq j}^d \boldsymbol{c}_k^\top(T)\boldsymbol{b}_j(T)\cdot\boldsymbol{c}_i^\top(T)\boldsymbol{c}_k(T)\Big) + \eta^2\bar{\boldsymbol{c}}_i^\top(T)\bar{\boldsymbol{b}}_j(T)\right|$$

$$= \left|\Big(1 - \eta\beta_1\big(\boldsymbol{c}_i^\top(T)\boldsymbol{c}_i(T) + \boldsymbol{b}_j^\top(T)\boldsymbol{b}_j(T)\big)\Big)\boldsymbol{c}_i^\top(T)\boldsymbol{b}_j(T)\right.$$

$$+ \eta\big(\beta_3 - \beta_1\boldsymbol{c}_i^\top(T)\boldsymbol{b}_i(T)\big)\boldsymbol{b}_i^\top(T)\boldsymbol{b}_j(T) - \eta\beta_1\sum_{k\neq i,k\neq j}^d \boldsymbol{c}_i^\top(T)\boldsymbol{b}_k(T)\cdot\boldsymbol{b}_k^\top(T)\boldsymbol{b}_j(T)$$

$$- \eta\beta_2\boldsymbol{c}_i^\top(T)\boldsymbol{b}(T)\cdot\boldsymbol{b}_j^\top(T)\boldsymbol{b}(T) + \eta\big(\beta_3 - \beta_1\boldsymbol{c}_j^\top(T)\boldsymbol{b}_j(T)\big)\boldsymbol{c}_i^\top(T)\boldsymbol{c}_j(T)$$

$$\left.- \eta\beta_1\sum_{k\neq i,k\neq j}^d \boldsymbol{c}_k^\top(T)\boldsymbol{b}_j(T)\cdot\boldsymbol{c}_i^\top(T)\boldsymbol{c}_k(T) + \eta^2\bar{\boldsymbol{c}}_i^\top(T)\bar{\boldsymbol{b}}_j(T)\right|$$

$$= \left|\prod_{s=0}^T \Big(1 - \eta\beta_1\big(\boldsymbol{c}_i^\top(s)\boldsymbol{c}_i(s) + \boldsymbol{b}_j^\top(s)\boldsymbol{b}_j(s)\big)\Big)\boldsymbol{c}_i^\top(0)\boldsymbol{b}_j(0)\right.$$

$$+ \sum_{s=0}^T \prod_{s'=s+1}^T \Big(1 - \eta\beta_1\big(\boldsymbol{c}_i^\top(s')\boldsymbol{c}_i(s') + \boldsymbol{b}_j^\top(s')\boldsymbol{b}_j(s')\big)\Big)$$

$$\cdot \Big(\underbrace{\eta\big(\beta_3 - \beta_1\boldsymbol{c}_i^\top(s)\boldsymbol{b}_i(s)\big)\boldsymbol{b}_i^\top(s)\boldsymbol{b}_j(s) + \eta\big(\beta_3 - \beta_1\boldsymbol{c}_j^\top(s)\boldsymbol{b}_j(s)\big)\boldsymbol{c}_i^\top(s)\boldsymbol{c}_j(s)}_{\spadesuit}$$

$$\underbrace{- \eta\beta_1\sum_{k\neq i,k\neq j}^d \boldsymbol{c}_i^\top(s)\boldsymbol{b}_k(s)\cdot\boldsymbol{b}_k^\top(s)\boldsymbol{b}_j(s) - \eta\beta_1\sum_{k\neq i,k\neq j}^d \boldsymbol{c}_k^\top(s)\boldsymbol{b}_j(s)\cdot\boldsymbol{c}_i^\top(s)\boldsymbol{c}_k(s)}_{\clubsuit}$$

$$\left.\underbrace{- \eta\beta_2\boldsymbol{c}_i^\top(s)\boldsymbol{b}(s)\cdot\boldsymbol{b}_j^\top(s)\boldsymbol{b}(s)}_{\diamondsuit} + \underbrace{\eta^2\bar{\boldsymbol{c}}_i^\top(s)\bar{\boldsymbol{b}}_j(s)}_{\heartsuit}\Big)\right|$$

$$\leq (1 - 2\eta\beta_1\gamma)^{T+1}\left|\boldsymbol{c}_i^\top(0)\boldsymbol{b}_j(0)\right|$$

$$+ \sum_{s=0}^T (1 - 2\eta\beta_1\gamma)^{T-s}\Big(\underbrace{2\eta\delta(s)^2\exp(-\eta\beta_1\gamma s)}_{\spadesuit}$$

$$+ \underbrace{2\eta\beta_1(d-2)\cdot 2\delta(s)^2\exp(-\eta\beta_1\gamma s)}_{\clubsuit}$$

$$+ \eta\beta_2 \cdot \underbrace{\left(2\delta(s)^2 \exp(-\eta\beta_2\gamma s) + \frac{\delta(s)^2}{\beta_2} \exp(-\eta\beta_1\gamma s)\right)}_{\diamondsuit}$$

$$+ \underbrace{\eta^2 \left(8d_h d^2 \delta(t)^2 \exp(-\eta\beta_1\gamma t) + 8\beta_2 d_h d\delta(t)^2 \exp(-\eta\beta_2\gamma t)\right)}_{\heartsuit} \Big)$$

$$\leq \exp(-2\eta\beta_1\gamma(T+1)) \left| \boldsymbol{c}_i^\top(0)\boldsymbol{b}_j(0) \right|$$

$$+ \sum_{s=0}^{T} \exp(2\eta\beta_1\gamma(s-T)) \Big( \big(2\eta\delta(s)^2 + 4\eta\beta_1(d-2)\delta(s)^2 + \eta\delta(s)^2 + 8\eta^2 d_h d^2 \delta(s)^2 \big) \exp(-\eta\beta_1\gamma s)$$

$$+ \big(2\eta\beta_2\delta(s)^2 + 8\eta^2\beta_2 d_h d\delta(s)^2\big)\exp(-\eta\beta_2\gamma s) \Big)$$

$$\leq \delta(T)\exp(-\eta\beta_1\gamma(T+1))$$

$$+ \left(2\eta\delta(T)^2 + 4\eta\beta_1(d-2)\delta(T)^2 + \eta\delta(T)^2 + 8\eta^2 d_h d^2 \delta(T)^2\right) \cdot \frac{2}{\eta\beta_1\gamma} \exp(-\eta\beta_1\gamma(T+1))$$

$$+ \left(2\eta\beta_2\delta(T)^2 + 8\eta^2\beta_2 d_h d\delta(T)^2\right) \cdot \frac{3}{\eta\beta_2\gamma} \exp(-\eta\beta_1\gamma(T+1))$$

$$= \left(1 + \frac{4\delta(T)}{\beta_1\gamma} + \frac{8(d-2)\delta(T)}{\gamma} + \frac{2\delta(T)}{\beta_1\gamma} + \frac{16\eta d_h d^2\delta(T)}{\beta_1\gamma} + \frac{6\delta(T)}{\gamma} + \frac{24\eta d_h d\delta(T)}{\gamma}\right)$$

$$\cdot \delta(T)\exp(-\eta\beta_1\gamma(T+1))$$

$$\leq 2\delta(T)\exp(-\eta\beta_1\gamma(T+1))$$

$$\leq 2\delta(T+1)\exp(-\eta\beta_1\gamma(T+1)) \tag{36}$$

This bound requires $\frac{\delta(T)}{\gamma}\left(\frac{4}{\beta_1} + 8(d-2) + \frac{2}{\beta_1} + \frac{16\eta d_h d^2}{\beta_1} + 6 + 24\eta d_h d\right) \leq 1$, which can be verified by $d_h = \widetilde{\Omega}(d^2) \geq 36\log(4d(2d+1)/\delta)\left(\frac{4}{\beta_1} + 8(d-2) + \frac{2}{\beta_1} + \frac{2}{\beta_1} + 6 + \frac{12}{d}\right)^2 \geq 36\log(4d(2d+1)/\delta)\left(\frac{4}{\beta_1} + 8(d-2) + \frac{2}{\beta_1} + \frac{16\eta d_h d^2}{\beta_1} + 6 + 24\eta d_h d\right)^2$.

**Bound of $\boldsymbol{c}_i^\top(T+1)\boldsymbol{b}(T+1)$**

$$\left| \boldsymbol{c}_i^\top(T+1)\boldsymbol{b}(T+1) \right|$$

$$= \Big| \boldsymbol{c}_i^\top(T)\boldsymbol{b}(T) + \eta\Big(\big(\beta_3 - \beta_1\boldsymbol{c}_i^\top(T)\boldsymbol{b}_i(T)\big)\boldsymbol{b}_i^\top(T)\boldsymbol{b}(T) - \beta_1 \sum_{k\neq i}^{d} \boldsymbol{c}_i^\top(T)\boldsymbol{b}_k(T) \cdot \boldsymbol{b}_k^\top(T)\boldsymbol{b}(T)$$

$$- \beta_2 \boldsymbol{c}_i^\top(T)\boldsymbol{b}(T) \cdot \boldsymbol{b}^\top(T)\boldsymbol{b}(T) - \beta_2 \sum_{k=1}^{d} \boldsymbol{c}_k^\top(T)\boldsymbol{b}(T) \cdot \boldsymbol{c}_k^\top(T)\boldsymbol{c}_i(T)\Big) + \eta^2 \bar{\boldsymbol{c}}_i^\top(T)\bar{\boldsymbol{b}}(T) \Big|$$

$$= \Big| \Big(1 - \eta\beta_2\big(\boldsymbol{b}^\top(T)\boldsymbol{b}(T) + \boldsymbol{c}_i^\top(T)\boldsymbol{c}_i(T)\big)\Big)\boldsymbol{c}_i^\top(T)\boldsymbol{b}(T)$$

$$+ \eta\big(\beta_3 - \beta_1\boldsymbol{c}_i^\top(T)\boldsymbol{b}_i(T)\big)\boldsymbol{b}_i^\top(T)\boldsymbol{b}(T) - \eta\beta_1 \sum_{k\neq i}^{d} \boldsymbol{c}_i^\top(T)\boldsymbol{b}_k(T) \cdot \boldsymbol{b}_k^\top(T)\boldsymbol{b}(T)$$

$$- \eta\beta_2 \sum_{k\neq i}^{d} \boldsymbol{c}_k^\top(T)\boldsymbol{b}(T) \cdot \boldsymbol{c}_k^\top(T)\boldsymbol{c}_i(T) + \eta^2 \bar{\boldsymbol{c}}_i^\top(T)\bar{\boldsymbol{b}}(T) \Big|$$

$$= \Big| \prod_{s=0}^{T}\Big(1 - \eta\beta_2\big(\boldsymbol{b}^\top(s)\boldsymbol{b}(s) + \boldsymbol{c}_i^\top(s)\boldsymbol{c}_i(s)\big)\Big)\boldsymbol{c}_i^\top(0)\boldsymbol{b}(0)$$

$$+ \sum_{s=0}^{T}\prod_{s'=s+1}^{T}\Big(1 - \eta\beta_2\big(\boldsymbol{b}^\top(s')\boldsymbol{b}(s') + \boldsymbol{c}_i^\top(s')\boldsymbol{c}_i(s')\big)\Big)$$

$$\cdot \Big( \underbrace{\eta\big(\beta_3 - \beta_1 \boldsymbol{c}_i^\top(s)\boldsymbol{b}_i(s)\big)\boldsymbol{b}_i^\top(s)\boldsymbol{b}(s)}_{\spadesuit} - \underbrace{\eta\beta_1 \sum_{k\neq i}^{d} \boldsymbol{c}_i^\top(s)\boldsymbol{b}_k(s) \cdot \boldsymbol{b}_k^\top(s)\boldsymbol{b}(s)}_{\clubsuit}$$

$$- \underbrace{\eta\beta_2 \sum_{k\neq i}^{d} \boldsymbol{c}_k^\top(s)\boldsymbol{b}(s) \cdot \boldsymbol{c}_k^\top(s)\boldsymbol{c}_i(s)}_{\diamondsuit} + \underbrace{\eta^2 \bar{\boldsymbol{c}}_i^\top(s)\bar{\boldsymbol{b}}(s)}_{\heartsuit} \Big) \Big|$$

$$\leq (1 - 2\eta\beta_2\gamma)^{T+1} \big| \boldsymbol{c}_i^\top(0)\boldsymbol{b}(0) \big|$$

$$+ \sum_{s=0}^{T} (1 - 2\eta\beta_2\gamma)^{T-s} \Big( \underbrace{\eta\delta(s)^2 \exp(-\eta\beta_1\gamma s)}_{\spadesuit}$$

$$+ \underbrace{\eta\beta_1(d-1) \cdot 2\delta(s)^2 \exp(-\eta\beta_1\gamma s)}_{\clubsuit}$$

$$+ \underbrace{\eta\beta_2(d-1) \cdot \big( 2\delta(s)^2 \exp(-\eta\beta_2\gamma s) + \frac{\delta(s)^2}{\beta_2} \exp(-\eta\beta_1\gamma s)\big)}_{\diamondsuit}$$

$$+ \underbrace{\eta^2 \big( 4 d_h d^2 \delta(s)^2 \exp(-\eta\beta_1\gamma s) + 28\beta_2^2 d_h d\delta(s)^2 \exp(-\eta\beta_2\gamma s)\big)}_{\heartsuit} \Big)$$

$$\leq \exp(-2\eta\beta_2\gamma(T+1)) \big| \boldsymbol{c}_i^\top(0)\boldsymbol{b}(0) \big|$$

$$+ \sum_{s=0}^{T} \exp(2\eta\beta_2\gamma(s-T)) \Big( \big( \eta\delta(T)^2 + 2\eta\beta_1(d-1)\delta(T)^2$$

$$+ \eta(d-1)\delta(T)^2 + 4\eta^2 d_h d^2 \delta(T)^2 \big) \exp(-\eta\beta_1\gamma s)$$

$$+ \big( 2\eta\beta_2(d-1)\delta(T)^2 + 28\eta^2\beta_2^2 d_h d\delta(s)^2 \big) \exp(-\eta\beta_2\gamma s) \Big)$$

$$\leq \delta(T) \exp(-\eta\beta_2\gamma(T+1))$$

$$+ \Big( \eta\delta(T)^2 + 2\eta\beta_1(d-1)\delta(T)^2 + \eta(d-1)\delta(T)^2 + 4\eta^2 d_h d^2 \delta(T)^2 \Big) \cdot \frac{2}{\eta\beta_2\gamma} \exp(-\eta\beta_1\gamma(T+1))$$

$$+ \Big( 2\eta\beta_2(d-1)\delta(T)^2 + 28\eta^2\beta_2^2 d_h d\delta(T)^2 \Big) \cdot \frac{2}{\eta\beta_2\gamma} \exp(-\eta\beta_2\gamma(T+1))$$

$$= \delta(T) \Big( 1 + \frac{4(d-1)\delta(T)}{\gamma} + \frac{56\eta\beta_2 d_h d\delta(T)}{\gamma} \Big) \exp(-\eta\beta_2\gamma(T+1))$$

$$+ \frac{\delta(T)}{\beta_2} \Big( \frac{2\delta(T)}{\gamma} + \frac{4\beta_1(d-1)\delta(T)}{\gamma} + \frac{2(d-1)\delta(T)}{\gamma} + \frac{8\eta d_h d^2 \delta(T)}{\gamma} \Big) \exp(-\eta\beta_1\gamma(T+1))$$

$$\leq 2\delta(T) \exp(-\eta\beta_2\gamma(T+1)) + \frac{\delta(T)}{\beta_2} \exp(-\eta\beta_1\gamma(T+1))$$

$$\leq 2\delta(T+1) \exp(-\eta\beta_2\gamma(T+1)) + \frac{\delta(T+1)}{\beta_2} \exp(-\eta\beta_1\gamma(T+1)) \tag{37}$$

This bound requires $\frac{\delta(T)}{\gamma}\Big( 4(d-1) + 56\eta\beta_2 d_h d\Big) \leq 1$ and $\frac{\delta(T)}{\gamma}\Big( 2 + 4\beta_1(d-1) + 2(d-1) + 8\eta d_h d^2\Big) \leq 1$, which can be verified by $d_h = \widetilde{\Omega}(d^2) \geq 36\log(4d(2d+1)/\delta)\Big( 4(d-1) + 56d\ln 2\Big)^2 \geq 36\log(4d(2d+1)/\delta)\Big( 4(d-1) + 56\eta\beta_2 d_h d\Big)^2$

and

$d_h = \widetilde{\Omega}(d^2) \geq 36\log(4d(2d+1)/\delta)\Big( 2 + 4\beta_1(d-1) + 2(d-1) + 4\Big)^2 \geq 36\log(4d(2d+1)/\delta)\Big( 2 + 4\beta_1(d-1) + 2(d-1) + 8\eta d_h d^2\Big)^2.$

Property $\mathcal{B}(T+1)$ is established by (Eq. (35)), (Eq. (36)) and (Eq. (37)).

### C.5.1 Auxiliary lemma

**Lemma C.5** *As long as $2\eta\beta_1\gamma \leq \ln 2$, and $2\eta\beta_2\gamma \leq \ln 2$, we have*

$$\sum_{s=0}^{T} \exp(2\eta\beta_1\gamma(s-T)) \cdot \exp(-\eta\beta_1\gamma s) \leq \frac{2}{\eta\beta_1\gamma} \exp(-\eta\beta_1\gamma(T+1))$$

$$\sum_{s=0}^{T} \exp(2\eta\beta_1\gamma(s-T)) \cdot \exp(-\eta\beta_2\gamma s) \leq \frac{3}{\eta\beta_2\gamma} \exp(-\eta\beta_1\gamma(T+1))$$

$$\sum_{s=0}^{T} \exp(2\eta\beta_2\gamma(s-T)) \cdot \exp(-\eta\beta_1\gamma s) \leq \frac{2}{\eta\beta_2\gamma} \exp(-\eta\beta_1\gamma(T+1))$$

$$\sum_{s=0}^{T} \exp(2\eta\beta_2\gamma(s-T)) \cdot \exp(-\eta\beta_2\gamma s) \leq \frac{2}{\eta\beta_2\gamma} \exp(-\eta\beta_2\gamma(T+1))$$

Proof of lemma C.5.

$$\sum_{s=0}^{T} \exp(2\eta\beta_1\gamma(s-T)) \cdot \exp(-\eta\beta_1\gamma s)$$

$$= \sum_{s=0}^{T} \exp(\eta\beta_1\gamma s - 2\eta\beta_1\gamma T)$$

$$\leq \int_{0}^{T+1} \exp(\eta\beta_1\gamma s - 2\eta\beta_1\gamma T)ds$$

$$\leq \frac{1}{\eta\beta_1\gamma} \big( \exp(-\eta\beta_1\gamma(T-1)) - \exp(-2\eta\beta_1\gamma T) \big)$$

$$\leq \frac{1}{\eta\beta_1\gamma} \exp(-\eta\beta_1\gamma(T-1))$$

$$\leq \frac{\exp(2\eta\beta_1\gamma)}{\eta\beta_1\gamma} \exp(-\eta\beta_1\gamma(T+1))$$

$$\leq \frac{2}{\eta\beta_1\gamma} \exp(-\eta\beta_1\gamma(T+1))$$

The first inequality is due to $\exp(\eta\beta_1\gamma s)$ is monotone increasing. The last inequality is due to $2\eta\beta_1\gamma \leq \ln 2$.

$$\sum_{s=0}^{T} \exp(2\eta\beta_1\gamma(s-T)) \cdot \exp(-\eta\beta_2\gamma s)$$

$$= \sum_{s=0}^{T} \exp(-\eta(\beta_2 - 2\beta_1)\gamma s - 2\eta\beta_1\gamma T)$$

$$\leq \int_{-1}^{T} \exp(-\eta(\beta_2 - 2\beta_1)\gamma s - 2\eta\beta_1\gamma T)ds$$

$$\leq \frac{1}{\eta(\beta_2 - 2\beta_1)\gamma} \big( \exp(\eta(\beta_2 - 2\beta_1)\gamma - 2\eta\beta_1\gamma T) - \exp(-\eta\beta_2\gamma T) \big)$$

$$\leq \frac{1}{\eta(\beta_2 - 2\beta_1)\gamma} \exp(\eta(\beta_2 - 2\beta_1)\gamma - 2\eta\beta_1\gamma T)$$

$$\leq \frac{\exp(\eta\beta_2\gamma)}{\eta(\beta_2 - 2\beta_1)\gamma} \exp(-2\eta\beta_1\gamma(T+1))$$

$$\leq \frac{2\exp(\eta\beta_2\gamma)}{\eta\beta_2\gamma}\exp(-2\eta\beta_1\gamma(T+1))$$

$$\leq \frac{3}{\eta\beta_2\gamma}\exp(-\eta\beta_1\gamma(T+1))$$

The first inequality is due to $\exp(-\eta(\beta_2 - 2\beta_1)\gamma s)$ is monotone decreasing. The fourth inequality is due to $\beta_2 \geq 4\beta_1$, thus $\frac{1}{\beta_2-2\beta_1} \leq \frac{2}{\beta_2}$. The last inequality is due to $\eta\beta_2\gamma \leq (\ln 2)/2 \leq \ln(3/2)$.

$$\sum_{s=0}^{T}\exp(2\eta\beta_2\gamma(s-T))\cdot\exp(-\eta\beta_1\gamma s)$$

$$=\sum_{s=0}^{T}\exp(\eta(2\beta_2-\beta_1)\gamma s - 2\eta\beta_2\gamma T)$$

$$\leq \int_0^{T+1}\exp(\eta(2\beta_2-\beta_1)\gamma s - 2\eta\beta_2\gamma T)ds$$

$$\leq \frac{1}{\eta(2\beta_2-\beta_1)\gamma}\big(\exp(2\eta\beta_2\gamma - \eta\beta_1\gamma(T+1)) - \exp(-2\eta\beta_2\gamma T)\big)$$

$$\leq \frac{1}{\eta(2\beta_2-\beta_1)\gamma}\exp(2\eta\beta_2\gamma - \eta\beta_1\gamma(T+1))$$

$$\leq \frac{\exp(2\eta\beta_2\gamma)}{\eta(2\beta_2-\beta_1)\gamma}\exp(-\eta\beta_1\gamma(T+1))$$

$$\leq \frac{2}{\eta\beta_2\gamma}\exp(-\eta\beta_1\gamma(T+1))$$

The first inequality is due to $\exp(\eta(2\beta_2 - \beta_1)\gamma s)$ is monotone increasing. The last inequality is due to $\beta_2 \geq \beta_1$ and $2\eta\beta_2\gamma \leq \ln 2$.

The proof of $\sum_{s=0}^{T}\exp(2\eta\beta_2\gamma(s-T))\cdot\exp(-\eta\beta_2\gamma s) \leq \frac{2}{\eta\beta_2\gamma}\exp(-\eta\beta_2\gamma(T+1))$ is similar to $\sum_{s=0}^{T}\exp(2\eta\beta_1\gamma(s-T))\cdot\exp(-\eta\beta_1\gamma s) \leq \frac{2}{\eta\beta_1\gamma}\exp(-\eta\beta_1\gamma(T+1))$. Just replace $\beta_1$ with $\beta_2$, and consider $2\eta\beta_2\gamma \leq \ln 2$.

### C.6   Proof of claim B.3

This Section presents the bounds for terms $\boldsymbol{b}_i^\top(T+1)\boldsymbol{b}_j(T+1)$, $\boldsymbol{c}_i^\top(T+1)\boldsymbol{c}_j(T+1)$ and $\boldsymbol{b}_i^\top(T+1)\boldsymbol{b}(T+1)$ with $i,j \in [1,d], i \neq j$, establishing the property $\mathcal{C}(T+1)$.

Recall the *Vector-coupled Dynamics* equations of $\boldsymbol{b}_i^\top(t+1)\boldsymbol{b}_j(t+1)$, $\boldsymbol{c}_i^\top(t+1)\boldsymbol{c}_j(t+1)$ and $\boldsymbol{b}_i^\top(t+1)\boldsymbol{b}(t+1)$ in lemma A.7:

$$\boldsymbol{b}_i^\top(t+1)\boldsymbol{b}_j(t+1)$$
$$= \boldsymbol{b}_i^\top(t)\boldsymbol{b}_j(t) + \eta\Big(2\big(\beta_3 - \beta_1\boldsymbol{c}_i^\top(t)\boldsymbol{b}_i(t)\big)\boldsymbol{c}_i^\top(t)\boldsymbol{b}_j(t) + 2\big(\beta_3 - \beta_1\boldsymbol{c}_j^\top(t)\boldsymbol{b}_j(t)\big)\boldsymbol{c}_j^\top(t)\boldsymbol{b}_i(t)$$
$$- \beta_3\big(\boldsymbol{c}_i^\top(t)\boldsymbol{b}_j(t) + \boldsymbol{c}_j^\top(t)\boldsymbol{b}_i(t)\big) - 2\beta_1\sum_{k\neq i,k\neq j}^{d}\boldsymbol{c}_k^\top(t)\boldsymbol{b}_i(t)\cdot\boldsymbol{c}_k^\top(t)\boldsymbol{b}_j(t)\Big) + \eta^2\bar{\boldsymbol{b}}_i^\top(t)\bar{\boldsymbol{b}}_j(t) \tag{38}$$

$$\boldsymbol{c}_i^\top(t+1)\boldsymbol{c}_j(t+1)$$
$$= \boldsymbol{c}_i^\top(t)\boldsymbol{c}_j(t) + \eta\Big(2\big(\beta_3 - \beta_1\boldsymbol{c}_i^\top(t)\boldsymbol{b}_i(t)\big)\boldsymbol{c}_j^\top(t)\boldsymbol{b}_i(t) + 2\big(\beta_3 - \beta_1\boldsymbol{c}_j^\top(t)\boldsymbol{b}_j(t)\big)\boldsymbol{c}_i^\top(t)\boldsymbol{b}_j(t)$$
$$- \beta_3\big(\boldsymbol{c}_i^\top(t)\boldsymbol{b}_j(t) + \boldsymbol{c}_j^\top(t)\boldsymbol{b}_i(t)\big) - 2\beta_1\sum_{k\neq i,k\neq j}^{d}\boldsymbol{c}_i^\top(t)\boldsymbol{b}_k(t)\cdot\boldsymbol{c}_j^\top(t)\boldsymbol{b}_k(t) \tag{39}$$
$$- 2\beta_2\boldsymbol{c}_i^\top(t)\boldsymbol{b}(t)\cdot\boldsymbol{c}_j^\top(t)\boldsymbol{b}(t)\Big) + \eta^2\bar{\boldsymbol{c}}_i^\top(t)\bar{\boldsymbol{c}}_j(t)$$

$$\boldsymbol{b}_i^\top(t+1)\boldsymbol{b}(t+1)$$

$$= \boldsymbol{b}_i^\top(t)\boldsymbol{b}(t) + \eta\Big(\big(\beta_3 - \beta_1\boldsymbol{c}_i^\top(t)\boldsymbol{b}_i(t)\big)\boldsymbol{c}_i^\top(t)\boldsymbol{b}(t) - \beta_1\sum_{k\neq i}^{d}\boldsymbol{c}_k^\top(t)\boldsymbol{b}_i(t)\cdot\boldsymbol{c}_k^\top(t)\boldsymbol{b}(t)$$

$$- \beta_2\sum_{k=1}^{d}\boldsymbol{c}_k^\top(t)\boldsymbol{b}(t)\cdot\boldsymbol{c}_k^\top(t)\boldsymbol{b}_i(t)\Big) + \eta^2\overline{\boldsymbol{b}}_i^\top(t)\overline{\boldsymbol{b}}(t) \tag{40}$$

To give bounds for the above three terms, we will use the following bounds from property $\mathcal{B}(t)$ and lemma C.7:

$$|\beta_3 - \beta_1\boldsymbol{c}_i^\top(t)\boldsymbol{b}_i(t)| \leq \delta(t)\exp(-\eta\beta_1\gamma t)$$

$$|\boldsymbol{c}_i^\top(t)\boldsymbol{b}_j(t)| \leq 2\delta(t)\exp(-\eta\beta_1\gamma t)$$

$$|\boldsymbol{c}_i^\top(t)\boldsymbol{b}(t)| \leq 2\delta(t)\exp(-\eta\beta_2\gamma t) + \frac{\delta(t)}{\beta_2}\exp(-\eta\beta_1\gamma t)$$

$$\left|\overline{\boldsymbol{b}}_i(t)^\top\overline{\boldsymbol{b}}_j(t)\right| \leq 8d_h d^2\delta(t)^2\exp(-\eta\beta_1\gamma t),$$

$$\left|\overline{\boldsymbol{c}}_i(t)^\top\overline{\boldsymbol{c}}_j(t)\right| \leq 8d_h d^2\delta(t)^2\exp(-\eta\beta_1\gamma t) + 40\beta_2^2 d_h\delta(t)^2\exp(-\eta\beta_2\gamma t),$$

$$\left|\overline{\boldsymbol{b}}_i^\top(t)\overline{\boldsymbol{b}}(t)\right| \leq 8\beta_2 d_h d^2\delta(t)^2\exp(-\eta\beta_2\gamma t) + 4d_h d^2\delta(t)^2\exp(-\eta\beta_1\gamma t)$$

Besides, by $\left|\beta_3 - \beta_1\boldsymbol{c}_i^\top(t)\boldsymbol{b}_i(t)\right| \leq \delta(t)\exp(-\eta\beta_1\gamma t)$, we have:

$$\left|\boldsymbol{c}_i^\top(t)\boldsymbol{b}_i(t)\right| \leq \frac{\delta(t)\exp(-\eta\beta_1\gamma t) + \beta_3}{\beta_1}$$

$$\leq \frac{4\delta(t) + 2\alpha}{\alpha^2}$$

$$\leq \frac{5\delta(t)}{\alpha^2}$$

$$\leq 6\delta(t)$$

The third inequality is by $\delta(s) \geq 2\sqrt{d_h\log(4d(2d+1)/\delta)} \geq 2\alpha = 2\alpha = 2\exp((-\ln 2)/N)$. For the last inequality, as long as $N \geq \frac{2\ln 2}{\ln 6 - \ln 5}$, we have $\frac{5}{\alpha^2} \leq 6$.

We will provide the upper bounds for $\left|\boldsymbol{b}_i^\top(T+1)\boldsymbol{b}_j(T+1)\right|, \left|\boldsymbol{c}_i^\top(T+1)\boldsymbol{c}_j(T+1)\right|$ and $\left|\boldsymbol{b}_i^\top(T+1)\boldsymbol{b}(T+1)\right|$ by substituting the above bounds into (Eq. 38), (Eq. 39) and (Eq. 40) respectively.

**Bound of** $\left| \boldsymbol{b}_i^\top (T+1) \boldsymbol{b}_j(T+1) \right|$

$$
\begin{aligned}
&\left| \boldsymbol{b}_i^\top (T+1) \boldsymbol{b}_j(T+1) \right| \\
&= \Big| \boldsymbol{b}_i^\top(T)\boldsymbol{b}_j(T) + \eta\Big(2\big(\beta_3 - \beta_1 \boldsymbol{c}_i^\top(T)\boldsymbol{b}_i(T)\big)\boldsymbol{c}_i^\top(T)\boldsymbol{b}_j(T) + 2\big(\beta_3 - \beta_1 \boldsymbol{c}_j^\top(T)\boldsymbol{b}_j(T)\big)\boldsymbol{c}_j^\top(T)\boldsymbol{b}_i(T) \\
&\quad - \beta_3\big(\boldsymbol{c}_i^\top(T)\boldsymbol{b}_j(T) + \boldsymbol{c}_j^\top(T)\boldsymbol{b}_i(T)\big) - 2\beta_1 \sum_{k\neq i, k\neq j}^d \boldsymbol{c}_k^\top(T)\boldsymbol{b}_i(T)\cdot \boldsymbol{c}_k^\top(T)\boldsymbol{b}_j(T)\Big) + \eta^2 \bar{\boldsymbol{b}}_i^\top(T)\bar{\boldsymbol{b}}_j(T)\Big| \\
&\leq \left| \boldsymbol{b}_i^\top(T)\boldsymbol{b}_j(T) \right| + 2\eta\left|\big(\beta_3 - \beta_1\boldsymbol{c}_i^\top(T)\boldsymbol{b}_i(T)\big)\boldsymbol{c}_i^\top(T)\boldsymbol{b}_j(T)\right| + 2\eta\left|\big(\beta_3 - \beta_1\boldsymbol{c}_j^\top(T)\boldsymbol{b}_j(T)\big)\boldsymbol{c}_j^\top(T)\boldsymbol{b}_i(T)\right| \\
&\quad + \eta\beta_3\left|\big(\boldsymbol{c}_i^\top(T)\boldsymbol{b}_j(T) + \boldsymbol{c}_j^\top(T)\boldsymbol{b}_i(T)\big)\right| + 2\eta\beta_1 \sum_{k\neq i, k\neq j}^d \left|\boldsymbol{c}_k^\top(T)\boldsymbol{b}_i(T)\cdot\boldsymbol{c}_k^\top(T)\boldsymbol{b}_j(T)\right| + \eta^2\left|\bar{\boldsymbol{b}}_i^\top(T)\bar{\boldsymbol{b}}_j(T)\right| \\
&\leq \delta(T) + 4\eta\cdot\delta(T)\exp(-\eta\beta_1\gamma T)\cdot 2\delta(T)\exp(-\eta\beta_1\gamma T) + 2\eta\beta_3\cdot 2\delta(T)\exp(-\eta\beta_1\gamma T) \\
&\quad + 2\eta\beta_1(d-2)\cdot\Big(2\delta(T)\exp(-\eta\beta_1\gamma T)\Big)^2 + \eta^2\cdot 8d_h d^2\delta(T)^2\exp(-\eta\beta_1\gamma T) \\
&\leq \delta(T) + \delta(T)\Big(8\eta\delta(T)\exp(-\eta\beta_1\gamma T) + 4\eta\beta_3 + 8\eta\beta_1(d-2)\delta(T)\exp(-\eta\beta_1\gamma T) \\
&\quad + 8\eta^2 d_h d^2\delta(T)\Big)\cdot\exp(-\eta\beta_1\gamma T) \\
&\leq \delta(T) + \delta(T)\Big(8\eta\delta_{\max} + 4\eta\beta_3 + 8\eta\beta_1(d-2)\delta_{\max} + 8\eta^2 d_h d^2\delta_{\max}\Big)\exp(-\eta\beta_1\gamma T)
\end{aligned}
$$

$$(41)$$

The first inequality is derived by triangle inequality. The second inequality is derived by $\left|\boldsymbol{b}_i^\top(T)\boldsymbol{b}_j(T)\right| \leq \delta(T)$ and substituting the bounds of $|\beta_3 - \beta_1\boldsymbol{c}_i^\top(t)\boldsymbol{b}_i(t)|$, $|\boldsymbol{c}_i^\top(t)\boldsymbol{b}_j(t)|$ and $\left|\bar{\boldsymbol{b}}_i(t)^\top\bar{\boldsymbol{b}}_j(t)\right|$. The third inequality is derived by factoring out the common factor $\delta(T)$. The last inequality is derived by $\delta(T) \leq \delta_{\max}$ and $\exp(-\eta\beta_1\gamma T) \leq 1$.

**Bound of** $\left|\boldsymbol{c}_i^\top(T+1)\boldsymbol{c}_j(T+1)\right|$

$$
\left|\boldsymbol{c}_i^\top(T+1)\boldsymbol{c}_j(T+1)\right|
$$

$$
= \left|\boldsymbol{c}_i^\top(T)\boldsymbol{c}_j(T) + \eta\Big(2\big(\beta_3 - \beta_1\boldsymbol{c}_i^\top(T)\boldsymbol{b}_i(T)\big)\boldsymbol{c}_j^\top(T)\boldsymbol{b}_i(T) + 2\big(\beta_3 - \beta_1\boldsymbol{c}_j^\top(T)\boldsymbol{b}_j(T)\big)\boldsymbol{c}_i^\top(T)\boldsymbol{b}_j(T)\right.
$$

$$
- \beta_3\big(\boldsymbol{c}_i^\top(T)\boldsymbol{b}_j(T) + \boldsymbol{c}_j^\top(T)\boldsymbol{b}_i(T)\big) - 2\beta_1 \sum_{k\neq i, k\neq j}^{d} \boldsymbol{c}_i^\top(T)\boldsymbol{b}_k(T) \cdot \boldsymbol{c}_j^\top(T)\boldsymbol{b}_k(T)
$$

$$
\left.- 2\beta_2\boldsymbol{c}_i^\top(T)\boldsymbol{b}(T) \cdot \boldsymbol{c}_j^\top(T)\boldsymbol{b}(T)\Big) + \eta^2\bar{\boldsymbol{c}}_i^\top(T)\bar{\boldsymbol{c}}_j(T)\right|
$$

$$
\leq \left|\boldsymbol{c}_i^\top(T)\boldsymbol{c}_j(T)\right| + 2\eta\left|\big(\beta_3 - \beta_1\boldsymbol{c}_i^\top(T)\boldsymbol{b}_i(T)\big)\boldsymbol{c}_j^\top(T)\boldsymbol{b}_i(T)\right| + 2\eta\left|\big(\beta_3 - \beta_1\boldsymbol{c}_j^\top(T)\boldsymbol{b}_j(T)\big)\boldsymbol{c}_i^\top(T)\boldsymbol{b}_j(T)\right|
$$

$$
+ \eta\beta_3\left|\big(\boldsymbol{c}_i^\top(T)\boldsymbol{b}_j(T) + \boldsymbol{c}_j^\top(T)\boldsymbol{b}_i(T)\big)\right| + 2\eta\beta_1 \sum_{k\neq i, k\neq j}^{d} \left|\boldsymbol{c}_i^\top(T)\boldsymbol{b}_k(T) \cdot \boldsymbol{c}_j^\top(T)\boldsymbol{b}_k(T)\right|
$$

$$
+ 2\eta\beta_2\left|\boldsymbol{c}_i^\top(T)\boldsymbol{b}(T) \cdot \boldsymbol{c}_j^\top(T)\boldsymbol{b}(T)\right| + \eta^2\left|\bar{\boldsymbol{c}}_i^\top(T)\bar{\boldsymbol{c}}_j(T)\right|
$$

$$
\leq \delta(T) + 4\eta \cdot \delta(T)\exp(-\eta\beta_1\gamma T) \cdot 2\delta(T)\exp(-\eta\beta_1\gamma T) + 2\eta\beta_3 \cdot 2\delta(T)\exp(-\eta\beta_1\gamma T)
$$

$$
+ 2\eta\beta_1(d-2) \cdot \Big(2\delta(T)\exp(-\eta\beta_1\gamma T)\Big)^2
$$

$$
+ 2\eta\beta_2\Big(2\delta(T)\exp(-\eta\beta_2\gamma T) + \frac{\delta(T)}{\beta_2}\exp(-\eta\beta_1\gamma T)\Big)^2
$$

$$
+ \eta^2 \cdot \Big(8d_h d^2\delta(T)^2\exp(-\eta\beta_1\gamma T) + 40\beta_2^2 d_h\delta(T)^2\exp(-\eta\beta_2\gamma T)\Big)
$$

$$
\leq \delta(T) + \delta(T)\Big(8\eta\delta(T)\exp(-\eta\beta_1\gamma T) + 4\eta\beta_3 + 8\eta\beta_1(d-2)\delta(T)\exp(-\eta\beta_1\gamma T) + 8\eta^2 d_h d^2\delta(T)
$$

$$
+ \frac{2\eta\delta(T)}{\beta_2}\exp(-\eta\beta_1\gamma T) + 8\eta\delta(T)\exp(-\eta\beta_2\gamma T)\Big) \cdot \exp(-\eta\beta_1\gamma T)
$$

$$
+ \delta(T)\Big(8\eta\beta_2\delta(T)\exp(-\eta\beta_2\gamma T) + 40\eta^2\beta_2^2 d_h\delta(T)\Big) \cdot \exp(-\eta\beta_2\gamma T)
$$

$$
\leq \delta(T) + \delta(T)\Big(16\eta\delta_{\max} + 4\eta\beta_3 + 8\eta\beta_1(d-2)\delta_{\max} + 8\eta^2 d_h d^2\delta_{\max} + \frac{2\eta\delta_{\max}}{\beta_2}\Big) \cdot \exp(-\eta\beta_1\gamma T)
$$

$$
+ \delta(T)\Big(8\eta\beta_2\delta_{\max} + 40\eta^2\beta_2^2 d_h\delta_{\max}\Big) \cdot \exp(-\eta\beta_2\gamma T)
$$

$$
\tag{42}
$$

The first inequality is derived by triangle inequality. The second inequality is derived by $\left|\boldsymbol{b}_i^\top(T)\boldsymbol{b}_j(T)\right| \leq \delta(T)$ and substituting the bounds of $|\beta_3 - \beta_1\boldsymbol{c}_i^\top(t)\boldsymbol{b}_i(t)|$, $|\boldsymbol{c}_i^\top(t)\boldsymbol{b}_j(t)|$ and $\left|\bar{\boldsymbol{c}}_i(t)^\top\bar{\boldsymbol{c}}_j(t)\right|$. The third inequality is derived by factoring out the common factor $\delta(T)$. The last inequality is derived by $\delta(T) \leq \delta_{\max}$, $\exp(-\eta\beta_1\gamma T) \leq 1$ and $\exp(-\eta\beta_2\gamma T) \leq 1$.

**Bound of** $\left|\boldsymbol{b}_i^\top(T+1)\boldsymbol{b}(T+1)\right|$

$$\left|\boldsymbol{b}_i^\top(T+1)\boldsymbol{b}(T+1)\right|$$

$$= \left|\boldsymbol{b}_i^\top(T)\boldsymbol{b}(T) + \eta\Big((\beta_3 - \beta_1\boldsymbol{c}_i^\top(T)\boldsymbol{b}_i(T))\boldsymbol{c}_i^\top(T)\boldsymbol{b}(T) - \beta_1\sum_{k\neq i}^d \boldsymbol{c}_k^\top(T)\boldsymbol{b}_i(T)\cdot\boldsymbol{c}_k^\top(T)\boldsymbol{b}(T)\right.$$

$$\left. - \beta_2\sum_{k=1}^d \boldsymbol{c}_k^\top(T)\boldsymbol{b}(T)\cdot\boldsymbol{c}_k^\top(T)\boldsymbol{b}_i(T)\Big) + \eta^2\bar{\boldsymbol{b}}_i^\top(T)\bar{\boldsymbol{b}}(T)\right|$$

$$= \left|\boldsymbol{b}_i^\top(T)\boldsymbol{b}(T) + \eta\Big((\beta_3 - \beta_1\boldsymbol{c}_i^\top(T)\boldsymbol{b}_i(T))\boldsymbol{c}_i^\top(T)\boldsymbol{b}(T) - \beta_1\sum_{k\neq i}^d \boldsymbol{c}_k^\top(T)\boldsymbol{b}_i(T)\cdot\boldsymbol{c}_k^\top(T)\boldsymbol{b}(T)\right.$$

$$\left. - \beta_2\sum_{k\neq i}^d \boldsymbol{c}_k^\top(T)\boldsymbol{b}(T)\cdot\boldsymbol{c}_k^\top(T)\boldsymbol{b}_i(T) - \beta_2\boldsymbol{c}_i^\top(T)\boldsymbol{b}(T)\cdot\boldsymbol{c}_i^\top(T)\boldsymbol{b}_i(T)\Big) + \eta^2\bar{\boldsymbol{b}}_i^\top(T)\bar{\boldsymbol{b}}(T)\right|$$

$$\leq \left|\boldsymbol{b}_i^\top(T)\boldsymbol{b}(T)\right| + \eta\left|(\beta_3 - \beta_1\boldsymbol{c}_i^\top(T)\boldsymbol{b}_i(T))\boldsymbol{c}_i^\top(T)\boldsymbol{b}(T)\right| + \eta\beta_1\sum_{k\neq i}^d\left|\boldsymbol{c}_k^\top(T)\boldsymbol{b}_i(T)\cdot\boldsymbol{c}_k^\top(T)\boldsymbol{b}(T)\right|$$

$$+ \eta\beta_2\sum_{k\neq i}^d\left|\boldsymbol{c}_k^\top(T)\boldsymbol{b}(T)\cdot\boldsymbol{c}_k^\top(T)\boldsymbol{b}_i(T)\right| + \eta\beta_2\left|\boldsymbol{c}_i^\top(T)\boldsymbol{b}(T)\cdot\boldsymbol{c}_i^\top(T)\boldsymbol{b}_i(T)\right| + \eta^2\left|\bar{\boldsymbol{b}}_i^\top(T)\bar{\boldsymbol{b}}(T)\right|$$

$$\leq \delta(T) + \eta\cdot\delta(T)\exp(-\eta\beta_1\gamma T)\cdot\Big(2\delta(T)\exp(-\eta\beta_2\gamma T) + \frac{\delta(T)}{\beta_2}\exp(-\eta\beta_1\gamma T)\Big)$$

$$+ \eta(\beta_1 + \beta_2)(d-1)\cdot 2\delta(T)\exp(-\eta\beta_1\gamma T)\cdot\Big(2\delta(T)\exp(-\eta\beta_2\gamma T) + \frac{\delta(T)}{\beta_2}\exp(-\eta\beta_1\gamma T)\Big)$$

$$+ \eta\beta_2\cdot 6\delta(T)\cdot\Big(2\delta(T)\exp(-\eta\beta_2\gamma T) + \frac{\delta(T)}{\beta_2}\exp(-\eta\beta_1\gamma T)\Big)$$

$$+ \eta^2\cdot\Big(8\beta_2 d_h d^2\delta(T)^2\exp(-\eta\beta_2\gamma T) + 4d_h d^2\delta(T)^2\exp(-\eta\beta_1\gamma T)\Big)$$

$$\leq \delta(T) + \delta(T)\Big(\eta\big(2(\beta_1 + \beta_2)(d-1) + 1\big)\cdot\frac{\delta(T)}{\beta_2}\exp(-\eta\beta_1\gamma T)$$

$$+ 6\eta\beta_2\cdot\frac{\delta(T)}{\beta_2} + 4\eta^2 d_h d^2\delta(T)\Big)\cdot\exp(-\eta\beta_1\gamma T)$$

$$+ \delta(T)\Big(2\eta\big(2(\beta_1 + \beta_2)(d-1) + 1\big)\delta(T)\exp(-\eta\beta_1\gamma T)$$

$$+ 12\eta\beta_2\delta(T) + 8\eta^2\beta_2 d_h d^2\delta(T)\Big)\cdot\exp(-\eta\beta_2\gamma T)$$

$$\leq \delta(T) + \delta(T)\Big(4\eta d\delta_{\max} + 4\eta^2 d_h d^2\delta_{\max}\Big)\cdot\exp(-\eta\beta_1\gamma T)$$

$$+ \delta(T)\Big(8\eta\beta_2 d\delta_{\max} + 12\eta\beta_2\delta_{\max} + 8\eta^2\beta_2 d_h d^2\delta_{\max}\Big)\cdot\exp(-\eta\beta_2\gamma T)$$

$$\tag{43}$$

The first inequality is derived by triangle inequality. The second inequality is derived by $\left|\boldsymbol{b}_i^\top(T)\boldsymbol{b}_j(T)\right| \leq \delta(T)$ and substituting the bounds of $|\beta_3 - \beta_1\boldsymbol{c}_i^\top(t)\boldsymbol{b}_i(t)|$, $|\boldsymbol{c}_i^\top(t)\boldsymbol{b}_j(t)|$, $|\boldsymbol{c}_i^\top(t)\boldsymbol{b}_i(t)|$ and $\left|\bar{\boldsymbol{b}}_i(t)^\top\bar{\boldsymbol{b}}(t)\right|$. The third inequality is derived by factoring out the common factor $\delta(T)$. The last inequality is derived by $\delta(T) \leq \delta_{\max}$, $\exp(-\eta\beta_1\gamma T) \leq 1$ and $2(\beta_1 + \beta_2)(d-1) + 1 \leq 4\beta_2 d$ since $\beta_2 \geq \beta_1$ and $\beta_2 \geq 1$.

We next provide the upper bound for $\delta(T+1)$.

$$\delta(T+1) = \max\{|\boldsymbol{b}_i^\top(T+1)\boldsymbol{b}_j(T+1)|, |\boldsymbol{c}_i^\top(T+1)\boldsymbol{c}_j(T+1)|, |\boldsymbol{b}_i^\top(T+1)\boldsymbol{b}(T+1)|\}$$

$$\leq \delta(T) + \delta(T)\left(16\eta d\delta_{\max} + 4\eta\beta_3 + 8\eta\beta_1(d-2)\delta_{\max} + 8\eta^2 d_h d^2 \delta_{\max} + \frac{2\eta\delta_{\max}}{\beta_2}\right) \cdot \exp(-\eta\beta_1\gamma T)$$

$$+ \delta(T)\left(8\eta\beta_2 d\delta_{\max} + 40\eta^2\beta_2^2 d_h \delta_{\max} + 12\eta\beta_2\delta_{\max} + 8\eta^2\beta_2 d_h d^2 \delta_{\max}\right) \cdot \exp(-\eta\beta_2\gamma T)$$

$$\tag{44}$$

This inequality can be verified by comparing with (Eq. (41)), (Eq. (42)), (Eq. (43)). To give more precise bound, we introduce the following lemma:

**Lemma C.6** *If $y(t+1) \leq y(t) + cy(t)\exp(-at) + dy(t)\exp(-bt)$, with $a, b, c, d > 0, t \geq 0$ and $a, b \leq \ln 2$, then $y(t)$ satisfies:*

$$y(t) \leq y(0)\exp(\frac{2c}{a} + \frac{2d}{b})$$

Proof of lemma C.6.

$$y(t+1) \leq y(t) + cy(t)\exp(-at) + dy(t)\exp(-bt)$$

$$\Longrightarrow y(t+1) \leq y(t)(1 + c\exp(-at) + d\exp(-bt))$$

$$\Longrightarrow y(t+1) \leq y(0)\prod_{s=0}^{t}(1 + c\exp(-as) + d\exp(-bs))$$

$$\Longrightarrow \ln y(t+1) \leq \ln y(0) + \sum_{s=0}^{t}\ln(1 + c\exp(-as) + d\exp(-bs))$$

$$\Longrightarrow \ln y(t+1) \leq \ln y(0) + \sum_{s=0}^{t}(c\exp(-as) + d\exp(-bs))$$

$$\Longrightarrow \ln y(t+1) \leq \ln y(0) + \int_{-1}^{t}(c\exp(-as) + d\exp(-bs))ds$$

$$\Longrightarrow \ln y(t+1) \leq \ln y(0) + (\frac{c}{a}(\exp(a) - \exp(-at)) + \frac{d}{b}(\exp(b) - \exp(-bt)))$$

$$\Longrightarrow \ln y(t+1) \leq \ln y(0) + (\frac{2c}{a} + \frac{2d}{b})$$

$$\Longrightarrow y(t+1) \leq y(0)\exp(\frac{2c}{a} + \frac{2d}{b})$$

The fourth arrow is due to $\ln(1+x) \leq x$ for $x \geq 0$. The fifth arrow is due to $\exp(-as), \exp(-bs)$ are monotone decreasing. the 7-th arrow is due to $a, b \leq \ln 2$ and $-\exp(-at) \leq 0, -\exp(-bt) \leq 0$.

Lemma C.6 presents the core idea of establishing property $\mathcal{C}(T+1)$. If $a \gg c$ and $b \gg d$ in the above lemma, we will have $y(t+1) \leq y(0) \cdot O(1)$. Similarly, as Mamba converges quickly ($\boldsymbol{C}^\top\boldsymbol{B} \to \frac{\beta_3}{\beta_1}\boldsymbol{I}$, $\boldsymbol{C}^\top\boldsymbol{b} \to \boldsymbol{0}$), we can prove that $|\boldsymbol{b}_i^\top(t)\boldsymbol{b}_j(t)|, |\boldsymbol{c}_i^\top(t)\boldsymbol{c}_j(t)|, |\boldsymbol{b}_i^\top(t)\boldsymbol{b}(t)|$ hold their magnitudes around their initial states.

We next combine (Eq. (44)) and lemma C.6 to give bound for $\delta(T+1)$.

$$\delta(T+1)$$

$$\leq \delta(T) + \delta(T)\left(16\eta d\delta_{\max} + 4\eta\beta_3 + 8\eta\beta_1(d-2)\delta_{\max} + 8\eta^2 d_h d^2 \delta_{\max} + \frac{2\eta\delta_{\max}}{\beta_2}\right) \cdot \exp(-\eta\beta_1\gamma T)$$

$$+ \delta(T)\left(8\eta\beta_2 d\delta_{\max} + 40\eta^2\beta_2^2 d_h \delta_{\max} + 12\eta\beta_2\delta_{\max} + 8\eta^2\beta_2 d_h d^2 \delta_{\max}\right) \cdot \exp(-\eta\beta_2\gamma T)$$

$$\leq \delta(0) \cdot \exp\left(\frac{2\left(16\eta d\delta_{\max} + 4\eta\beta_3 + 8\eta\beta_1(d-2)\delta_{\max} + 8\eta^2 d_h d^2 \delta_{\max} + \frac{2\eta\delta_{\max}}{\beta_2}\right)}{\eta\beta_1\gamma}\right.$$

$$+ \frac{2\big(8\eta\beta_2 d\delta_{\max} + 40\eta^2\beta_2^2 d_h\delta_{\max} + 12\eta\beta_2\delta_{\max} + 8\eta^2\beta_2 d_h d^2\delta_{\max}\big)}{\eta\beta_2\gamma}\bigg)$$

$$\leq \delta(0) \cdot \exp\bigg(\frac{3\sqrt{d_h \log(4d(2d+1)/\delta)}}{\frac{1}{2}d_h} \cdot \bigg(\frac{32d}{\beta_1} + \frac{8\beta_3}{\beta_1} + 16(d-2) + \frac{16\eta d_h d^2}{\beta_1} + \frac{4}{\beta_1\beta_2}$$

$$+ 16d + 80\eta\beta_2 d_h + 24 + 16\eta d_h d^2\bigg)\bigg)$$

$$\leq \frac{3}{2} \cdot \delta(0)$$

$$\leq 3\sqrt{d_h \log(4d(2d+1)/\delta)}$$

The first inequality is derived by (Eq. (44)). The second inequality is derived by lemma C.6. The third inequality is derived by $\gamma = \frac{1}{2}d_h$. The last inequality is derived by

$$d_h = \widetilde{\Omega}(d^2)$$

$$\geq \frac{36}{(\ln(3/2))^2}\log(4d(2d+1)/\delta)\bigg(\frac{32d}{\beta_1} + \frac{8\beta_3}{\beta_1}$$

$$+ 16(d-2) + \frac{8}{\beta_1} + \frac{4}{\beta_1\beta_2} + 16d + 80\ln 2 + 24 + 8\bigg)^2$$

$$\geq \frac{36}{(\ln(3/2))^2}\log(4d(2d+1)/\delta)\bigg(\frac{32d}{\beta_1} + \frac{8\beta_3}{\beta_1}$$

$$+ 16(d-2) + \frac{16\eta d_h d^2}{\beta_1} + \frac{4}{\beta_1\beta_2} + 16d + 80\eta\beta_2 d_h + 24 + 16\eta d_h d^2\bigg)^2$$

### C.7  Bounds of $\eta^2$ terms

This Section presents the bounds for $\bar{\boldsymbol{b}}_i(t)^\top\bar{\boldsymbol{b}}_j(t)$, $\bar{\boldsymbol{c}}_i(t)^\top\bar{\boldsymbol{c}}_j(t)$, $\left\|\bar{\boldsymbol{b}}(t)\right\|_2^2$, $\bar{\boldsymbol{c}}_i^\top(t)\bar{\boldsymbol{b}}_j(t)$, $\bar{\boldsymbol{c}}_i^\top(t)\bar{\boldsymbol{b}}(t)$, $\bar{\boldsymbol{b}}_i^\top(t)\bar{\boldsymbol{b}}(t)$ (these terms usually appear in the *Vector-coupled Dynamics* equations with a $\eta^2$ factor) with $i, j \in [1, d]$ under the assumption that $\mathcal{A}(t)$, $\mathcal{B}(t)$, and $\mathcal{C}(t)$ hold.

**Lemma C.7** *Under the assumption that $\mathcal{A}(t)$, $\mathcal{B}(t)$, and $\mathcal{C}(t)$ hold, we have the following bounds:*

$$\left|\bar{\boldsymbol{b}}_i(t)^\top\bar{\boldsymbol{b}}_j(t)\right| \leq 8d_h d^2\delta(t)^2\exp(-\eta\beta_1\gamma t),$$

$$\left|\bar{\boldsymbol{c}}_i(t)^\top\bar{\boldsymbol{c}}_j(t)\right| \leq 8d_h d^2\delta(t)^2\exp(-\eta\beta_1\gamma t) + 24\beta_2^2 d_h\delta(t)^2\exp(-\eta\beta_2\gamma t),$$

$$\left\|\bar{\boldsymbol{b}}(t)\right\|_2^2 \leq 16d_h\beta_2^2 d^2\delta(t)^2\exp(-\eta\beta_2\gamma t) + 2d_h d^2\delta(t)^2\exp(-\eta\beta_1\gamma t),$$

$$\left|\bar{\boldsymbol{c}}_i^\top(t)\bar{\boldsymbol{b}}_j(t)\right| \leq 8d_h d^2\delta(t)^2\exp(-\eta\beta_1\gamma t) + 8\beta_2 d_h d\delta(t)^2\exp(-\eta\beta_2\gamma t),$$

$$\left|\bar{\boldsymbol{c}}_i^\top(t)\bar{\boldsymbol{b}}(t)\right| \leq 4d_h d^2\delta(t)^2\exp(-\eta\beta_1\gamma t) + 28\beta_2^2 d_h d\delta(t)^2\exp(-2\eta\beta_2\gamma t),$$

$$\left|\bar{\boldsymbol{b}}_i^\top(t)\bar{\boldsymbol{b}}(t)\right| \leq 8\beta_2 d_h d^2\delta(t)^2\exp(-\eta\beta_2\gamma t) + 4d_h d^2\delta(t)^2\exp(-\eta\beta_1\gamma t)$$

*where $i, j \in [1, d]$. Note that this lemma does not require $i \neq j$.*

Firstly, recall the following dynamics equation in lemma A.6:

$$\boldsymbol{b}_i(t+1) = \boldsymbol{b}_i(t) + \eta\bigg(\big(\beta_3 - \beta_1\boldsymbol{c}_i^\top(t)\boldsymbol{b}_i(t)\big)\boldsymbol{c}_i(t) - \beta_1\sum_{k\neq i}^d \boldsymbol{c}_k^\top(t)\boldsymbol{b}_i(t) \cdot \boldsymbol{c}_k(t)\bigg)$$

$$=: \boldsymbol{b}_i(t) + \eta\bar{\boldsymbol{b}}_i(t)$$

$$\boldsymbol{c}_i(t+1) = \boldsymbol{c}_i(t) + \eta\bigg(\big(\beta_3 - \beta_1\boldsymbol{c}_i^\top(t)\boldsymbol{b}_i(t)\big)\boldsymbol{b}_i(t) - \beta_1\sum_{k\neq i}^d \boldsymbol{c}_i^\top(t)\boldsymbol{b}_k(t) \cdot \boldsymbol{b}_k(t)$$

$$- \beta_2 \boldsymbol{c}_i^\top(t)\boldsymbol{b}(t) \cdot \boldsymbol{b}(t)\Big)$$
$$=: \boldsymbol{c}_i(t) + \eta \bar{\boldsymbol{c}}_i(t)$$

$$\boldsymbol{b}(t+1) = \boldsymbol{b}(t) - \eta\Big(\beta_2 \sum_{k=1}^d \boldsymbol{c}_k^\top(t)\boldsymbol{b}(t) \cdot \boldsymbol{c}_k(t)\Big) =: \boldsymbol{b}(t) + \eta\bar{\boldsymbol{b}}(t)$$

Thus we have

$$\bar{\boldsymbol{b}}_i(t) = \big(\beta_3 - \beta_1 \boldsymbol{c}_i^\top(t)\boldsymbol{b}_i(t)\big)\boldsymbol{c}_i(t) - \beta_1 \sum_{k\neq i}^d \boldsymbol{c}_k^\top(t)\boldsymbol{b}_i(t) \cdot \boldsymbol{c}_k(t)$$

$$\bar{\boldsymbol{c}}_i(t) = \big(\beta_3 - \beta_1 \boldsymbol{c}_i^\top(t)\boldsymbol{b}_i(t)\big)\boldsymbol{b}_i(t) - \beta_1 \sum_{k\neq i}^d \boldsymbol{c}_i^\top(t)\boldsymbol{b}_k(t) \cdot \boldsymbol{b}_k(t) - \beta_2 \boldsymbol{c}_i^\top(t)\boldsymbol{b}(t) \cdot \boldsymbol{b}(t)$$

$$\bar{\boldsymbol{b}}(t) = \beta_2 \sum_{k=1}^d \boldsymbol{c}_k^\top(t)\boldsymbol{b}(t) \cdot \boldsymbol{c}_k(t)$$

Recalling the properties $\mathcal{A}(t)$ and $\mathcal{B}(t)$:

$\mathcal{A}(t)$ :
$$d_h/2 \leq \boldsymbol{b}_i^\top(t)\boldsymbol{b}_i(t), \boldsymbol{c}_i^\top(t)\boldsymbol{c}_i(t), \boldsymbol{b}^\top(t)\boldsymbol{b}(t) \leq 2d_h$$

$\mathcal{B}(t)$ :
$$|\beta_3/\beta_1 - \boldsymbol{c}_i^\top(t)\boldsymbol{b}_i(t)| \leq \delta(t)\exp(-\eta\beta_1\gamma t)$$
$$|\boldsymbol{c}_i^\top(t)\boldsymbol{b}_j(t)| \leq 2\delta(t)\exp(-\eta\beta_1\gamma t)$$
$$|\boldsymbol{c}_i^\top(t)\boldsymbol{b}(t)| \leq 2\delta(t)\exp(-\eta\beta_2\gamma t) + \frac{\delta(t)}{\beta_2}\exp(-\eta\beta_1\gamma t)$$

We can derive the follow bounds for the norm of $\bar{\boldsymbol{b}}_i(t)$, $\bar{\boldsymbol{c}}_i(t)$ and $\bar{\boldsymbol{b}}(t)$:

$$\left\|\bar{\boldsymbol{b}}_i(t)\right\| = \left\|\big(\beta_3 - \beta_1 \boldsymbol{c}_i^\top(t)\boldsymbol{b}_i(t)\big)\boldsymbol{c}_i(t) - \beta_1 \sum_{k\neq i}^d \boldsymbol{c}_k^\top(t)\boldsymbol{b}_i(t) \cdot \boldsymbol{c}_k(t)\right\|$$

$$\leq \left\|\boldsymbol{c}_i(t)\right\| \cdot \left|\big(\beta_3 - \beta_1 \boldsymbol{c}_i^\top(t)\boldsymbol{b}_i(t)\big)\right| + \beta_1 \sum_{k\neq i}^d \left\|\boldsymbol{c}_k(t)\right\| \cdot \left|\boldsymbol{c}_k^\top(t)\boldsymbol{b}_i(t)\right|$$

$$\leq \sqrt{2d_h}\delta(t)\exp(-\eta\beta_1\gamma t) + \sqrt{2d_h}\beta_1(d-1) \cdot 2\delta(t)\exp(-\eta\beta_1\gamma t)$$
$$\leq 2\sqrt{2d_h}d\delta(t)\exp(-\eta\beta_1\gamma t)$$

The last inequality is by $\beta_1 \leq 1$.

$$\left\|\bar{\boldsymbol{c}}_i(t)\right\| = \left\|\big(\beta_3 - \beta_1 \boldsymbol{c}_i^\top(t)\boldsymbol{b}_i(t)\big)\boldsymbol{b}_i(t) - \beta_1 \sum_{k\neq i}^d \boldsymbol{c}_i^\top(t)\boldsymbol{b}_k(t) \cdot \boldsymbol{b}_k(t) - \beta_2 \boldsymbol{c}_i^\top(t)\boldsymbol{b}(t) \cdot \boldsymbol{b}(t)\right\|$$

$$\leq \left\|\boldsymbol{b}_i(t)\right\| \cdot \left|\big(\beta_3 - \beta_1 \boldsymbol{c}_i^\top(t)\boldsymbol{b}_i(t)\big)\right| + \beta_1 \sum_{k\neq i}^d \left\|\boldsymbol{b}_k(t)\right\| \cdot \left|\boldsymbol{c}_i^\top(t)\boldsymbol{b}_k(t)\right| + \beta_2\left\|\boldsymbol{b}(t)\right\| \cdot \left|\boldsymbol{c}_i^\top(t)\boldsymbol{b}(t)\right|$$

$$\leq \sqrt{2d_h}\delta(t)\exp(-\eta\beta_1\gamma t) + \sqrt{2d_h}\beta_1(d-1) \cdot 2\delta(t)\exp(-\eta\beta_1\gamma t)$$
$$+ \sqrt{2d_h}\beta_2 \cdot \Big(2\delta(t)\exp(-\eta\beta_2\gamma t) + \frac{\delta(t)}{\beta_2}\exp(-\eta\beta_1\gamma t)\Big)$$
$$\leq 2\sqrt{2d_h}d\delta(t)\exp(-\eta\beta_1\gamma t) + 2\sqrt{2d_h}\beta_2\delta(t)\exp(-\eta\beta_2\gamma t)$$

The last inequality is by $\beta_1 \leq 1$.

$$\left\|\bar{\boldsymbol{b}}(t)\right\| = \left\|\beta_2 \sum_{k=1}^d \boldsymbol{c}_k^\top(t)\boldsymbol{b}(t) \cdot \boldsymbol{c}_k(t)\right\|$$

$$\leq \beta_2 \sum_{k=1}^{d} \left\| \boldsymbol{c}_k(t) \right\| \cdot \left| \boldsymbol{c}_k^\top(t) \boldsymbol{b}(t) \right|$$

$$\leq \sqrt{2d_h} \beta_2 d \cdot \left( 2\delta(t) \exp(-\eta\beta_2\gamma t) + \frac{\delta(t)}{\beta_2} \exp(-\eta\beta_1\gamma t) \right)$$

$$\leq 2\sqrt{2d_h} \beta_2 d\delta(t) \exp(-\eta\beta_2\gamma t) + \sqrt{2d_h} d\delta(t) \exp(-\eta\beta_1\gamma t)$$

By multiplying them pairwise, we obtain:

$$\left| \bar{\boldsymbol{b}}_i(t)^\top \bar{\boldsymbol{b}}_j(t) \right| \leq \left( 2\sqrt{2d_h} d\delta(t) \exp(-\eta\beta_1\gamma t) \right)^2 \leq 8 d_h d^2 \delta(t)^2 \exp(-\eta\beta_1\gamma t)$$

The last inequality is by $\exp(-2\eta\beta_1\gamma t) \leq \exp(-\eta\beta_1\gamma t)$.

$$\left| \bar{\boldsymbol{c}}_i(t)^\top \bar{\boldsymbol{c}}_j(t) \right| \leq \left( 2\sqrt{2d_h} d\delta(t) \exp(-\eta\beta_1\gamma t) + 2\sqrt{2d_h} \beta_2 \delta(t) \exp(-\eta\beta_2\gamma t) \right)^2$$

$$= 8 d_h d^2 \delta(t)^2 \exp(-2\eta\beta_1\gamma t) + 8\beta_2^2 d_h \delta(t)^2 \exp(-2\eta\beta_2\gamma t)$$

$$+ 16 d_h \beta_2 d\delta(t)^2 \exp(-\eta(\beta_1 + \beta_2)\gamma t)$$

$$\leq 8 d_h d^2 \delta(t)^2 \exp(-\eta\beta_1\gamma t) + 40\beta_2^2 d_h \delta(t)^2 \exp(-\eta\beta_2\gamma t)$$

The last inequality is by $\beta_2 d \leq 2\beta_2^2$ because $\beta_2 = \Omega(d^2) \geq d$, and $\exp(-2\eta\beta_1\gamma t) \leq \exp(-\eta\beta_1\gamma t)$, $\exp(-2\eta\beta_2\gamma t) \leq \exp(-\eta\beta_2\gamma t)$, $\exp(-\eta(\beta_1 + \beta_2)\gamma t) \leq \exp(-\eta\beta_2\gamma t)$.

$$\left\| \bar{\boldsymbol{b}}(t) \right\|_2^2 \leq \left( 2\sqrt{2d_h} \beta_2 d\delta(t) \exp(-\eta\beta_2\gamma t) + \sqrt{2d_h} d\delta(t) \exp(-\eta\beta_1\gamma t) \right)^2$$

$$= 8 d_h \beta_2^2 d^2 \delta(t)^2 \exp(-2\eta\beta_2\gamma t) + 2 d_h d^2 \delta(t)^2 \exp(-2\eta\beta_1\gamma t)$$

$$+ 8 d_h \beta_2 d^2 \delta(t)^2 \exp(-\eta(\beta_1 + \beta_2)\gamma t)$$

$$\leq 16 d_h \beta_2^2 d^2 \delta(t)^2 \exp(-\eta\beta_2\gamma t) + 2 d_h d^2 \delta(t)^2 \exp(-\eta\beta_1\gamma t)$$

The last inequality is by $\beta_2 \leq \beta_2^2$ because $\beta_2 \geq 1$, and $\exp(-2\eta\beta_1\gamma t) \leq \exp(-\eta\beta_1\gamma t)$, $\exp(-2\eta\beta_2\gamma t) \leq \exp(-\eta\beta_2\gamma t)$, $\exp(-\eta(\beta_1 + \beta_2)\gamma t) \leq \exp(-\eta\beta_2\gamma t)$.

$$\left| \bar{\boldsymbol{c}}_i^\top(t) \bar{\boldsymbol{b}}_j(t) \right| \leq \left\| \bar{\boldsymbol{c}}_i(t) \right\| \cdot \left\| \bar{\boldsymbol{b}}_j(t) \right\|$$

$$\leq \left( 2\sqrt{2d_h} d\delta(t) \exp(-\eta\beta_1\gamma t) + 2\sqrt{2d_h} \beta_2 \delta(t) \exp(-\eta\beta_2\gamma t) \right) \cdot 2\sqrt{2d_h} d\delta(t) \exp(-\eta\beta_1\gamma t)$$

$$\leq 8 d_h d^2 \delta(t)^2 \exp(-\eta\beta_1\gamma t) + 8\beta_2 d_h d\delta(t)^2 \exp(-\eta\beta_2\gamma t)$$

The last inequality is by $\exp(-2\eta\beta_1\gamma t) \leq \exp(-\eta\beta_1\gamma t)$, $\exp(-\eta(\beta_1 + \beta_2)\gamma t) \leq \exp(-\eta\beta_2\gamma t)$.

$$\left| \bar{\boldsymbol{c}}_i^\top(t) \bar{\boldsymbol{b}}(t) \right| \leq \left\| \bar{\boldsymbol{c}}_i(t) \right\| \cdot \left\| \bar{\boldsymbol{b}}(t) \right\|$$

$$\leq \left( 2\sqrt{2d_h} d\delta(t) \exp(-\eta\beta_1\gamma t) + 2\sqrt{2d_h} \beta_2 \delta(t) \exp(-\eta\beta_2\gamma t) \right)$$

$$\cdot \left( 2\sqrt{2d_h} \beta_2 d\delta(t) \exp(-\eta\beta_2\gamma t) + \sqrt{2d_h} d\delta(t) \exp(-\eta\beta_1\gamma t) \right)$$

$$= 4 d_h d^2 \delta(t)^2 \exp(-2\eta\beta_1\gamma t) + 8\beta_2^2 d_h d\delta(t)^2 \exp(-2\eta\beta_2\gamma t)$$

$$+ 8\beta_2 d_h d^2 \delta(t)^2 \exp(-\eta(\beta_1 + \beta_2)\gamma t) + 4\beta_2 d_h d\delta(t)^2 \exp(-\eta(\beta_1 + \beta_2)\gamma t)$$

$$\leq 4 d_h d^2 \delta(t)^2 \exp(-\eta\beta_1\gamma t) + 28\beta_2^2 d_h d\delta(t)^2 \exp(-2\eta\beta_2\gamma t)$$

The last inequality is by $\beta_2 d \leq 2\beta_2^2$, $\beta_2 \leq \beta_2^2$, and $\exp(-2\eta\beta_1\gamma t) \leq \exp(-\eta\beta_1\gamma t)$, $\exp(-2\eta\beta_2\gamma t) \leq \exp(-\eta\beta_2\gamma t)$, $\exp(-\eta(\beta_1 + \beta_2)\gamma t) \leq \exp(-\eta\beta_2\gamma t)$.

$$\left| \bar{\boldsymbol{b}}_i^\top(t) \bar{\boldsymbol{b}}(t) \right| \leq \left\| \bar{\boldsymbol{b}}_i(t) \right\| \cdot \left\| \bar{\boldsymbol{b}}(t) \right\|$$

$$\leq 2\sqrt{2d_h} d\delta(t) \exp(-\eta\beta_1\gamma t) \cdot \left( 2\sqrt{2d_h} \beta_2 d\delta(t) \exp(-\eta\beta_2\gamma t) + \sqrt{2d_h} d\delta(t) \exp(-\eta\beta_1\gamma t) \right)$$

$$\leq 8\beta_2 d_h d^2 \delta(t)^2 \exp(-\eta\beta_2\gamma t) + 4 d_h d^2 \delta(t)^2 \exp(-\eta\beta_1\gamma t)$$

The last inequality is by $\exp(-2\eta\beta_1\gamma t) \leq \exp(-\eta\beta_1\gamma t)$, $\exp(-\eta(\beta_1 + \beta_2)\gamma t) \leq \exp(-\eta\beta_2\gamma t)$.

# D Discussion

In this section, we show that orthogonal initialization Mamba can be trained to ICL solution, and compare our method with some previous works.

**Orthogonal Initialization**   Now we assume that each column of $\boldsymbol{W}_B$ and $\boldsymbol{W}_C$ are initialized with orthogonal columns of unit norm. Then we have

$$\boldsymbol{C}^\top(0)\boldsymbol{C}(0) = \boldsymbol{B}^\top(0)\boldsymbol{B}(0) = \boldsymbol{I}, \quad \boldsymbol{B}^\top(0)\boldsymbol{b}(0) = \boldsymbol{C}^\top(0)\boldsymbol{b}(0) = \boldsymbol{0}.$$

Consider the following update rule as part of lemma 5.1.

$$\boldsymbol{B}(t+1) = \boldsymbol{B}(t) + \eta\beta_3\boldsymbol{C}(t) - \eta\beta_1\boldsymbol{C}(t)\boldsymbol{C}(t)^\top\boldsymbol{B}(t), \tag{45}$$

$$\boldsymbol{C}(t+1) = \boldsymbol{C}(t) + \eta\beta_3\boldsymbol{B}(t) - \eta\beta_1\boldsymbol{B}(t)\boldsymbol{B}(t)^\top\boldsymbol{C}(t) - \eta\beta_2\boldsymbol{b}(t)\boldsymbol{b}(t)^\top\boldsymbol{C}(t), \tag{46}$$

$$\boldsymbol{b}(t+1) = \boldsymbol{b}(t) - \eta\beta_2\boldsymbol{C}(t)\boldsymbol{C}(t)^\top\boldsymbol{b}(t). \tag{47}$$

By (Eq. (45)), (Eq. (46)) and (Eq. (47)), we have:

$$\boldsymbol{B}^\top(t+1)\boldsymbol{b}(t+1) = \boldsymbol{B}(t)^\top\boldsymbol{b}(t) + \eta\beta_3\boldsymbol{C}(t)^\top\boldsymbol{b}(t) - \eta\beta_1\boldsymbol{B}(t)^\top\boldsymbol{C}(t)\boldsymbol{C}(t)^\top\boldsymbol{b}(t)$$
$$- \eta\beta_2\boldsymbol{B}(t)^\top\boldsymbol{C}(t)\boldsymbol{C}(t)^\top\boldsymbol{b}(t) - \eta^2\beta_2\beta_3\boldsymbol{C}(t)^\top\boldsymbol{C}(t)\boldsymbol{C}(t)^\top\boldsymbol{b}(t) + \eta^2\beta_1\beta_2\boldsymbol{B}(t)^\top\boldsymbol{C}(t)\boldsymbol{C}(t)^\top\boldsymbol{C}(t)\boldsymbol{C}(t)^\top\boldsymbol{b}(t)$$

$$\boldsymbol{C}(t+1)^\top\boldsymbol{b}(t+1) = \boldsymbol{C}(t)^\top\boldsymbol{b}(t) + \eta\beta_3\boldsymbol{B}(t)^\top\boldsymbol{b}(t) - \eta\beta_1\boldsymbol{C}(t)^\top\boldsymbol{B}(t)\boldsymbol{B}(t)^\top\boldsymbol{b}(t) - \eta\beta_2\boldsymbol{C}(t)^\top\boldsymbol{b}(t)\boldsymbol{b}(t)^\top\boldsymbol{b}(t)$$
$$- \eta\beta_2\boldsymbol{C}(t)^\top\boldsymbol{C}(t)\boldsymbol{C}(t)^\top\boldsymbol{b}(t) - \eta^2\beta_2\beta_3\boldsymbol{B}(t)^\top\boldsymbol{C}(t)\boldsymbol{C}(t)^\top\boldsymbol{b}(t)$$
$$+ \eta^2\beta_1\beta_2\boldsymbol{C}(t)^\top\boldsymbol{B}(t)\boldsymbol{B}(t)^\top\boldsymbol{C}(t)\boldsymbol{C}(t)^\top\boldsymbol{b}(t) + \eta^2\beta_2^2\boldsymbol{C}(t)^\top\boldsymbol{b}(t)\boldsymbol{b}(t)^\top\boldsymbol{C}(t)\boldsymbol{C}(t)^\top\boldsymbol{b}(t)$$

Combining $\boldsymbol{B}^\top(0)\boldsymbol{b}(0) = \boldsymbol{C}^\top(0)\boldsymbol{b}(0) = \boldsymbol{0}$ with induction, we can derive that $\boldsymbol{B}^\top(t)\boldsymbol{b}(t) = \boldsymbol{C}^\top(t)\boldsymbol{b}(t) = \boldsymbol{0}$ for $t \geq 0$. Thus we only need to consider the following dynamics.

$$\boldsymbol{B}(t+1) = \boldsymbol{B}(t) + \eta\beta_3\boldsymbol{C}(t) - \eta\beta_1\boldsymbol{C}(t)\boldsymbol{C}(t)^\top\boldsymbol{B}(t), \tag{48}$$

$$\boldsymbol{C}(t+1) = \boldsymbol{C}(t) + \eta\beta_3\boldsymbol{B}(t) - \eta\beta_1\boldsymbol{B}(t)\boldsymbol{B}(t)^\top\boldsymbol{C}(t) \tag{49}$$

Denote $\boldsymbol{B}(t)^\top\boldsymbol{B}(t) = D(t)$, $\boldsymbol{C}(t)^\top\boldsymbol{C}(t) = E(t)$ and $\boldsymbol{C}(t)^\top\boldsymbol{B}(t) = F(t)$ then by (Eq. (48)) and (Eq. (49)), we have

$$\begin{aligned} \boldsymbol{F}(t+1) = {} & \boldsymbol{F}(t) + \eta\beta_3(\boldsymbol{D}(t) + \boldsymbol{E}(t)) - \eta\beta_1\boldsymbol{F}(t)\boldsymbol{D}(t) \\ & + \eta^2\beta_3^2\boldsymbol{F}(t)^\top - 2\eta^2\beta_1\beta_3\boldsymbol{F}(t)\boldsymbol{F}(t)^\top - \eta\beta_1\boldsymbol{E}(t)\boldsymbol{F}(t) \\ & + \eta^2\beta_1^2\boldsymbol{F}(t)\boldsymbol{F}(t)^\top\boldsymbol{F}(t) \end{aligned} \tag{50}$$

$$\begin{aligned} \boldsymbol{D}(t+1) = {} & \boldsymbol{D}(t) + \eta\beta_3(\boldsymbol{F}(t) + \boldsymbol{F}(t)^\top) - \eta\beta_1(\boldsymbol{F}(t)^\top\boldsymbol{F}(t) + \boldsymbol{F}(t)^\top\boldsymbol{F}(t)) \\ & + \eta^2\beta_3^2\boldsymbol{E}(t) - \eta^2\beta_1\beta_3\boldsymbol{F}(t)^\top\boldsymbol{E}(t) \\ & - \eta^2\beta_1\beta_3\boldsymbol{E}(t)\boldsymbol{F}(t) + \eta^2\beta_1^2\boldsymbol{F}(t)^\top\boldsymbol{E}(t)\boldsymbol{F}(t) \end{aligned} \tag{51}$$

$$\begin{aligned} \boldsymbol{E}(t+1) = {} & \boldsymbol{E}(t) + \eta\beta_3(\boldsymbol{F}(t) + \boldsymbol{F}(t)^\top) - \eta\beta_1(\boldsymbol{F}(t)^\top\boldsymbol{F}(t) + \boldsymbol{F}(t)^\top\boldsymbol{F}(t)) \\ & + \eta^2\beta_3^2\boldsymbol{D}(t) - \eta^2\beta_1\beta_3\boldsymbol{F}(t)^\top\boldsymbol{D}(t) \\ & - \eta^2\beta_1\beta_3\boldsymbol{D}(t)\boldsymbol{F}(t) + \eta^2\beta_1^2\boldsymbol{F}(t)^\top\boldsymbol{D}(t)\boldsymbol{F}(t) \end{aligned} \tag{52}$$

Note that $D(0) = E(0) = \boldsymbol{I}$ and $F(0) = \boldsymbol{0}$. By induction we can see that $D(t), E(t)$ and $F(t)$ are diagonal matrix for $t > 0$. Because of the symmetry, we have $D(t) = E(t)$. Now we denote $D(t) = E(t) = g(t)\boldsymbol{I}$ and $F(t) = h(t)\boldsymbol{I}$. Then based on (Eq. (50)), (Eq. (51)) and (Eq. (52)), we have:

$$g(t+1) = g(t) + \eta(2h(t) + \eta\beta_3 g(t) - \eta\beta_1 g(t)h(t))(\beta_3 - \beta_1 h(t)) \tag{53}$$

$$h(t+1) = h(t) + \eta g(t)(\beta_3 - \beta_1 h(t)) + \eta^2 h(t)(\beta_3 - \beta_1 h(t))^2 \tag{54}$$

Since $g(0) = 1$ and $h(0) = 0$ at initialization, $h(t)$ will converge to $\frac{\beta_3}{\beta_1}$ (i.e. $\boldsymbol{C}^\top\boldsymbol{B} \to \frac{\beta_3}{\beta_1}\boldsymbol{I}$).

**Compare with Other Techniques** (Eq. (45)), (Eq. (46)) and (Eq. (47)) can be viewed as the gradient descent that minimize the following target:

$$\frac{1}{2}\|\boldsymbol{C}^\top \boldsymbol{W}_B \boldsymbol{X} - \boldsymbol{Y}\|_F^2 \tag{55}$$

where $\boldsymbol{X}$ and $\boldsymbol{Y}$ satisfy:

$$\boldsymbol{X}\boldsymbol{X}^\top = \begin{bmatrix} \beta_1 & & & \\ & \ddots & & \\ & & \beta_1 & \\ & & & \beta_2 \end{bmatrix} \in \mathbb{R}^{(d+1)\times(d+1)}, \boldsymbol{X}\boldsymbol{Y}^\top = \begin{bmatrix} \beta_3 & & \\ & \ddots & \\ & & \beta_3 \\ & & \boldsymbol{0}_{1\times d} \end{bmatrix} \in \mathbb{R}^{(d+1)\times(d)}.$$

To establish convergence for this problem under gaussian initialization, Arora et al. (2019) require the standard deviation to be small enough, while Du and Hu (2019) require larger dimension $d_h$ because their method relies on the condition number of $\boldsymbol{X}$. Our method balances the requirements on initialization and dimension. The fine-grained nature of our analysis (particularly the *Vector-coupled Dynamics*) enables extension to various problem beyong (Eq. (55)).

# E  Additional Experimental Results

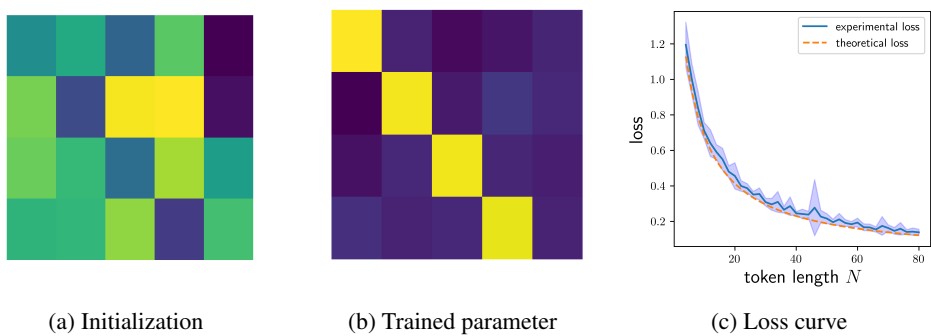

(a) Initialization            (b) Trained parameter            (c) Loss curve

Figure 2: (a) Visualization of matrix product $\boldsymbol{C}^\top \boldsymbol{W}_B$ before training; (b) Post-training visualization of matrix product $\boldsymbol{C}^\top \boldsymbol{W}_B$; (c) Test loss versus token sequence length $N$. Blue curve: experimental loss; orange dashed line: theoretical loss $\frac{d}{2}\left(1 - \frac{\beta_3^2}{\beta_1}\right)$.

**Experiments Setting**  We follow Section 3 to generate the dateset and initialize the model. Specifically, we set dimension $d = 4$, $d_h = 80$, prompt token length $N = 50$, and train the Mamba model on 3000 sequences by gradient descent. Moreover, we vary the length of the prompt token $N$ from 4 to 80 and compare the test loss with the theoretical loss. For each $N$, we conduct 10 independent experiments and report the averaged results. All experiments are performed on an NVIDIA A800 GPU.

**Experiment Result**  Figure 2a and Figure 2b show that $\boldsymbol{C}^\top \boldsymbol{B}$ can be trained to diagonal matrix from random initialization. Figure 2c show that the experimental loss aligns with the theoretical loss $\mathcal{L}(\boldsymbol{\theta}) = \frac{d}{2}\left(1 - \frac{\beta_3^2}{\beta_1}\right)$, noting that the theoretical loss $\left(1 - \frac{\beta_3^2}{\beta_1}\right)$ has an upper bound $\frac{3d(d+1)}{2N}$ that decays linearly with N. These experimental results further verified our theoretical proof.

**Mamba vs Linear Attention**  Optimal linear attention outperforms Mamba under our construction, and they have $O(1/N)$ error upper bound with different constant factors. We provide a comprarison of loss between optimal Mamba (under our Assumption 4.1) with optimal linear attention as in Table 2 with setting $d = 10, N = 10, 20, \ldots, 80$.

**When N is smaller than d**  We also test the case when $N \le d$ in Table 3 with setting $d = 20, N = 4, 6, \ldots, 20$.

**Convergence of $w_\Delta$**   We set $w_\Delta = 0$ in the assumption. Now we show that random initializd $w_\Delta = 0$ can converge to 0 experimental. The results is in Table 4.

**Different $d_h$**   Table 5 shows the mean value and standard deviation of the loss for smaller $d_h$ (in 10 repeated experiments). We set $d = 4, N = 30$, and the theoretical loss is 0.2954.

Table 2: Comparison of Mamba and Linear Attention

| N | 10 | 20 | 30 | 40 | 50 | 60 | 70 | 80 |
|---|---|---|---|---|---|---|---|---|
| **Mamba** | 2.6671 | 1.8189 | 1.3800 | 1.1117 | 0.9308 | 0.8005 | 0.7022 | 0.6254 |
| **Linear Attention** | 2.6190 | 1.7742 | 1.3415 | 1.0784 | 0.9016 | 0.7746 | 0.6790 | 0.6044 |

Table 3: Experiment for $N \leq d$

| N | 4 | 6 | 8 | 10 | 12 | 14 | 16 | 18 | 20 |
|---|---|---|---|---|---|---|---|---|---|
| **Experimental Loss** | 8.5911 | 7.8292 | 7.7009 | 6.8235 | 6.4004 | 6.0612 | 5.9689 | 5.6193 | 5.1426 |
| **Theoretical Loss** | 8.4484 | 7.8425 | 7.3173 | 6.8579 | 6.4526 | 6.0926 | 5.7706 | 5.4810 | 5.2190 |

Table 4: Convergence of $w_\Delta$

| Epoch | 0 | 10 | 20 | 30 | 40 | 50 | 60 | 70 | 80 |
|---|---|---|---|---|---|---|---|---|---|
| $\|w_\Delta\|_2$ | 0.8883 | 0.7513 | 0.4821 | 0.3331 | 0.2444 | 0.2026 | 0.1868 | 0.1799 | 0.1773 |
| $\|w_\Delta\|_2^2$ | 0.7891 | 0.5645 | 0.2324 | 0.1109 | 0.0597 | 0.0410 | 0.0349 | 0.0324 | 0.0314 |

Table 5: Different $d_h$

| $d_h$ | 6 | 8 | 10 | 12 | 14 | 16 | 18 | 20 |
|---|---|---|---|---|---|---|---|---|
| **mean(loss)** | 0.2912 | 0.2933 | 0.2899 | 0.2887 | 0.2929 | 0.2951 | 0.2967 | 0.2959 |
| **std(loss)** | 0.0075 | 0.0055 | 0.0116 | 0.0052 | 0.0105 | 0.0097 | 0.0110 | 0.0142 |

