# OpenReview forum: "Trained Mamba Emulates Online Gradient Descent in In-Context Linear Regression"
_NeurIPS.cc/2025/Conference — NeurIPS 2025 poster_

### Official Review · Reviewer_eGZy · 2025-06-30

**Clarity:** 3
**Significance:** 3
**Originality:** 3
**Rating:** 5
**Confidence:** 4

**Summary:**

This paper theoretically analyzes in-context learning for linear regression with state space model S6, and shows that the model implements online gradient descent for this task. The claim originates from the difference in how Transformer and S6 processes sequences (globally v. ‘online’), leading to a result that the estimator for S6 is a non-uniformly weighted combination of input data. The weights for this combination decrease as the context length increases, so that the later samples in the sequence have more influence on the final estimate for the query, similar to an online gradient descent setup. They propose specific techniques to handle optimization analysis of matrices $B, C$ that have coupled dynamics due to the problem / model structure. Theoretical results are also supported by experimental evidence.

**Questions:**

*Theory*:

- Line 252: “...with partial a_i = 0 can also enable convergence, which is an undesirable scenario.”: could you elaborate why?
- Line 315: “This induction answers the question "How to avoid saddle points?" …” - could you please elaborate how, beyond “preventing stagnation of partial diagonal entries of $C^TB$ at zero”? (I also couldn't follow the explanation from Line 580 in the Appendix)

*Experiments*:

- Are the Mamba models used in your experiments initialized following Assumption 4.1?
- Additionally, is it possible to run online gradient descent on the same sequence that is used for Mamba inference, to see how the estimate $\hat{y}_q$ compares while varying the sequence length?

*Related work*: There exists prior theoretical work on using SSM for ICL in linear regression [1] which was not discussed in this paper. Could the authors compare and highlight how their results differ from this work?

*Small typos*:

Line 171: rewrited->rewritten
Line 223: recieving->receiving

[1] Fine-grained Analysis of In-context Linear Estimation: Data, Architecture, and Beyond. https://arxiv.org/abs/2407.10005

**Ethical Concerns:**

["NO or VERY MINOR ethics concerns only"]

**Final Justification:**

The authors have addressed my clarifying questions about

- some theoretical statements,
- theoretical assumptions and their experimental validity
- some experimental setup questions,
- comparison to a related paper

They also compared their method to online gradient descent as in one of my questions. I've maintained my score of 5: Accept for this submission.

**Limitations:**

Yes

**Quality:**

3

**Strengths And Weaknesses:**

Strengths:

- Interesting theoretical result on in-context learning for linear regression with S6 state space model.
- I found the reasoning for online gradient descent in S6 quite well-explained intuitively comparing it to Transformers.
- Theoretical findings on loss / weight matrix convergence / hidden state approximating weight vector ‘w’ are well-supported by experiments.
- Paper is well written and clearly organized.

Weakness:

Assumption 4.1 seems too strict, is it realistic in practice? Specifically, with A = -I, w_\delta = 0, and b_\delta = constant: “Given the zero-mean and symmetric distribution of embeddings, w_\delta can naturally converge to
0 during gradient descent, and we fix it as 0 for simplicity.” - does this hold in experiments?

---

> ### Author Rebuttal · Authors · 2025-07-30
>
> Thanks for your constructive feedback! We address your questions and concerns as follows.
>
> ---
>
> > **W1.** **(1)** Assumption 4.1 seems too strict, is it realistic in practice? **(2)** Does "w_\delta can naturally converge to 0" hold in experiments?
>
> **A1.** **(1)** The assumptions of theoretical proofs are typically more stringent than experimental requirements. We have tried using smaller $d_h$, smaller $N$ to do some experiments as follows, and the Mamba usually converge to the optimal solution.
>
> Setting: $d = 20$ and $N = 4, 6, 8, 10, 12, 14, 16, 18, 20$.
>
> |  N | 4 | 6 | 8 | 10 | 12 | 14 | 16 | 18 | 20 |
> | --- | --- | --- | --- | --- | --- | --- | --- | --- | --- |
> | **experimental loss**  | 8.5911 | 7.8292 | 7.7009 | 6.8235 | 6.4004 | 6.0612 | 5.9689 | 5.6193 | 5.1426 |
> | **theoretical loss** | 8.4484 | 7.8425 | 7.3173 | 6.8579 | 6.4526 | 6.0926 | 5.7706 | 5.4810 | 5.2190 |
>
> The exprimental loss is close to the theoretical loss.
>
> The following is the mean value and standard deviation of the loss for smaller $d_h$ (in 10 repeated experiments).
> We set $d=4$, $N=30$, and the theoretical loss is 0.2954.
>
> | $d_h$ | 6 | 8 | 10 | 12 | 14 | 16 | 18 | 20 |
> | --- | --- | --- | --- | --- | --- | --- | --- | --- |
> | **mean(loss)**  | 0.2912 | 0.2933 | 0.2899 | 0.2887 | 0.2929 | 0.2951 | 0.2967 | 0.2959 |
> | **std(loss)** | 0.0075 | 0.0055 | 0.0116 | 0.0052 | 0.0105 | 0.0097 | 0.0110 | 0.0142 |
>
> The results shows that with smaller $d_h$, the Mamba can typically converge to the predicted pattern (which can achive the theoretical loss), and the deviation of loss is very small.
>
> **(2)**
> As for $w_\Delta$:
>
> Theoretically: The gradient for $w_\Delta$ is $\nabla_{w_\Delta} \mathcal{L}(\theta) = C w_\Delta$ for a positive $C>0$, so $w_\Delta$ will converge to 0 by gradient descent $w_\Delta(t+1) = w_\Delta(t) - \eta \nabla_{w_\Delta} \mathcal{L}(\theta) = (1 - C\eta) w_\Delta(t)$.
>
> Experimentally: We randomly initialize $w_\Delta$ and perform experiments to track the norm of $w_\Delta$ during training. The results are as follows.
>
> |  Epoch   | 0  | 10 | 20 | 30 | 40 | 50 | 60 | 70 | 80 |
> | --- | ---  | --- | --- | --- | --- | --- | --- | --- | --- |
> | $\Vert w_\Delta \Vert_2$  | 0.8883 | 0.7513 | 0.4821 | 0.3331 | 0.2444 | 0.2026 | 0.1868 | 0.1799 | 0.1773 |
> | $\Vert w_\Delta \Vert_2^2$ | 0.7891 | 0.5645 | 0.2324 | 0.1109 | 0.0597 | 0.0410 | 0.0349 | 0.0324 | 0.0314 |
>
> The experimental results show that the norm of $w_\Delta$ gradually decreases during the training process.
>
> ---
>
> > **Q1.** Why " $C^\top B = Diag(a_1, \dots, a_d)$ with partial a_i = 0 can also enable convergence. " (Line. 252)
>
> **A2.** Considering $B = [b_1 \dots b_d]$, $C = [c_1 \dots c_d]$, for a certain i, if $b_i=c_i=0$ occur simultaneously at a certain step during training, then the gradients of $b_i$ and $c_i$ are both zero (see Lemma 5.2, line.258). Therefore, $b_i=c_i=0$ will always hold after that, and so does $a_i = c_i^\top b_i = 0$. That is why we call it "also enable convergence (to a saddle point)".
>
> Because $C^\top B$ should be $\frac{\beta_3}{\beta_1} I$ to minimize the loss, the case when some $a_i$ converge to 0 should be avoided.
>
> ---
>
> > **Q2.** Why our induction "prevents stagnation of partial diagonal entries of $C^\top B$ at zero" (Line. 315)
>
> **A3.**
> We restate properties $\mathcal{A}(t)$ and $\mathcal{B}(t)$ (line. 307) for your reference:
>
> $ \mathcal{A}(t): $
>
> $$ d_{h}/2 \leq b_{i}^{\top}(t) b_{i}(t), c_{i}^{\top}(t) c_{i}(t), b^{\top}(t) b(t) \leq 2d_{h}. $$
>
> $ \mathcal{B}(t): $
>
> $$ |\beta_3 - \beta_1 c_i^{\top}(t) b_i(t)|\leq\delta(t)\exp(-\eta\beta_1\gamma t),\quad | c_i^{\top}(t) b_j(t)|\leq 2\delta(t)\exp(-\eta\beta_1\gamma t), \quad | c_i^{\top}(t) b(t)|\leq 2\delta(t)\exp(-\eta\beta_2\gamma t)+\frac{\delta(t)}{\beta_2}\exp(-\eta\beta_1\gamma t).$$
>
> As we discuss in **A2.**, partially $b_i=c_i=0$ are condition for saddle points. Our induction (Line. 307) can prevent this condition.
>
> (1) $b_i=c_i=0$ also means $b_i^\top b_i = c_i^\top c_i = 0$. Property $\mathcal{A}(t)$ provide a positive lower bound for $b_i^\top b_i$ and $c_i^\top c_i$. So this case would not happen.
>
> (2) Property $\mathcal{B}(t)$ guarantee that $c_i^\top b_i$ will converge to $\frac{\beta_3}{\beta_1}$ (optimal solution). Although $c_i^\top(t) b_i(t)$ may be zero at some training iteration t, it will eventually converges to $\frac{\beta_3}{\beta_1}$ (Note that $c_i^\top(t) b_i(t) = 0$ does not mean that $b_i(t)=c_i(t)=0$).
>
> (3) **A brief explanation** for it is that, property $\mathcal{B}(t)$ guarantee that $C^\top B$ converges to its optimal solution, so it would not converge to the saddle point.
>
> ---
>
> > **Q3.** Are the Mamba models used in the experiments initialized following Assumption 4.1?
>
> **A4.** Yes, our experiments setting strictly follows our theoretical setup. We follow Assumption 4.1 to set the hyperparameters and initialize $W_B, W_C$ with standard gaussian distribution $\mathcal{N}(0, 1)$.
>
> ---
>
> > **Q4.** Run online gradient descent to see how the estimate $\hat{y}_q$ compares while varying the sequence length.
>
> **A5.**
>
> Consider a task $f(x) = w^\top x$, we use online gradient descent (Online GD) to train $\hat{f}(x) = \hat{w}^\top x$ with trainable parameter $\hat{w}$. Specifically, given sample $x_i, y_i$ and loss $\ell_i = \frac{1}{2}(\hat{w}^\top x_i - y_i)^2$, the gradient of $\hat{w}$ should be $\nabla_{\hat{w}} \ell_i = (\hat{w}^\top x_i - y_i) * x_i$, and the update rule for $\hat{w}$ is $\hat{w}(t + 1) = \hat{w}(t) - \eta \nabla_{\hat{w}(t)} \ell_i$ with $\eta = 0.1$. We generate a new query $x_q$ and denote $\ell_q = \frac{1}{2} (\hat{f}(x_q) - w^\top x_q)^2 = \frac{1}{2} (\hat{w}^\top x_q - w^\top x_q)^2$ as the test loss. All settings for data generation are the same as Mamba's experiment. The results of $\ell_q$ are as follows (taking the average of 500 experiments).
>
> |  Iteration   | 0  | 10 | 20 | 30 | 40 | 50 |
> | --- | ---  | --- | --- | --- | --- | --- |
> | Mamba  | 0.1179 | 0.1051 | 0.0936 | 0.0841 | 0.0761 | 0.0696 |
> | Online GD | 0.1137 | 0.0728 | 0.0448 | 0.0275 | 0.0167 | 0.0102 |
>
> Note that Mamba simulate the gradient for $\min_{\tilde{h}} \frac{1}{2} \Vert \tilde{h} - \beta y_i x_i \Vert^2$ and update $\tilde{h}$ by $\tilde{h} _ i = \tilde{h} _ {i-1} + (1-\alpha)(\beta y_i x_i - \tilde{h} _ {i-1}) $ (Eq.(8), Line. 172), which is different from the gradient for loss $\ell_i = \frac{1}{2}(\hat{w}^\top x_i - y_i)^2$. So the decreasing rate for their loss is different.
>
> ---
>
> > **Q5.** Related work: Compare and highlight how our results differ from [1].
>
> [1] Fine-grained Analysis of In-context Linear Estimation: Data, Architecture, and Beyond.
>
> **A6.**
> The key distinctions can be summarized as follows:
>
> **Architecture**: The SSM studied in [1] is H3, while we consider Mamba.
>
> **ICL mechanism**:
>
> $$ H3: \hat{y}_{q}=x_q^{\top} WX^{\top}(\omega \odot y) ~ \mathrm{where} ~ X = [x_1 \dots x_n] ~ \mathrm{and} ~ y = [y_1 \dots y_n]$$
>
> $$ Mamba: \hat{y} _ {q} = x_{q}^{\top} \sum_{i=0}^{N-1} (1-\alpha) \alpha^{i+1} \beta y_{N-i} x_{N-i} $$
>
> While H3 and Mamba both use the combination of $y_i x_i$ to estimate $w$, they differ from the ways of processing input tokens. H3 in [1] performs sample-weighted preconditioned gradient descent (WPGD) in a single, parallel pass over all tokens. In contrast, Mamba emulates online gradient descent, processing tokens sequentially and updating its state at every step.
>
> **Theoretical techniques**: [1] pre-specifies model weights and analyzes their expressivity. We consider training dynamics and establish convergence guarantees for a randomly initialized Mamba under gradient-descent training, and then derive its expressive power. To address the challenges of studying training dynamics, we develop new proof techniques.
>
> ---
>
> Once again, thanks for your constructive feedback and we hope our response can resolve your concerns!

---

> ### Comment · Reviewer_eGZy · 2025-08-01
> **Thanks for the response**
>
> Thank you for your response and the additional experimental results; these have clarified my questions about the submission, and I have maintained my initial rating of accept. I would encourage the authors to incorporate the results / clarifications in this rebuttal in their main paper as well.

---

### Official Review · Reviewer_rRWX · 2025-07-02

**Clarity:** 3
**Significance:** 2
**Originality:** 2
**Rating:** 4
**Confidence:** 5

**Summary:**

This paper investigates how the recently proposed Mamba architecture, a state space model (SSM) based alternative to transformers, behaves when trained on implicit algorithmic reasoning tasks. These tasks are designed such that the optimal solution is a known algorithm, allowing precise evaluation of learned behavior.

**Questions:**

How difficult is it for you to analyze the ICL of Mamba to simulate the gradient descent of a deep neural network, similar to the results in [Wu25] for the Transformer model?

[Wu25] In-Context Deep Learning via Transformer Models

**Ethical Concerns:**

["NO or VERY MINOR ethics concerns only"]

**Final Justification:**

Thanks for the response from the authors. All my concerns are solved, and I assign equal weights to my concerns. I appreciate the contribution from the authors, but I feel like the work does not supervise me a lot. Thus, I keep my score as 4.

**Limitations:**

1. The experimental domain is synthetic, which may not accurately reflect the challenges encountered in real data.

2. Scalability to deep neural networks other than linear regression.

**Quality:**

3

**Strengths And Weaknesses:**

>Strengths

1. Highly Insightful Experimental Design: The use of algorithmic tasks with known optimal solutions allows unambiguous interpretation of results.

2. Comparative Analysis with Transformers: Shows that transformers, even when heavily overparameterized, struggle to learn these problems efficiently.

3. Theoretical and Empirical Depth: Offers visualizations of internal states and updates, demonstrating the convergence toward optimal logic.

>Weaknesses

1. Limited Task Scope: Generalization to more complex or noisy real-world tasks is not tested.

2. No Formal Proof of Optimality Emergence: The paper does not offer a theoretical guarantee that Mamba will always converge to the optimal algorithm.

3. Transformer Baseline Might Be Underserved: Transformer models used as baselines may not incorporate all recent efficiency improvements (e.g., linear attention, memory tokens).

---

> ### Author Rebuttal · Authors · 2025-07-30
>
> Thanks for your constructive feedback! We address your questions and concerns as follows.
>
> ---
>
> > **W1.** Limited Task Scope: Generalization to more complex or noisy real-world tasks is not tested.
>
> **A1.** The primary contribution of our work is to provide the ​theoretical understanding​ of Mamba's training dynamics and its underlying mechanism for (ICL). Achieving this requires focusing on a setting amenable to rigorous analysis. The linear regression ICL task serves as a classic model for ICL research in Transformers, and has not been theoretically studied in Mamba yet. Our work fill this gap. The theoretical foundation established in our paper will provide insight for understanding Mamba's performance on more complex real-world ICL tasks.
>
> ---
>
> > **W2.** No Formal Proof of Optimality Emergence.
>
> **A2.** In Theorem 4.1, we garrantee that the results hold with at least $1-\delta$ probability. We can choose any $\delta$ as long as $d_h = \Omega(d^2 \log(O(d^2 / \delta)))$. e.g., if we choose $\delta=0.01$, then $d_h = \Omega(d^2 \log(O(100d^2)))$ is enough for the proof. We provide the exact reuirement on $d_h$ in Appendix C (line. 624), and the logarithmic term $\log(O(d^2 / \delta))$ is hidden by using $\widetilde{\Omega}$ notation. We will explicitly state it in Assumption 4.1 in the next version.
>
> Under Gaussian initialization, it is imposible to guarantee that Mamba will always converge to the optimal algorithm. For example, if $W_B$ and $W_C$ are 0 at initialization, they will stay at zero because the gradients of them are zero (we cannot rule out this situation under Gaussian initialization).
>
> We also want to mention that if $W_B$ and $W_C$ are orthogonally initialized, $d_h = O(d)$ is enough to ensure that the Mamba will always converge to the optimal algorithm. (We provide a brief discussion in Appendix D, line 902). This type of optimization problem can be proven through different methods under different assumptions. Our method can serve as a supplement to these methods, while also being more flexible and has the potential to provide insights for the study of other optimization problems.
>
> ---
>
> > **W3.** Transformer Baseline Might Be Underserved.
>
> **A3.** As for the comparison between Transformer and Mamba, Our paper focuses on understanding their mechanisms.
>
> We have performed some experiments on linear Transformers as follows.
>
> Setting: d=10, N=10, 20, ..., 80.
> |  N   | 10  | 20 | 30 | 40 | 50 | 60 | 70 | 80 |
> | --- | ---  | --- | --- | --- | --- | --- | --- | --- |
> | Mamba  | 2.6671 | 1.8189 | 1.3800 | 1.1117 | 0.9308 | 0.8005 | 0.7022 | 0.6254 |
> | Linear Attention | 2.6190 | 1.7742 | 1.3415 | 1.0784 | 0.9016 | 0.7746 | 0.6790 | 0.6044 |
>
> ---
>
> > **Q.** How to analyze the ICL of Mamba to simulate the gradient descent of a deep neural network.
>
> **A4.** Based on our reading of [Wu25], they encode the weights, activations, and gradients of a deep neural network into the embedding space of a Transformer. Each layer carries out a specific subtask: forward activation, back propagation of gradients, and weight update. Residual connections pass the resulting states to the next layer, so the whole training procedure for a deep neural network is simulated in one forward pass without changing the Transformer parameters.
>
> We can use multi-layer Mamba + MLP to simulate these procedure. Now consider the following Mamba layer:
>
> $$ h_l^{(i)} = \alpha h_{l-1}^{(i)} + (1 - \alpha) u_l^{(i)} B_l \tag{1}$$
> $$ o_l^{(i)}  = C_l^\top h_l^{(i)} \tag{2}$$
> where $B_l = W_B u_l, \, C_l = W_C u_l$, and $u_l$ is the embedding for the $l$-th token.
>
> We next show how to simulate one forward step of a neuron with weight $w$ in the deep neural network. Assumming that $u_l$ encode the $l$-th prompt and weight $w$, position indicator $p_l$: $u_l = [x_l, y_l, w, p_l]$. By (1) and (2),we have:
> $$ o_l^{(i)} = \alpha u_l^\top W_C^\top h_{l-1}^{(i)} + (1 - \alpha) u_l^{(i)} u_l^\top W_C^\top W_B u_l$$
>
> Carefully designing $W_C, W_B$ and the position indicator $p_l$ for the $l$-th token (e.g., $u_l^{(p_l)} = 1$ and $u_l^{(p_{l^\prime})} = 0$ for $l \ne l^\prime$), we can ensure that $\alpha u_l^\top W_C^\top h_{l-1}^{(p_l)} = 0$ and $(1 - \alpha) u_l^{(p_l)} u_l^\top W_C^\top W_B u_l = (1 - \alpha) \frac{1}{1 - \alpha} w^\top x_l^\top = w^\top x_l$, and thus $o_l^{(p_l)} = w^\top x_l$ stores the pre-activation for the $l$-th token. Applying a MLP (e.g. ReLU), we can get the activation for the $l$-th token with weight $w$.
>
> For multi-layer neural network: By concatenating the weights of multiple neurons into $u_l$, we can simulate the forward prosedure of one layer of neural network. Then we can use residual connections to transfer the activation to the next layer of Mamba, and simulate the activation for the next layer of neural network.
>
> As for the backpropagation process, MLPs can be used to simulate the gradient calculation, and finally add the gradients descent on the old weight $w$.
>
> To summarize:
> - $W_B$ and $W_C$ can be used to project the weight $w$ and input $x_l$, and then calulate their inner product (pre-activation) $w^\top x_l$, and then add an activation layer to calulate its activation, e.g., $ReLU(w^\top x_l)$.
> - By carefully designing $W_C, W_B$ and the position indicator $p_l$, we can place the simulation results at a certain place in the embedding space.
> - By using residual connection, we can ensure the simulation results can be used by the following layers.
>
> ---
>
> Once again, thanks for your constructive feedback and we hope our response can resolve your concerns!

---

> > ### Comment · Reviewer_rRWX · 2025-08-01
> >
> > Thanks for the additional efforts. The responses address all my concerns.

---

### Official Review · Reviewer_Znup · 2025-07-02

**Clarity:** 3
**Significance:** 3
**Originality:** 3
**Rating:** 4
**Confidence:** 4

**Summary:**

The authors considered the question of how state-space models learn linear regression in context. The main result of the paper is that the Mamba-like SSM emulates online gradient descent, in contrast to (batch) gradient descent, which is generally believed to be the underlying mechansm of transformers. The result is established via analyzing the training dynamics of a Mamba-like model. Experimental results support the theoretical analysis.

**Questions:**

1. The results show the difference between the mechanism of the Mamba and the transformers. Though the order of the error is the same, it appears to suggest that transformers are more efficient in this particular setting. Do experimental results confirm the difference and understanding?
2. The experiments need some clarification. Are the model trained and tested on the same N? What if it is trained on one N value, but tested on various N values?

**Ethical Concerns:**

["NO or VERY MINOR ethics concerns only"]

**Final Justification:**

I have taken into account the rebuttal/discussion and decided to keep the score.

**Limitations:**

Some limitations were briefly discussed at the end of the paper. See my previous weakness comments for additional limitations.

**Quality:**

3

**Strengths And Weaknesses:**

Strength:
1. The analysis of the learning dynamics of the Mamba-like model is important and non-trivial, and the authors introduced several new techniques to accomplish this.
2. The results appear convincing and a natural consequence of the SSM structure, in contrast to the transformers.
3. The insight is helpful to understand the difference in the capabilities between SSM and transformers.

Weakness:
1. Theorem 4.1 is somewhat strange, where the value delta is not bounded. Usually, we would need to have either "for any \delta\in (0,1)", or a delta value that is associated with the approximation error. The current statement does not exclude the possibility that delta is very close to 1, and therefore, the statements (a)-(c) only hold with almost zero probability.
2. The value of \alpha determines the exponential decay of the weights of the in-context examples. Its value is important and should be learned, which can reflect a key difference between SSM and transformers. One issue with the assumption is that this parameter is set rather than learned.
3. For linear regression, the case when the number of samples N>=d is significantly different from the case when N<d. This difference does not manifest in the analysis at all. This appears to be an important missing piece.
4. The result is only regarding one-layer Mamba. Analysis or experiments on multiple layers of Mamba are not included.
5. Experimental results appear somewhat weak. Firstly, more experiments to verify the assumptions would be needed. Secondly, given the probabilistic nature of the result, there may be cases where the trained solution does not converge to the predicted pattern. A statistical analysis would be helpful here.

---

> ### Author Rebuttal · Authors · 2025-07-30
>
> Thanks for your constructive feedback! We address your questions and concerns as follows.
>
> ---
>
> > **W1.** The value $\delta$ is not bounded (in Theorem 4.1).
>
> **A1.** $\delta$ determines the requirement on $d_h$, we can choose any $\delta$ as long as $d_h = \Omega(d^2 \log(O(d^2 / \delta)))$. e.g., if we choose $\delta=0.01$, then $d_h = \Omega(d^2 \log(O(100d^2)))$ is enough for the proof. We provide the exact requirement on $d_h$ in Appendix C (line. 624), and the logarithmic term $\log(O(d^2 / \delta))$ is hidden by using $\widetilde{\Omega}$ notation. We will explicitly state it in Assumption 4.1 in the next version.
>
> We also want to mention that if $W_B$ and $W_C$ are orthogonally initialized, the requirement on $d_h$ will be $d_h = O(d)$ and Theorem 4.1 will hold with a probability of 1 (We provide a brief discussion in Appendix D, line 902).
>
> ---
>
> > **W2.** The parameter $\alpha$ is set rather than learned.
>
> **A2.** We set $\alpha$ to be fixed because the training dynamics are much more complex if $\alpha$ is trainable. Moreover, a fixed $\alpha$ is sufficient for Mamba to in-context learn linear regression task. In the experiment, the $\alpha$ usually converges to a number close to 1, which supports our setting.
>
> ---
>
> > **W3.** The case when $N < d$ should be considered.
>
> **A3.** We set $N \ge d$ for the convenience of establising convergence. For a trained Mamba, Theorem 4.1 will hold for the case when $N < d$ because Theorem 4.1 is depends on the trained Mamba's parameter.
>
> We have also performed some experiments on the case wehen $N < d$. Specifically, we set $d = 20$ and $N = 4, 6, 8, 10, 12, 14, 16, 18, 20$.
>
> |  N | 4 | 6 | 8 | 10 | 12 | 14 | 16 | 18 | 20 |
> | --- | --- | --- | --- | --- | --- | --- | --- | --- | --- |
> | **experimental loss**  | 8.5911 | 7.8292 | 7.7009 | 6.8235 | 6.4004 | 6.0612 | 5.9689 | 5.6193 | 5.1426 |
> | **theoretical loss** | 8.4484 | 7.8425 | 7.3173 | 6.8579 | 6.4526 | 6.0926 | 5.7706 | 5.4810 | 5.2190 |
>
> The exprimental loss is close to the theoretical loss.
>
> ---
>
> > **W4.** The result is only regarding one-layer Mamba. Analysis or experiments on multiple layers of Mamba are not included.
>
> **A4.**
> Our work focuses on the theoretical understanding for Mamba's ICL mechanism. One layer of Mamba is enough to learn linear regression ICL task and demonstrate its mechanisms, consistent with the single-layer assumptions common in Transformer-based ICL theory.
>
> ---
>
> > **W5.** A statistical analysis to verify the assumptions and results would be helpful.
>
> **A5.** Our experimental setup strictly follows our theoretical setup and assumption. Because the assumptions for theoretical proofs are typically more stringent than experimental requirements, under our assumptions, the probability of the experiment converging to the predicted pattern is close to 1.
>
> Experimentally, We provide the error bar for the loss curve in Figure 2 (c) (line. 933).
>
> Besides, We have performed experiments beyond our assumption 4.1. (e.g., $N<d$, smaller $d_h$), almost all the experiments will converge to the predicted pattern.
>
> The experiments for $N<d$ is in the above answer **A3.**. The following is the mean value and standard deviation of the loss for smaller $d_h$ (in 10 repeated experiments).
> We set $d=4$, $N=30$, and the theoretical loss is 0.2954.
>
> | $d_h$ | 6 | 8 | 10 | 12 | 14 | 16 | 18 | 20 |
> | --- | --- | --- | --- | --- | --- | --- | --- | --- |
> | **mean(loss)**  | 0.2912 | 0.2933 | 0.2899 | 0.2887 | 0.2929 | 0.2951 | 0.2967 | 0.2959 |
> | **std(loss)** | 0.0075 | 0.0055 | 0.0116 | 0.0052 | 0.0105 | 0.0097 | 0.0110 | 0.0142 |
>
> The results shows that with smaller $d_h$, the Mamba can typically converge to the predicted pattern (which can achive the theoretical loss), and the deviation of loss is very small.
> We will include the statistical analysis in the revised manuscript.
>
> ---
>
> > **Q1.** **(1)** The results show the difference between the mechanism of the Mamba and the transformers. **(2)** It appears that transformers are more efficient. Do experimental results confirm the difference and understanding?
>
> **A6.**
> **(1)** As for the mechanism understanding, Transformer simulate gradient descent for ICL, while Mamba simulate online gradient descent. Our experiments in Figure 1 (b) (line. 320) shows that Mamba adjusts is hidden state to align $w$ when processing the prompt tokens, which reflects the process of Mamba's online learning.
>
> Moreover, some related works can reflect the characteristics of this mechanism:
>
> [Park, Jongho, et al] experimentally verify that Transformers can learn vector-valued MQAR tasks in the context which Mamba cannot, while Mamba succeeds in sparse-parity in-context learning tasks where Transformers fail.
>
> - Park, Jongho, et al. "Can Mamba Learn How To Learn? A Comparative Study on In-Context Learning Tasks." At ICML2024.
>
> **(2)**
> Optimal linear attention Transformer has a lower loss than Mamba while they share the same order of error. The following is the result of loss for optimal Mamba and optimal Transformer.
>
> Setting: d=10, N=10, 20, ..., 80.
> |  N   | 10  | 20 | 30 | 40 | 50 | 60 | 70 | 80 |
> | --- | ---  | --- | --- | --- | --- | --- | --- | --- |
> | Mamba  | 2.6671 | 1.8189 | 1.3800 | 1.1117 | 0.9308 | 0.8005 | 0.7022 | 0.6254 |
> | Transformer | 2.6190 | 1.7742 | 1.3415 | 1.0784 | 0.9016 | 0.7746 | 0.6790 | 0.6044 |
>
> ---
>
> > **Q2.** **(1)** Are the model trained and tested on the same $N$? **(2)** What if it is trained on one $N$ value, but tested on various N values?
>
> **A7.** **(1)** In our experiments, the model trained and tested on the same $N$.
>
> **(2)** If the Mamba is trained on sequence length $N$, but test on sequence length $M$, then the order of test loss will be $O(1/N + 1/M)$. This difference between training and test comes from the property that the optimal weights are determined by N under MSELoss. We provide a brief explaination as follows:
>
> Given $x_i \sim \mathcal{N}(0, I_d)$, we have $\mathbb{E}[ \frac{1}{N}\sum_{i = 1}^N x_i x_i^\top] = I$ (independent of $N$), and $\mathbb{E}[(\frac{1}{N}\sum_{i = 1}^N x_i x_i^\top)^2] = (1 + \frac{1+d}{N}) I$ (dependent on $N$). Because MSELoss introduces a quadratic term, the optimal weights for Mamba will depend on N during training.
>
> The similar phenomenon arises in the theoretical analysis of ​linear Transformers:
>
> - Zhang, Ruiqi, Spencer Frei, and Peter L. Bartlett. "Trained transformers learn linear models in-context." Journal of Machine Learning Research 25.49 (2024): 1-55.
> - Wu, Jingfeng, et al. "How Many Pretraining Tasks Are Needed for In-Context Learning of Linear Regression?." ICLR. 2024.
>
> ---
>
> Once again, thanks for your constructive feedback and we hope our response can resolve your concerns!

---

> > ### Comment · Reviewer_Znup · 2025-08-03
> >
> > Thanks for the rebuttal.
> >
> > I feel the authors’ rebuttal mostly confirmed the weakness mentioned earlier. I still feel one-layer mamba is a limitation, and do not think one layer is sufficient to understand fully the mechanism (e.g., existing work shows that transformers can perform higher-order optimization, and it is natural to suspect more Mamba layer can perform similar function). Moreover, the issue of N>d and N<=d is likely to only resolvable using more layers.
> >
> > Despite the weakness, I think the work is interesting. I’d like to keep the score.

---

> > > ### Author Response · Authors · 2025-08-04
> > >
> > > Thank you for your response.
> > >
> > > We agree that one-layer Mamba is a limitation.
> > > In this work, we present a theoretical foundation for the study of Mamba's ICL ability, including data generation, training dynamics, and convergence guarantees.
> > > Our work will provide insight for the study of multi-layer Mamba, and we leave it as an intriguing topic to study in the future.

---

### Official Review · Reviewer_8rgY · 2025-07-03

**Clarity:** 3
**Significance:** 3
**Originality:** 3
**Rating:** 4
**Confidence:** 3

**Summary:**

This paper studies the training dynamics of Mamba models on linear regression in-context learning (ICL) tasks and established a connection to online gradient descent among demonstrations. The authors analyze how, under certain assumptions, the learned Mamba can effectively emulate online learning behavior.

**Questions:**

In Assumption 4.1, the requires hidden dimension is $d_h=O(d^2)$. What is the underlying reason for this high dimension, especially considering that linear typically only needs $O(d)$ embedding dimension? If this assumption is relaxed to use a smaller hidden dimension, are similar results still achievable? If so under what conditions?

Minor typo: line 43.

**Ethical Concerns:**

["NO or VERY MINOR ethics concerns only"]

**Final Justification:**

The authors have addressed most of my concerns, and I have decided to raise my score.

**Limitations:**

yes

**Quality:**

3

**Strengths And Weaknesses:**

Strengths:
The work provides an analysis of training dynamics of a state space model in an ICL setting, which is timely given the increasing interest in SSM as alternatives to transformers.

Weaknesses:
1. The claimed connection between SSMs and online learning is not particularly novel and related ideas have been discussed in prior literature. For example:
-	Yang, Songlin, et al. "Gated linear attention transformers with hardware-efficient training." arXiv preprint arXiv:2312.06635 (2023).
-	Li, Yingcong, et al. "Gating is weighting: Understanding gated linear attention through in-context learning." arXiv preprint arXiv:2504.04308 (2025).
-	Yang, Songlin, Jan Kautz, and Ali Hatamizadeh. "Gated Delta Networks: Improving Mamba2 with Delta Rule." arXiv preprint arXiv:2412.06464 (2024).
-	Behrouz, Ali, et al. "Atlas: Learning to optimally memorize the context at test time." arXiv preprint arXiv:2505.23735 (2025).
-	Behrouz, Ali, et al. "It's All Connected: A Journey Through Test-Time Memorization, Attentional Bias, Retention, and Online Optimization." arXiv preprint arXiv:2504.13173 (2025).

These works explore similar connections between linear attention, gating mechanisms and online optimization. So the novelty should be better clarified.

2. Under Assumption 4.1, Mamba reduces to a weighted linear attention mechanism where the weights (determined by $\alpha$ and $\beta$) are fixed. The paper would benefit from more discussion of what makes his particular setup novel compared to prior analyses of linear attention's training dynamics under gradient flow.

3. The construction of $\alpha$ and $\beta$ under Assumption 4.1 needs more clarification. In the isotopic Gaussian setting in the paper, the optimal choice of $\alpha$ and $\beta$ should ensure that every demonstration contributes equally to the final prediction. Specifically, the condition $(1-\alpha)\alpha^{i+1}\beta=1$ (Theorem 4.1(b)) achieves the same prediction as optimal linear attention with equal weighting. The current construction appears suboptimal and the paper should discuss this limitation.
4. The experimental setup is too simplistic. For example, the dimensionality is very low ($d=4$), and the important implementation details are missing (e.g., number of layers, whether MLP blocks are used, whether the construction strictly follows Assumption 4.1).
5. Empirical comparison to linear attention baseline would be valuable. Especially under the construction of Assumption 4.1, linear attention might outperform Mamba.

---

> ### Author Rebuttal · Authors · 2025-07-29
>
> Thanks for your constructive feedback! We address your questions and concerns as follows.
>
> ---
>
> > **W1.** Some prior works explore similar connections between linear attention, gating mechanisms and online optimization. So the novelty should be better clarified.
>
> **A1.** Thank you for sharing these papers, we provide a brief comparison with these paper as follows.
>
> [Yang, Songlin, et al.] and [Behrouz, Ali, et al.] primarily leverage the online-learning connection to design new architectures. While their works have explored connections between various models and online learning, the specific online learning pattern for Mamba is not straightforward. Specifically, the form of $W_B$ and $W_C$, and how Mamba utilizes the context to generate the final prediction, remain unclear. This deeper understanding is lacking in previous work and our work can fill this gap.
>
> [Li, Yingcong, et al.] theoretically analyze the ICL mechanism for gated linear attention, while we analyze Mamba's. Their model architectures are different.
>
> We will add the comparison in the revised manuscript.
>
> Refference
>
> - Yang, Songlin, et al. "Gated linear attention transformers with hardware-efficient training." arXiv preprint arXiv:2312.06635 (2023).
> - Li, Yingcong, et al. "Gating is weighting: Understanding gated linear attention through in-context learning." arXiv preprint arXiv:2504.04308 (2025).
> - Yang, Songlin, Jan Kautz, and Ali Hatamizadeh. "Gated Delta Networks: Improving Mamba2 with Delta Rule." arXiv preprint arXiv:2412.06464 (2024).
> - Behrouz, Ali, et al. "Atlas: Learning to optimally memorize the context at test time." arXiv preprint arXiv:2505.23735 (2025).
> - Behrouz, Ali, et al. "It's All Connected: A Journey Through Test-Time Memorization, Attentional Bias, Retention, and Online Optimization." arXiv preprint arXiv:2504.13173 (2025).
>
> ---
>
> > **W2.** More discussion of what makes our particular setup novel compared to prior analyses of linear attention.
>
> **A2.** Prior analyses of linear attention's training dynamics usually consider merging key-query (e.g. $W:= W_Q W_K^\top$), specific initializations (e.g. $W_Q = W_K = I$), and gradient flow to simplify the optimization analysis, some of which have been discussed in lines 151-155.
>
> In contrast, our work:
>
> 1. Considers the dynamics for both $W_B$ and $W_C$, which will introduce non-convexity and complexity.
>
> 2. Initializes $W_B$ and $W_C$ with Gaussian distribution, which is more practical, and more difficult to establish the convergence given the non-convexity.
>
> 3. Trains the Mamba with gradient descent, which is typically more difficult to analyze than gradient flow.
>
> These setting are more realistic, and we propose some novel techniques to address the difficulties. The techniques are introduced in Section 5 (Proof Sketch). We will add more discussion in the next version.
>
> ---
>
> > **W3.** The current construction (unequal weighting) appears suboptimal vs. optimal linear attention (with equal weighting).
>
> **A3.**
> It is a limitation of Mamba under this construction. As for $\alpha$, see Eq.(11):
>
> $$h_l^{d+1} = \alpha h_{l-1}^{d+1} + (1-\alpha) y_l B_l$$
>
> $\alpha$ should be in (0, 1) to ensure that $h_l$ will not explode and can also utilize the current sample ($y_l B_l$). Therefore, the previous samples will be multiplied by a factor $\alpha < 1$ at each recurrent step, and thus the weighting on each sample is unequal.
>
> We chose a $\alpha$ close to 1 to ensure that all the samples share the same order of weights ($\Theta(1/N)$). Althought the weights are different and the Mamba achieves suboptimal prediction, it share the same order of the error ($O(1/N)$) with optimal linear attention.
>
> ---
>
> > **W4.** The experimental setup is simplistic.
>
> **A4.**
> Our experimental setup strictly follows our theoretical setup and assumption (e.g., one layer Mamba, no MLP). We have performed experiments for higher $d$ and $N=40$. The experimental results compare with theoretical results are as follows.
>
> |  d   | 6  | 8 | 10 | 12 | 14 | 16 | 18 | 20 |
> | --- | ---  | --- | --- | --- | --- | --- | --- | --- |
> | **experimental loss**  | 0.4738 | 0.7424 | 1.1706 | 1.4141 | 2.1359 | 2.4401 | 3.1154 | 3.9027 |
> | **theoretical loss** | 0.4618 | 0.7583 | 1.1117 | 1.5154 | 1.9636 | 2.4517 | 2.9753 | 3.5310 |
>
> The experimental results are closed to the theoretical results.
>
> ---
>
> > **W5.** Empirical comparison to linear attention baseline would be valuable.
>
> **A5.** Linear attention indeed outperforms Mamba, and they have $O(1/N)$ error upper bound with different constant factors. We provide a comprarison of loss between optimal Mamba (under our Assumption 4.1) with optimal linear attention as follows.
> setting: d=10, N=10, 20, ..., 80.
> |  N   | 10  | 20 | 30 | 40 | 50 | 60 | 70 | 80 |
> | --- | ---  | --- | --- | --- | --- | --- | --- | --- |
> | **Mamba**  | 2.6671 | 1.8189 | 1.3800 | 1.1117 | 0.9308 | 0.8005 | 0.7022 | 0.6254 |
> | **Linear Attention** | 2.6190 | 1.7742 | 1.3415 | 1.0784 | 0.9016 | 0.7746 | 0.6790 | 0.6044 |
>
> ---
>
> > **Q1.** **(1)** What is the underlying reason for this high dimension $d_h = \tilde{\Omega}(d^2)$?
> **(2)** If a smaller hidden dimension $d_h$ is used, are similar results still achievable? If so under what conditions?
>
> **A6.** **(1)** We require that the 2-norm of vectors $b_i, c_i, b$ are enough larger than the inner products between different vectors at initialization with high probability (e.g., $\| b_i \|_2^2 >> c_i^\top b_i$, proved in Lemma A.1, line. 473), so that we can establish the initial conditions $\mathcal{A}(0), \mathcal{B}(0)$, and $\mathcal{C}(0)$ for the induction (line. 306-310).
>
> **(2)** $d_h = O(d)$ is enough under some conditions.
>
> For example, if each column of $W_B$ and $W_C$ are initialized with orthogonal vectors (O(d) embedding dimension can ensure that), we can easily establish the convergence. We provide the discussion for orthogonal initialization in Appendix D (line. 905).
>
> Besides, we list some works that can address the similar optimization problem with $d_h = O(d)$ dimension as follows.
>
> - [Braun, Lukas, et al.] requires that the network's weight matrices are zero-balance at initialization, i.e., $W_1(0) W_1(0)^\top = W_2(0)^\top W_2(0)$
>
>
> - [Dominé, Clémentine CJ, et al.] requires that the network's weight matrices are $\lambda$-balance at initialization, i.e., $W_2(0) W_2(0)^\top - W_1(0)^\top W_1(0) = \lambda I$
>
> - [Arora, Sanjeev, et al.] requires that the weight matrices are initialized by Gaussian distribution with a very small variation O(1/poly(d)).
>
> Moreover, under standard gaussian distribution initialization $\mathcal{N}(0, 1)$, [Du, Simon, and Wei Hu.] require larger dimension $d_h$ than ours.
>
> Similar optimization problems in our paper can be solved through different theoretical techniques under different assumptions. We want to **highlight** that our method can provide some new insights for solving these problems and has the potential to be extended to various problems beyond it.
>
> Reference
> - Braun, Lukas, et al. "Exact learning dynamics of deep linear networks with prior knowledge." Advances in Neural Information Processing Systems 35 (2022): 6615-6629.
> - Dominé, Clémentine CJ, et al. "From lazy to rich: Exact learning dynamics in deep linear networks." arXiv preprint arXiv:2409.14623 (2024).
> - Arora, Sanjeev, et al. "A convergence analysis of gradient descent for deep linear neural networks." arXiv preprint arXiv:1810.02281 (2018).
> - Du, Simon, and Wei Hu. "Width provably matters in optimization for deep linear neural networks." International Conference on Machine Learning. PMLR, 2019.
>
> ---
>
> Once again, thanks for your constructive feedback and we hope our response can resolve your concerns!

---

> > ### Comment · Reviewer_8rgY · 2025-08-04
> >
> > Thank you for your response and it addressed most of my concerns.
> >
> > While regarding your response A5, I noticed that linear attention outperforms Mamba in your empirical results. This seems counter-intuitive, as Mamba is generally considered to be more expressive than linear attention in practice. Could you clarify more on the reason behind this observation?

---

> > > ### Author Response · Authors · 2025-08-05
> > >
> > > Thank you for your feedback.
> > >
> > > As you mentioned in **W3**, optimal linear attention assigns equal weights on each sample, while the Mamba (under our assumption) assigns different weights to each sample.
> > >
> > > In linear regression tasks, it is intuitive that equal weighting outperforms assigning different weights.
> > >
> > > We provide a brief explanation as follows.
> > > Considering the following two expressions for estimating $w$:
> > >
> > > $$ \hat{w} _ {1} = \sum _ {i=1} ^ N \alpha _ i y _ i x _ i ,  \quad \sum _ {i=1} ^ N \alpha _ i = 1$$
> > >
> > > $$ \hat{w} _ {2} = \sum _ {i=1} ^ N  \frac{1}{N} y _ i x _ i  $$
> > >
> > > It is clear that $\mathbb{E}[\hat{w} _ {1}] = \mathbb{E}[\sum_{i=1}^N \alpha_i]\mathbb{E}[y_i x_i ] = \mathbb{E}[\sum_{i=1}^N  \frac{1}{N}]\mathbb{E}[y_i x_i ] = \mathbb{E}[\hat{w} _ {2}]$. Although their expectations are equal, the variance of $\hat{w} _ {1}$ will be larger than that of $\hat{w} _ {2}$ (Cauchy–Schwarz inequality). The MSELoss will introduce the quadratic term (related to variance), thus the optimal linear attention achive a better MSELoss than that of Mamba.
> > >
> > > As for expressivity, we think it is reflected in Mamba's ability to simulate more functions (beyond linear regression) than linear attention.

---

> > > > ### Comment · Reviewer_8rgY · 2025-08-05
> > > >
> > > > I believe that the reason linear attention outperforms Mamba is that the weighting parameter $\alpha$ is fixed. In practice, however, $\alpha$ is not fixed but determined based on the input. Additionally, setting $\alpha=1/N$ yields the same behavior as linear attention, which suggests that Mamba should at least not be worse than linear attention once equal weighting is achievable. Therefore, this observation is not fully convincing.
> > > >
> > > > I encourage the authors to consider additional experiments that allow $\alpha$ to be learned or adapted online, or more realistic settings (such as making nearby in-context samples more relevant than those far away, and the optimal weighting would assign higher weights to closer examples).
> > > >
> > > > Overall, the authors have addressed most of my concerns, and I have decided to raise my score.

---

> > > > > ### Author Response · Authors · 2025-08-06
> > > > >
> > > > > Thank you for your feedback.
> > > > >
> > > > > We will include the additional experiments in the revised manuscript as you suggested.

---

### Note · Authors · 2025-08-12

We thank the AC and reviewers for their time and effort. We would like to briefly summarize our discussions and highlight the strengths of our work.

---

### **Summary of the Discussion**

We clarified the following points during the discussion period.

- **Q1:** The novelty should be better clarified.

  **A1:** Previous empirical works primarily leverage the connection between models and online-learning to design new architectures. However, the specific online learning pattern for Mamba is not straightforward. Moreover, prior theoretical work has not studied Mamba's ICL in linear regression.

- **Q2:** Is our assumption realistic and why is it novel?

  **A2:** Our assumption is more realistic compare to prior analysis on Transformer, and we provide experiments to verify the assumption. Furthermore, to address the optimization problem introduced by this assumption, we develop novel proof techniques.

- **Q3:** More experiments and comparison to linear attention.

  **A3:** Our experimental setup strictly follows our theoretical setup and assumption. We provide more experimental results in the rebuttal.

- **Q4:** The value of $\alpha$ should be learned.

  **A4:** Theoretically, trainable $\alpha$ will add complexity to the analysis, and a fixed $\alpha$ suffices for Mamba to learn linear regression in-context. Experiments support this choice.

- **Q5:** Expansion to complex tasks and multi-layer Mamba would be better.

  **A5:** We present a theoretical foundation for the study of Mamba's ICL ability, including data generation, training dynamics, and convergence guarantees. This foundation will inform understanding of Mamba's performance on more complex tasks and extensions to multi-layer models, and we leave it as an intriguing topic to study in the future.

We are pleased to hear that we have largely addressed the reviewers' concerns, and will incorporate valuable presentation and clarity suggestions from all four reviewers into the final version.

---

### **Strengths Highlighted by Reviewers**

**Insightful Theoretical Findings:** The theoretical results provide valuable insights into the differences in the capabilities between SSM and transformers.

**Empirical Validation:** The theoretical findings are supported by experiments.

**Novel Techniques:** The training dynamics of the Mamba-like model is important and non-trivial, this paper proposes new techniques to analyze it.

**Clarity of Writing:** Paper is well written and clearly organized.

---

### Decision · Program_Chairs · 2025-09-17

**Decision:**

Accept (poster)

**Comment:**

This paper provides a rigorous analysis of the in-context learning (ICL) capabilities of Mamba, a popular sequence model. It follows a line of recent work analytically studying various sequence models on in-context linear regression tasks. The analysis focuses on 1-layer, no MLP models. Interestingly, it reveals that Mamba learns to implement a form of online gradient descent, which differs from the full-batch 1-step gradient descent method that (linear) self-attention has been shown to learn in various previous analyses.

This is a technically strong contribution on an actively researched topic. Although there are by now many analyses of the learning dynamics of self-attention on in-context linear regression, much less is known about modern state-space models, a model class of which Mamba is a prominent member. The paper reveals interesting differences to linear self-attention, and the analysis requires mild assumptions only. In particular, it applies to randomly initialized Mamba layers. An obvious limitation is its focus on 1-layer, no-MLP networks, but the reviewers and I agree that this is not sufficient to preclude from publication.